# A three filament mechanistic model of musculotendon force and impedance

**Matthew Millard[1,2]\*[†], David W Franklin[3,4,5][‡], Walter Herzog[6][§]**

[1]Institute for Sport and Movement Science, University of Stuttgart, Stuttgart, Germany; [2]Institute of Engineering and Computational Mechanics, University of Stuttgart, Stuttgart, Germany; [3]Neuromuscular Diagnostics, TUM School of Medicine and Health, Technical University of Munich, Munich, Germany; [4]Munich School of Robotics and Machine Intelligence (MIRMI), Technical University of Munich, Munich, Germany; [5]Munich Data Science Institute (MDSI), Technical University of Munich, Munich, Germany; [6]Human Performance Laboratory, University of Calgary, Calgary, Canada

**\*For correspondence:**
matthew.millard@inspo.uni-stuttgart.de

**Present address:** [†]University of Stuttgart, Stuttgart, Germany; [‡]Technical University of Munich, Munich, Germany; [§]University of Calgary, Calgary, Canada

**Competing interest:** The authors declare that no competing interests exist.

**Abstract** The force developed by actively lengthened muscle depends on different structures across different scales of lengthening. For small perturbations, the active response of muscle is well captured by a linear-time-invariant (LTI) system: a stiff spring in parallel with a light damper. The force response of muscle to longer stretches is better represented by a compliant spring that can fix its end when activated. Experimental work has shown that the stiffness and damping (impedance) of muscle in response to small perturbations is of fundamental importance to motor learning and mechanical stability, while the huge forces developed during long active stretches are critical for simulating and predicting injury. Outside of motor learning and injury, muscle is actively lengthened as a part of nearly all terrestrial locomotion. Despite the functional importance of impedance and active lengthening, no single muscle model has all these mechanical properties. In this work, we present the viscoelastic-crossbridge active-titin (VEXAT) model that can replicate the response of muscle to length changes great and small. To evaluate the VEXAT model, we compare its response to biological muscle by simulating experiments that measure the impedance of muscle, and the forces developed during long active stretches. In addition, we have also compared the responses of the VEXAT model to a popular Hill-type muscle model. The VEXAT model more accurately captures the impedance of biological muscle and its responses to long active stretches than a Hill-type model and can still reproduce the force-velocity and force-length relations of muscle. While the comparison between the VEXAT model and biological muscle is favorable, there are some phenomena that can be improved: the low frequency phase response of the model, and a mechanism to support passive force enhancement.

## eLife assessment

This is a **valuable** study that develops a new model of the way muscle responds to perturbations, synthesizing models of how it responds to small and large perturbations, both of which are used to predict how muscles function for stability but also how they can be injured, and which tend to be predicted poorly by classic Hill-type models. The evidence presented to support the model is **solid**, since it outperforms Hill-type models in a variety of conditions. Although the combination of phenomenological and mechanistic aspects of the model may sometimes make it challenging to interpret the output, the work will be of interest to those developing realistic models of the stability and control of movement in humans or other animals.

## Introduction

The stiffness and damping of muscle are properties of fundamental importance for motor control, and the accurate simulation of muscle force. The CNS exploits the activation-dependent stiffness and damping (impedance) of muscle when learning new movements (*Franklin et al., 2003*), and when moving in unstable (*Trumbower et al., 2009*) or noisy environments (*Selen et al., 2009*). Reaching experiments using haptic manipulanda show that the CNS uses co-contraction to increase the stiffness of the arm when perturbed by an unstable force field (*Burdet et al., 2001*). With time and repetition, the force field becomes learned and co-contraction is reduced (*Franklin et al., 2003*).

The force response of muscle is not uniform, but varies with both the length and time of perturbation. Under constant activation and at a consistent nominal length, *Kirsch et al., 1994* were able to show that muscle behaves like a linear-time-invariant (LTI) system in response to small perturbations (see Appendix 9, Note 1): a spring-damper of best fit captured over 90% of the observed variation in muscle force for small perturbations (1–3.8% optimal length) over a wide range of bandwidths (4–90 Hz). When active muscle is stretched appreciably, titin can develop enormous forces (*Herzog and Leonard, 2002*; *Leonard et al., 2010*), which may prevent further lengthening and injury. The stiffness that best captures the response of muscle to the small perturbations of *Kirsch et al., 1994* is far greater than the stiffness that best captures the response of muscle to large perturbations (*Herzog and Leonard, 2002*; *Leonard et al., 2010*). Since everyday movements are often accompanied by both large and small kinematic perturbations, it is important to accurately capture these two processes.

However, there is likely no single muscle model that can replicate the force response of muscle to small (*Kirsch et al., 1994*) and large perturbations (*Herzog and Leonard, 2002*; *Leonard et al., 2010*) while also retaining the capability to reproduce the experiments of *Hill, 1938* and *Gordon et al., 1966*. Unfortunately, this means that simulation studies that depend on an accurate replication of the perturbation response may reach conclusions well justified in simulation but not in reality. In this work, we focus on formulating a mechanistic muscle model (see Appendix 9, Note 2), that can replicate the force response of active muscle to length perturbations both great and small.

There are predominantly three classes of models that are used to simulate musculoskeletal responses: phenomenological models constructed using the famous force-velocity relationship of *Hill, 1938*, mechanistic Huxley (*Huxley, 1957*; *Huxley and Simmons, 1971*; *Kosta et al., 2022*) models in which individual elastic crossbridges are incorporated, and linearized muscle models (*Hogan, 1985*; *Mussa-Ivaldi et al., 1985*) which are accurate for small changes in muscle length. *Kirsch et al., 1994* demonstrated that, for small perturbations, the force response of muscle is well represented by an activation-dependent spring and damper that are connected in parallel. Neither Hill nor Huxley models are likely to replicate the experiments of *Kirsch et al., 1994* because a Hill muscle model (*Zajac, 1989*; *Millard et al., 2013*) does not contain any active spring elements; while a Huxley model lacks an active damping element. Although linearized muscle models can replicate the experiment of *Kirsch et al., 1994*, these models are only accurate for small changes in length and cannot replicate the nonlinear force-velocity relation of *Hill, 1938*, nor the nonlinear force-length relation of *Gordon et al., 1966*. However, there have been significant improvements to the canonical forms of phenomenological, mechanistic, and linearized muscle models that warrant closer inspection.

Several novel muscle models have been proposed to improve upon the accuracy of Hill-type muscle models during large active stretches. *Forcinito et al., 1998* modeled the velocity dependence of muscle using a rheological element (see Appendix 9, Note 3) and an elastic rack rather than embedding the force-velocity relationship in equations directly, as is done in a typical Hill model (*Zajac, 1989*; *Millard et al., 2013*). This modification allows the model of *Forcinito et al., 1998* to more faithfully replicate the force development of active muscle, as compared to a Hill-type model, during ramp length changes of ≈10% (see Appendix 9, Note 4) of the optimal CE length, and across velocities of 4–11% of the maximum contraction velocity (see Appendix 9, Note 5). *Tamura and Saito, 2002* extended the work of *Forcinito et al., 1998* by formulating a rheological muscle model with two Maxwell elements (a spring-damper in series) where one develops force quickly (high stiffness) and the other develops force slowly (low stiffness). By carefully selecting the dynamics that drive the two elements, the model of *Tamura and Saito, 2002* replicated the force-length-velocity relations (*Gordon et al., 1966*; *Hill, 1938*) as well as qualitatively reproducing the force and stiffness profiles (*Tamura et al., 2005*) of force-enhancement and force-depression as measured by *Sugi and Tsuchiya,*

*1988*. *Haeufle et al., 2014* made use of a serial-parallel network of spring-dampers to allow their model to reproduce the force-velocity relationship of *Hill, 1938* mechanistically rather than embedding the experimental curve directly in their model. *Günther et al., 2018* evaluated how accurately a variety of spring-damper models were able to reproduce the microscopic increases in crossbridge force in response to small length changes. While each of these models improves upon the force response of the Hill model to ramp length changes, none are likely to reproduce the experiment of *Kirsch et al., 1994* because the linearized versions of these models lead to a serial, rather than a parallel, connection of a spring and a damper: *Kirsch et al., 1994* specifically showed (see Figure 3 of *Kirsch et al., 1994*) that a serial connection of a spring-damper fails to reproduce the phase shift between force and length present in their experimental data.

Titin (*Maruyama, 1976*; *Wang et al., 1979*) has been more recently investigated to explain how lengthened muscle can develop active force when lengthened both within, and beyond, actin-myosin overlap (*Leonard et al., 2010*). Titin is a gigantic multi-segmented protein that spans a half-sarcomere, attaching to the Z-line at one end and the middle of the thick filament at the other end (*Maruyama et al., 1985*). In skeletal muscle, the two sections nearest to the Z-line, the proximal immunoglobulin (IgP) segment and the PEVK segment — rich in the amino acids proline (P), glutamate (E), valine (V) and lysine (K) — are the most compliant (*Trombitás et al., 1998b*) since the distal immunoglobulin (IgD) segments bind strongly to the thick filament (*Houmeida et al., 1995*). Titin has proven to be a complex filament, varying in composition and geometry between different muscle types (*Tomalka et al., 2019*; *Boldt et al., 2020*), widely between species (*Lindstedt and Nishikawa, 2017*), and can apply activation dependent forces to actin (*Kellermayer and Granzier, 1996*). It has proven challenging to determine which interactions dominate between the various segments of titin and the other filaments in a sarcomere. Experimental observations have reported titin-actin interactions at myosin-actin binding sites (*Astier et al., 1998*; *Niederländer et al., 2004*), between titin's PEVK region and actin (*Bianco et al., 2007*; *Nagy et al., 2004*), between titin's N2A region and actin (*Dutta et al., 2018*), and between the PEVK-IgD regions of titin and myosin (*DuVall et al., 2017*). This large variety of experimental observations has led to a correspondingly large number of proposed hypotheses and models, most of which involve titin interacting with actin (*Rode et al., 2009*; *Nishikawa et al., 2012*; *Schappacher-Tilp et al., 2015*; *Tahir et al., 2018*; *Heidlauf et al., 2016*; *Heidlauf et al., 2017*), and more recently with myosin (*DuVall et al., 2016*).

The addition of a titin element to a model will result in more accurate force production during large active length changes, but does not affect the stiffness and damping of muscle at modest sarcomere lengths because of titin's relatively low stiffness. At sarcomere lengths of $1.62\ell_o^{\mathrm{M}}$ or less, the stiffness of the actin-myosin load path with a single attached crossbridge (0.22–1.15pN/nm) equals or exceeds the stiffness of 6 passive titin filaments (0.0348–0.173pN/nm), and our estimated stiffness of 6 active titin filaments (0.0696–0.346pN/nm, see Appendix 1 for further details). When fully activated, the stiffness of the actin-myosin load path (4.05–18.4pN/nm) far exceeds that of both the passive titin (0.0348–0.173pN/nm), and our estimated active titin ($0.0696 - 0.346\mathrm{pN/nm}$) load paths. Since titin-focused models have not made any changes to the modeled myosin-actin interaction beyond a Hill (*Zajac, 1989*; *Millard et al., 2013*) or Huxley (*Huxley, 1957*; *Huxley and Simmons, 1971*) model, it is unlikely that these models would be able to replicate the experiments of *Kirsch et al., 1994*.

Although most motor control simulations (*Kearney et al., 1997*; *Perreault et al., 2004*; *Schouten et al., 2008*; *Trumbower et al., 2009*; *Mitrovic et al., 2010*) make use of the canonical linearized muscle model, phenomenological muscle models have also been used and modified to include stiffness. *Sartori et al., 2015* modeled muscle stiffness by evaluating the partial derivative of the force developed by a Hill-type muscle model with respect to the contractile element (CE) length. Although this approach is mathematically correct, the resulting stiffness is heavily influenced by the shape of the force-length curve and can lead to inaccurate results: at the optimal CE length this approach would predict an active muscle stiffness of zero since the slope of the force-length curve is zero; on the descending limb this approach would predict a negative active muscle stiffness since the slope of the force-length curve is negative. In contrast, CE stiffness is large and positive near the optimal length (*Kirsch et al., 1994*), and there is no evidence for negative stiffness on the descending limb of the force-length curve (*Herzog and Leonard, 2002*). Although the stiffness of the CE can be kept positive by shifting the passive force-length curve, which is at times used in finite-element-models of muscle (*Heidlauf et al., 2017*), this introduces a new problem: the resulting passive CE stiffness cannot be

lowered to match a more flexible muscle. In contrast, *De Groote et al., 2017* and *De Groote et al., 2018* modeled short-range-stiffness using a stiff spring in parallel with the active force element of a Hill-type muscle model. While the approach developed in *De Groote et al., 2017* and *De Groote et al., 2018* likely does improve the response of a Hill-type muscle model for small perturbations, there are several drawbacks: the short-range-stiffness of the muscle sharply goes to zero outside of the specified range whereas in reality the stiffness is only reduced (*Kirsch et al., 1994*, see Figure 9A); the damping of the canonical Hill-model has been left unchanged and likely differs substantially from biological muscle (*Kirsch et al., 1994*).

In this work, we propose a model that can capture the force development of muscle to perturbations that vary in size and timescale, and yet is described using only a few states making it well suited for large-scale simulations. When active, the response of the model to perturbations within actin-myosin overlap is dominated by a viscoelastic crossbridge element that has different dynamics across time-scales: over brief time-scales the viscoelasticity of the lumped crossbridge dominates the response of the muscle (*Kirsch et al., 1994*), while over longer time-scales the force-velocity (*Hill, 1938*) and force-length (*Gordon et al., 1966*) properties of muscle dominate. To capture the active forces developed by muscle beyond actin-myosin overlap we added an active titin element which, similar to the models of *Rode et al., 2009* and (*Schappacher-Tilp et al., 2015*), features an activation-dependent (see Appendix 9, Note 6) interaction between titin and actin. To ensure that the various parts of the model are bounded by reality, we have estimated the physical properties of the visco-elastic crossbridge element as well as the active titin element using data from the literature.

While our main focus is to develop a more accurate muscle model, we would like the model to be well suited to simulating systems that contain tens to hundreds of muscles. Although Huxley models have been used to simulate whole-body movements such as jumping (*van Soest et al., 2019*), the

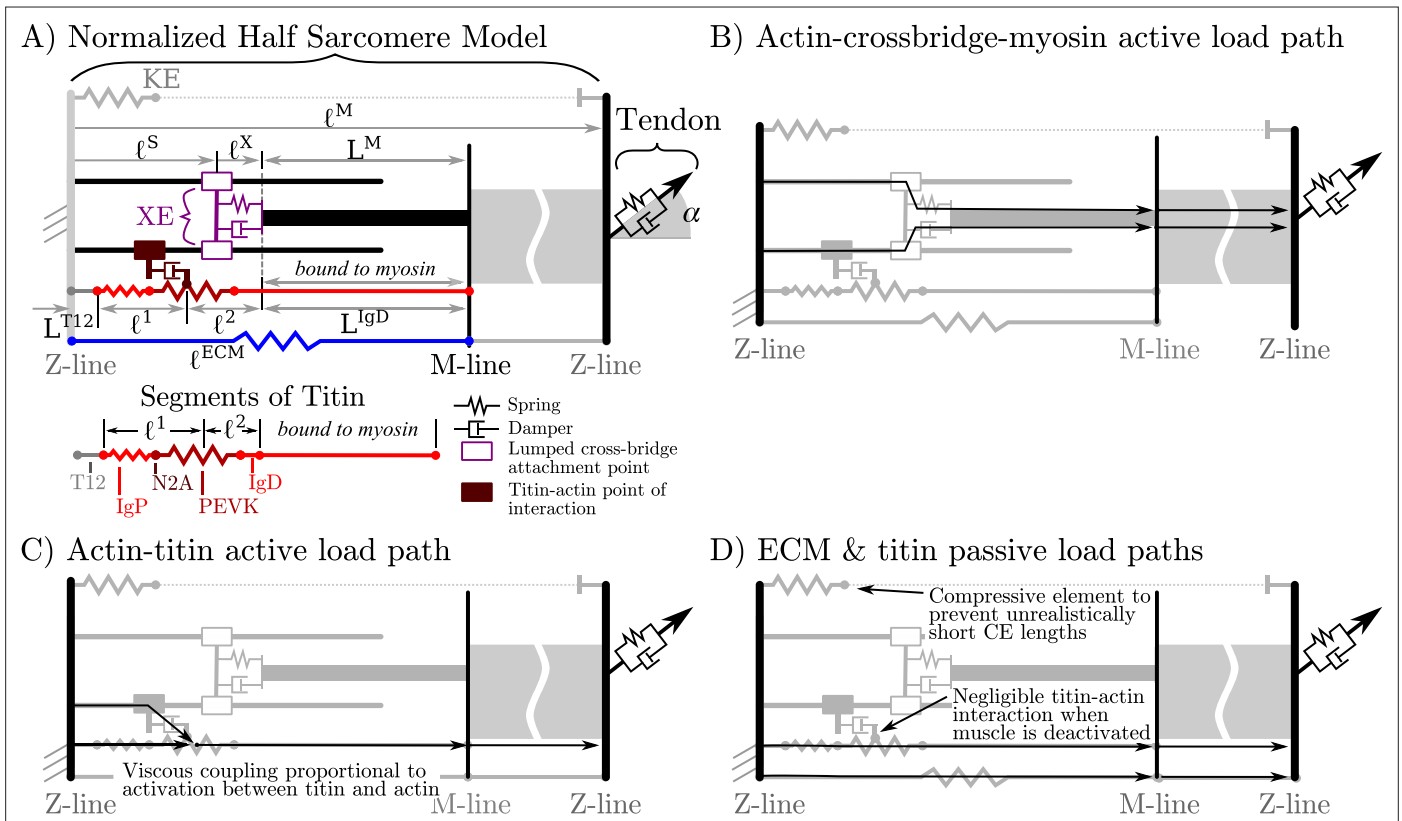

**Figure 1.** Overview of the VEXAT model and its components. The name of the VEXAT model comes from the viscoelastic crossbridge and active titin elements (**A**) in the model. Active tension generated by the lumped crossbridge flows through actin, myosin, and the adjacent sarcomeres to the attached tendon (**B**). Titin is modeled as two springs of length $\ell^1$ and $\ell^2$ in series with the rigid segments $L^{T12}$ and $L^{IgD}$. Viscous forces act between titin and actin in proportion to the activation of the muscle (**C**), which reduces to negligible values in a purely passive muscle (**D**). We modeled actin and myosin as rigid elements; the XE, titin, and the tendon as viscoelastic elements; and the ECM as an elastic element.

memory and processing requirements associated with simulating a single muscle with thousands of states is high. Instead of modeling the force development of individual crossbridges, we lump all of the crossbridges in a muscle together so that we have a small number of states to simulate per muscle.

To evaluate the proposed model, we compare simulations of experiments to original data. We examine the response of active muscle to small perturbations over a wide bandwidth by simulating the stochastic perturbation experiments of *Kirsch et al., 1994*. The active-lengthening experiments of *Herzog and Leonard, 2002* are used to evaluate the response of the model when it is actively lengthened within actin-myosin overlap. Next, we use the active-lengthening experiments of *Leonard et al., 2010* to see how the model compares to reality when it is actively lengthened beyond actin-myosin overlap. In addition, we examine how well the model can reproduce the force-velocity experiments of *Hill, 1938* and force-length experiments of *Gordon et al., 1966*. Since Hill-type models are so commonly used, we also replicate all of the simulated experiments using the Hill-type muscle model of *Millard et al., 2013* to make the differences between these two types of models clear.

## Model

We begin by treating whole muscle as a scaled half-sarcomere that is pennated at an angle $\alpha$ with respect to a tendon (*Figure 1A*). The assumption that mechanical properties scale with size is commonly used when modeling muscle (*Zajac, 1989*) and makes it possible to model vastly different musculo-tendon units (MTUs) by simply changing the architectural and contraction properties: the maximum isometric force $f_o^M$, the optimal CE length $\ell_o^M$ (at which the CE develops $f_o^M$), the pennation angle $\alpha_o$ of the CE (at a length of $\ell_o^M$) with respect to the tendon, the maximum shortening velocity $v_{max}^M$ of the CE, and the slack length of the tendon $\ell_s^T$. Many properties of sarcomeres scale with $f_o^M$ and $\ell_o^M$: $f_o^M$ scales with physiological cross-sectional area (*Maganaris et al., 2001*), the force-length property scales with $\ell_o^M$ (*Winters et al., 2011*), the maximum normalized shortening velocity of different CE types scales with $\ell_o^M$ across animals great and small (*Rome et al., 1990*), and titin's passive-force-length properties scale from single molecules to myofibrils (*Herzog et al., 2012*; *Prado et al., 2005*).

The proposed model has several additional properties that we assume scale with $f_o^M$ and inversely with $\ell_o^M$: the maximum active isometric stiffness $k_o^X$ and damping $\beta_o^X$, the passive forces due to the extracellular matrix (ECM), and passive forces due to titin. As crossbridge stiffness is well studied (*Kaya and Higuchi, 2013*), we assume that muscle stiffness due to crossbridges scales such that

$$k_o^X = \tilde{k}_o^X \frac{f_o^M}{\ell_o^M},$$

(1)

where $\tilde{k}_o^X$ is the maximum normalized stiffness. This scaling is just what would be expected when many crossbridges (*Kaya and Higuchi, 2013*) act in parallel across the cross-sectional area of the muscle, and act in series along the length of the muscle. Although the intrinsic damping properties of crossbridges are not well studied, we assume that the linear increase in damping with activation observed by *Kirsch et al., 1994* is due to the intrinsic damping properties of individual crossbridges which will also scale linearly with $f_o^M$ and inversely with $\ell_o^M$

$$\beta_o^X = \tilde{\beta}_o^X \frac{f_o^M}{\ell_o^M},$$

(2)

where $\tilde{\beta}_o^X$ is the maximum normalized damping. For the remainder of the paper, we refer to the proposed model as the VEXAT model due to the viscoelastic (VE) crossbridge (X) and active-titin (AT) elements of the model.

To reduce the number of states needed to simulate the VEXAT model, we lump all of the attached crossbridges into a single lumped crossbridge element (XE) that attaches at $\ell^S$ (*Figure 1A*) and has intrinsic stiffness and damping properties that vary with the activation and force-length properties of muscle. The active force developed by the XE at the attachment point to actin is transmitted to the main myosin filament, the M-line, and ultimately to the tendon (*Figure 1B*). In addition, since the stiffness of actin (*Higuchi et al., 1995*) and myosin filaments (*Tajima et al., 1994*) greatly exceeds that of crossbridges (*Veigel et al., 1998*), we treat actin and myosin filaments as rigid to reduce the number of states needed to simulate this model. Similarly, we have lumped the six titin filaments per

half-sarcomere (*Figure 1A*) together to further reduce the number of states needed to simulate this model.

The addition of a titin filament to the model introduces an additional active load-path (*Figure 1C*) and an additional passive load-path (*Figure 1D*). As is typical (*Zajac, 1989*; *Millard et al., 2013*), we assume that the passive elasticity of these structures scale linearly with $f_o^M$ and inversely with $\ell_o^M$. Since the VEXAT model has two passive load paths (*Figure 1D*), we further assume that the proportion of the passive force due to the extra-cellular-matrix (ECM) and titin does not follow a scale-dependent pattern, but varies from muscle-to-muscle as observed by *Prado et al., 2005*.

As previously mentioned, there are several theories to explain how titin interacts with the other filaments in activated muscle. While there is evidence for titin-actin interaction near titin's N2A region (*Dutta et al., 2018*), there is also support for a titin-actin interaction occurring near titin's PEVK region (*Bianco et al., 2007*; *Nagy et al., 2004*), and for a titin-myosin interaction near the PEVK-IgD region (*DuVall et al., 2017*). For the purposes of our model, we will assume a titin-actin interaction because current evidence weighs more heavily towards a titin-actin interaction than a titin-myosin interaction. Next, we assume that the titin-actin interaction takes place somewhere in the PEVK segment for two reasons: first, there is evidence for a titin-actin interaction (*Bianco et al., 2007*; *Nagy et al., 2004*) in the PEVK segment; and second, there is evidence supporting an interaction at the proximal end of the PEVK segment (*Dutta et al., 2018*, N2A-actin interaction). We have left the point within the PEVK segment that attaches to actin as a free variable since there is some uncertainty about what part of the PEVK segment interacts with actin.

The nature of the mechanical interaction between titin and the other filaments in an active sarcomere remains uncertain. Here, we assume that this interaction is not a rigid attachment, but instead is an activation dependent damping to be consistent with the observations of *Kellermayer and Granzier, 1996* and *Dutta et al., 2018*: adding titin filaments and calcium slowed, but did not stop, the progression of actin filaments across a plate covered in active crossbridges (an in vitro motility assay). When activated, we assume that the amount of damping between titin and actin scales linearly with $f_o^M$ and inversely with $\ell_o^M$.

After lumping all of the crossbridges and titin filaments together, we are left with a rigid-tendon MTU model that has two generalized positions

$$\underline{q}^R = (\ell^S, \ell^1) \tag{3}$$

and an elastic-tendon MTU model that has three generalized positions

$$\underline{q}^E = (\ell^M, \ell^S, \ell^1). \tag{4}$$

Given these generalized positions, the path length $\ell^P$, and a pennation model, all other lengths in the model can be calculated. Here, we use a constant thickness

$$\mathrm{H} = \ell_o^M \sin \alpha_o \tag{5}$$

pennation model to evaluate the pennation angle

$$\alpha = \arctan \left( \frac{\mathrm{H}}{\ell^P - \ell_s^T} \right) \tag{6}$$

of a CE with a rigid-tendon, and

$$\alpha = \arcsin \left( \frac{\mathrm{H}}{\ell^M} \right) \tag{7}$$

to evaluate the pennation angle of a CE with an elastic-tendon. We have added a small compressive element KE (*Figure 1A*) to prevent the model from reaching the numerical singularity that exists as $\tilde{\ell}^M$ approaches $\tilde{\ell}_{\min}^M$, the length at which $\alpha \to 90°$ in *Equation 6* and *Equation 7*. The tendon length

$$\ell^T = \ell^P - \ell^M \cos \alpha, \tag{8}$$

of an elastic-tendon model is the difference between the path length and the CE length along the tendon. The length of the XE

$$\ell^X = \frac{1}{2}\ell^M - \left(\ell^S + L^M\right) \tag{9}$$

is the difference between the half-sarcomere length and the sum of the average point of attachment $\ell^S$ and the length of the myosin filament $L^M$. The length of $\ell^2$, the lumped PEVK-IgD segment, is

$$\ell^2 = \frac{1}{2}\ell^M - \left(\ell^1 + L^{T12} + L^{IgD}\right) \tag{10}$$

the difference between the half-sarcomere length and the sum of the length from the Z-line to the actin binding site on titin ($\ell^1 + L^{T12}$) and the length of the IgD segment that is bound to myosin ($L^{IgD}$). Finally, the length of the extra-cellular-matrix $\ell^{ECM}$ is simply

$$\ell^{ECM} = \frac{1}{2}\ell^M \tag{11}$$

half the length of the CE since we are modeling a half-sarcomere.

We have some freedom to choose the state vector of the model and the differential equations that define how the muscle responds to length and activation changes. The experiments we hope to replicate depend on phenomena that take place at different time-scales: the stochastic perturbations of *Kirsch et al., 1994* evolve over brief time-scales, while all of the other experiments take place at much longer time-scales. Here, we mathematically decouple phenomena that affect brief and long time-scales by making a second-order model that has states of the average point of crossbridge attachment $\ell^S$, and velocity $v^S$. When the activation state $a$ and the titin-actin interaction model are included, the resulting rigid-tendon model has a total of four states

$$\underline{x} = \left(a, v^S, \ell^S, \ell^1\right) \tag{12}$$

and the elastic-tendon model has

$$\underline{x} = \left(a, v^S, \ell^S, \ell^1, \ell^M\right) \tag{13}$$

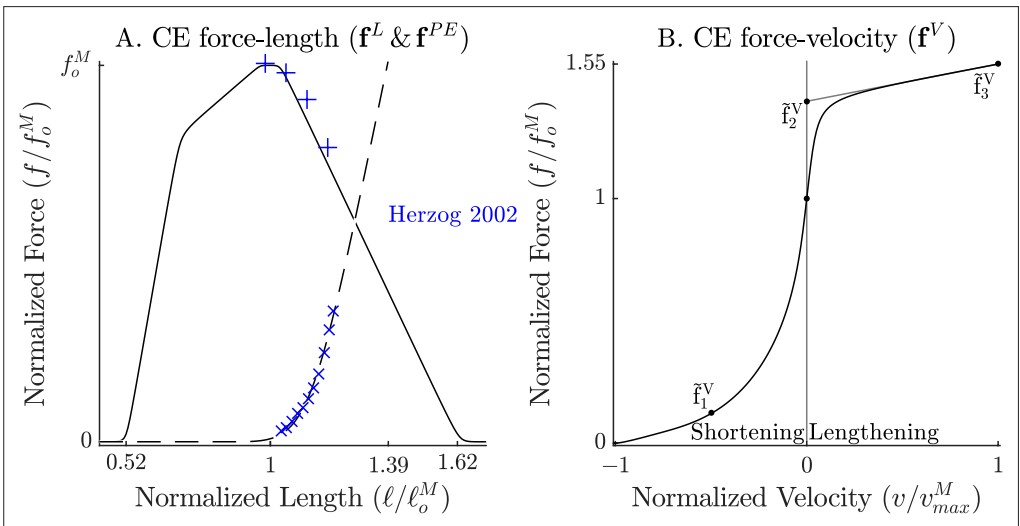

**Figure 2.** The force-length and force-velocity relations of the cat soleus model. The model relies on Bézier curves to model the nonlinear effects of the active-force-length curve, the passive-force-length curve (**A**), and the force-velocity curve (**B**). Since nearly all of the reference experiments used in the 'Biological benchmark simulations' section have used cat soleus, we have fit the active-force-length curve ($\mathbf{f}^L(\cdot)$) and passive-force-length curves ($\mathbf{f}^{PE}(\cdot)$) to the cat soleus data of *Herzog and Leonard, 2002*. The concentric side of the force-velocity curve ($\mathbf{f}^V(\cdot)$) has been fitted to the cat soleus data of *Herzog and Leonard, 1997*.

five states. For the purpose of comparison, a Hill-type muscle model with a rigid-tendon has a single state ($a$), while an elastic-tendon model has two states ($a$ and $\ell^M$) (*Millard et al., 2013*).

Before proceeding, a small note on notation: throughout this work we will use an underbar to indicate a vector, bold font to indicate a curve, a tilde for a normalized quantity, and a capital letter to indicate a constant. Unless indicated otherwise, curves are constructed using $C^2$ continuous (see Appendix 9, Note 7) Bézier splines so that the model is compatible with gradient-based optimization. Normalized quantities within the CE follow a specific convention: lengths and velocities are normalized by the optimal CE length $\ell^M_o$, forces by the maximum active isometric tension $f^M_o$, stiffness and damping by $f^M_o/\ell^M_o$. Velocities used as input to the force-velocity relation $\mathbf{f}^V$ are further normalized by $v^M_{max}$ and annotated using a hat: $\hat{v}^M = v^M/v^M_{max}$. Tendon lengths and velocities are normalized by $\ell^T_s$ tendon slack length, while forces are normalized by $f^M_o$.

To evaluate the state derivative of the model, we require equations for $\dot{a}$, $\dot{v}^S$, $v^1$, and $v^M$ if the tendon is elastic. For $\dot{a}$ we use of the first order activation dynamics model described in *Millard et al., 2013* (see Appendix 9, Note 8) which uses a lumped first-order ordinary-differential-equation (ODE) to describe how a fused tetanus electrical excitation leads to force development in an isometric muscle. We formulated the equation for $\dot{v}^S$ with the intention of having the model behave like a spring-damper over small time-scales, but to converge to the tension developed by a Hill-type model

$$\tilde{f}^M = a\,\mathbf{f}^L(\tilde{\ell}^M)\mathbf{f}^V(\tilde{v}^M) + \mathbf{f}^{PE}(\tilde{\ell}^M) \tag{14}$$

over longer time-scales, where $\mathbf{f}^L(\cdot)$ is the active-force-length curve (*Figure 2A*), $\mathbf{f}^{PE}(\cdot)$ is the passive-force-length curve (*Figure 2A*), and $\mathbf{f}^V(\cdot)$ is the force-velocity curve (*Figure 2B*).

The normalized tension developed by the VEXAT model

$$\tilde{f}^M = a\mathbf{f}^L(\tilde{\ell}^S + \tilde{L}^M)(\tilde{k}^X_o\tilde{\ell}^X + \tilde{\beta}^X_o\tilde{v}^X) + \mathbf{f}^2(\tilde{\ell}^2) + \mathbf{f}^{ECM}(\tilde{\ell}^{ECM}) + \tilde{\beta}^{\,\epsilon}\tilde{v}^M - \frac{\mathbf{f}^{KE}(\tilde{\ell}^M)}{\cos\alpha} \tag{15}$$

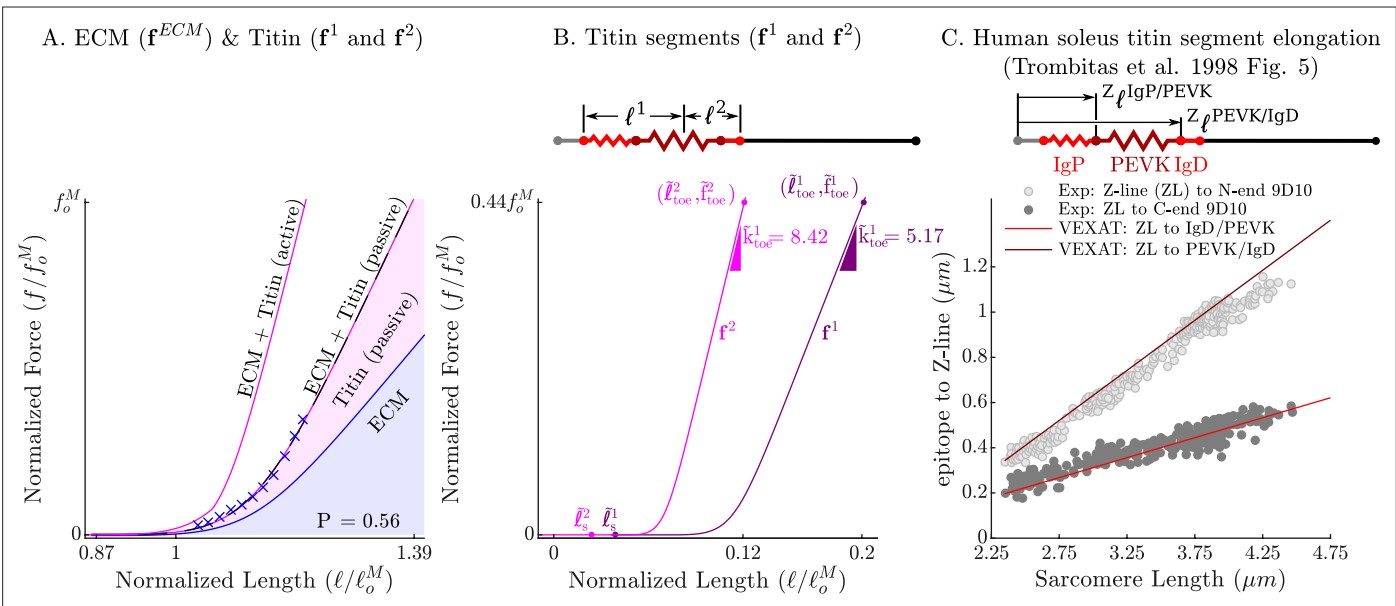

**Figure 3.** The passive force-length relations of the ECM, titin, and titin's segments. The passive force-length curve has been decomposed such that 56% of it comes from the ECM while 44% comes from titin to match the average of ECM-titin passive force distribution (which ranges from 43% to 76%) reported by *Prado et al., 2005* (**A**). The elasticity of the titin segment has been further decomposed into two serially connected sections: the proximal section consisting of the T12, proximal IgP segment and part of the PEVK segment, and the distal section consisting of the remaining PEVK section and the distal Ig segment (**B**) The stiffness of the IgP and PEVK segments has been chosen so that the model can reproduce the movements of IgP/PEVK and PEVK/IgD boundaries that *Trombitás et al., 1998b* (**C**) observed in their experiments. The curves that appear in subplots A and B come from scaling the two-segmented human soleus titin model to cat soleus muscle. The curves that appear in subplot C compare the human soleus titin model's IgP, PEVK, and IgD force-length relations to the data of *Trombitás et al., 1998b* (see in Appendix 2 for details).

differs from that of a Hill model, *Equation 14*, because it has no explicit dependency on $\tilde{v}^M$, includes four passive terms, and a lumped viscoelastic crossbridge element. The four passive terms come from the ECM element $\mathbf{f}^{ECM}(\tilde{\ell}^{ECM})$ (*Figure 3A*), the PEVK-IgD element $\mathbf{f}^2(\tilde{\ell}^2)$ (*Figure 3A and B*), the compressive term $\mathbf{f}^{KE}(\tilde{\ell}^M)$ (prevents $\tilde{\ell}^M \cos\alpha$ from reaching a length of 0), and a numerical damping term $\tilde{\beta}^{\epsilon}\tilde{v}^M$ (where $\tilde{\beta}^{\epsilon}$ is small). The active force developed by the XE's lumped crossbridge $\tilde{k}_o^X\tilde{\ell}^X + \tilde{\beta}_o^X\tilde{v}^X$ is scaled by the fraction of the XE that is active and attached, $a\mathbf{f}^L(\tilde{\ell}^S + \tilde{L}^M)$, where $\mathbf{f}^L(\cdot)$ is the active-force-length relation (*Figure 2A*). We evaluate $\mathbf{f}^L$ using $\tilde{\ell}^S + \tilde{L}^M$ in *Equation 15*, rather than $\tilde{\ell}^M$ as in *Equation 14*, since actin-myosin overlap is independent of crossbridge strain. With $\tilde{f}^M$ derived, we can proceed to model the acceleration of the CE, $\dot{v}^S$, so that it is driven over time by the force imbalance between the XE's active tension and that of a Hill model.

We set the first term of $\dot{v}^S$ so that, over time, the CE is driven to develop the same active tension as the Hill-type model of *Millard et al., 2013* (terms highlighted in blue)

$$\dot{v}^S = \frac{a\mathbf{f}^L(\tilde{\ell}^S + \tilde{L}^M)(\tilde{k}_o^X\tilde{\ell}^X + \tilde{\beta}_o^X\tilde{v}^X) - a\mathbf{f}^L(\tilde{\ell}^S + \tilde{L}^M)\mathbf{f}^V(\hat{v}^S)}{\tau^S} - D\tilde{v}^S + e^{-(a/a_L)^2}(G_L\tilde{\ell}^X + G_V\tilde{v}^X) \quad (16)$$

where $\tau^S$ is a time constant and $\mathbf{f}^V(\hat{v}^S)$ is the force-velocity curve (*Figure 2B*). The rate of adaptation of the model's tension, to the embedded Hill model, is set by the time constant $\tau^S$: as $\tau^S$ is decreased the VEXAT model converges more rapidly to a Hill-type model; as $\tau^S$ is increased the active force produced by the model will look more like a spring-damper. Our preliminary simulations indicate that there is a trade-off to choosing $\tau^S$: when $\tau^S$ is large the model will not shorten rapidly enough to replicate Hill's experiments, while if $\tau^S$ is small the low-frequency response of the model is compromised when the experiments of *Kirsch et al., 1994* are simulated.

The remaining two terms, $D\tilde{v}^S$ and $e^{-(a/a_L)^2}(G_L\tilde{\ell}^X + G_V\tilde{v}^X)$, have been included for numerical reasons specific to this model formulation rather than muscle physiology. We include a term that damps the rate of actin-myosin translation, $D\tilde{v}^S$, to prevent this second-order system from unrealistically oscillating (see Appendix 9, Note 9). The final term $e^{-(a/a_L)^2}(G_L\tilde{v}^X + G_V\tilde{v}^X)$, where $G_L$ and $G_V$ are scalar gains, and $a_L$ is a low-activation threshold ($a_L$ is 0.05 in this work). This final term has been included as a consequence of the generalized positions we have chosen. When the CE is nearly deactivated (as $a$ approaches $a_L$), this term forces $\tilde{\ell}^S$ and $\tilde{v}^S$ to shadow the location and velocity of the XE attachment point. This ensures that if the XE is suddenly activated, that it attaches with little strain. We had to include this term because we made $\ell^S$ a state variable, rather than $\ell^X$. We chose $\ell^S$ as a state variable, rather than $\ell^X$, so that the states are more equally scaled for numerical integration.

The passive force developed by the CE in *Equation 15* is the sum of the elastic forces (*Figure 3A*) developed by the force-length curves of titin ($\mathbf{f}^1(\tilde{\ell}^1)$ and $\mathbf{f}^2(\tilde{\ell}^2)$) and the ECM ($\mathbf{f}^{ECM}(\tilde{\ell}^{ECM})$). We model titin's elasticity as being due to two serially connected elastic segments: the first elastic segment $\mathbf{f}^1(\tilde{\ell}^1)$ is formed by lumping together the IgP segment and a fraction Q of the PEVK segment, while the second elastic segment $\mathbf{f}^2(\tilde{\ell}^2)$ is formed by lumping together the remaining $(1-Q)$ of the PEVK segment with the free IgD section. Our preliminary simulations of the active lengthening experiment of *Herzog and Leonard, 2002* indicate that a Q value of 0.5, positioning the PEVK-actin attachment point that is near the middle of the PEVK segment, allows the model to develop sufficient tension when actively lengthened. The large section of the IgD segment that is bound to myosin is treated as rigid.

The curves that form $\mathbf{f}^{ECM}(\tilde{\ell}^{ECM})$, $\mathbf{f}^1(\tilde{\ell}^1)$, and $\mathbf{f}^2(\tilde{\ell}^2)$ have been carefully constructed to satisfy three experimental observations: that the total passive force-length curve of titin and the ECM match the observed passive force-length curve of the muscle (*Figure 2A* and *Figure 3A*) as in the experiments of *Prado et al., 2005*; that the proportion of the passive force developed by titin and the ECM (*Figure 3A*) is within experimental observations of *Prado et al., 2005*; and that the Ig domains and PEVK residues show the same relative elongation (*Figure 3C*) as observed by *Trombitás et al., 1998a*. Even though the measurements of *Trombitás et al., 1998b* come from human soleus titin, we can construct the force-length curves of other titin isoforms if the number of proximal Ig domains, PEVK residues, and distal Ig domains are known (see Appendix 2.4). Since the passive–force-length relation and the results of *Trombitás et al., 1998b* are at modest lengths, we consider two different extensions to the force-length relation to simulate extreme lengths: first, a simple linear extrapolation; second, we extend the force-length relation of each of titin's segments to follow a worm-like-chain

(WLC) model similar to *Trombitás et al., 1998b* (see Appendix 2.4 for details on the WLC model). With titin's passive force-length relations defined, we can next consider what happens to titin in active muscle.

When active muscle is lengthened, it produces an enhanced force that persists long after the lengthening has ceased called residual force enhancement (RFE) (*Herzog and Leonard, 2002*). For the purposes of the VEXAT model, we assume that RFE is produced by titin. Experiments have demonstrated RFE on both the ascending limb (*Peterson et al., 2004*) and descending limb of the force-length (*Herzog and Leonard, 2002*) relation. The amount of RFE depends both on the final length of the stretch (*Hisey et al., 2009*) and the magnitude of the stretch: on the ascending limb the amount of RFE varies with the final length but not with stretch magnitude, while on the descending limb RFE varies with stretch magnitude.

To develop RFE, we assume that some point of the PEVK segment bonds with actin through an activation-dependent damper. The VEXAT model's distal segment of titin, $\ell^2$, can contribute to RFE when the titin-actin bond is formed and CE is lengthened beyond $\tilde{\ell}_s^{\mathrm{M}}$, the shortest CE length at which the PEVK-actin bond can form. In this work, we set $\tilde{\ell}_s^{\mathrm{M}}$ to be equal to the slack length of the CE's force-length relation $\tilde{\ell}_s^{\mathrm{PE}}$ (see *Table 1E and H*). To incorporate the asymmetric length dependence of RFE observed by *Hisey et al., 2009*, we introduce a smooth step-up function

$$u^{\mathrm{S}} = \frac{1}{2} + \frac{1}{2} \tanh \left( \frac{\tilde{\ell}^{\mathrm{M}} - \tilde{\ell}_s^{\mathrm{M}}}{\mathrm{R}} \right) \tag{17}$$

that transitions from zero to one as $\tilde{\ell}^{\mathrm{M}}$ extends beyond $\tilde{\ell}_s^{\mathrm{M}}$, where the sharpness of the transition is controlled by R. Similar to the experimental work of *Hisey et al., 2009*, active lengthening on the ascending limb will produce similar amounts of RFE since $\tilde{\ell}_s^{\mathrm{M}} < \ell_o^{\mathrm{M}}$ and the titin-actin bond is prevented from forming below $\tilde{\ell}_s^{\mathrm{M}}$. In contrast, the amount of RFE on the descending limb increases with increasing stretch magnitudes since the titin-actin bond is able to form across the entire descending limb.

At very long CE lengths, the modeled titin-actin bond can literally slip off of the end of the actin filament (*Figure 1A*) when the distance between the Z-line and the bond, $\tilde{\ell}^1 + \tilde{\mathrm{L}}^{\mathrm{T12}}$, exceeds the length of the actin filament, $\tilde{\mathrm{L}}^{\mathrm{A}}$. To break the titin-actin bond at long CE lengths, we introduce a smooth step-down function

$$u^{\mathrm{L}} = \frac{1}{2} - \frac{1}{2} \tanh \left( \frac{(\tilde{\ell}^1 + \tilde{\mathrm{L}}^{\mathrm{T12}}) - \tilde{\mathrm{L}}^{\mathrm{A}}}{\mathrm{R}} \right) \tag{18}$$

The step-down function $u^{\mathrm{L}}$ transitions from one to zero when the titin-actin bond $(\tilde{\ell}^1 + \tilde{\mathrm{L}}^{\mathrm{T12}})$ reaches $\tilde{\mathrm{L}}^{\mathrm{A}}$, the end of the actin filament.

The strength of the titin-actin bond also appears to vary nonlinearly with activation. *Fukutani and Herzog, 2018* observed that the absolute RFE magnitude produced by actively lengthened fibers is similar between normal and reduced contractile force states. Since the experiments of *Fukutani and Herzog, 2018* were performed beyond the optimal CE length, titin could be contributing to the observed RFE as previously described. The consistent pattern of absolute RFE values observed by *Fukutani and Herzog, 2018* could be produced if the titin-actin bond saturated at its maximum strength even at a reduced contractile force state. To saturate the titin-actin bond, we use a final smooth step function

$$u^{\mathrm{A}} = 1 - e^{-\left( \frac{a}{\mathrm{A}_\circ} \right)^2} \tag{19}$$

where $\mathrm{A}_\circ$ is the threshold activation level at which the bond saturates. While we model the strength of the titin-actin bond as being a function of activation, which is proportional $\mathrm{Ca}^{2+}$ concentration (*Millar and Homsher, 1990*), this is a mathematical convenience. The work of *Leonard et al., 2010* makes it clear that both $\mathrm{Ca}^{2+}$ and crossbridge cycling are needed to allow titin to develop enhanced forces during active lengthening: no enhanced forces are observed in the presence of $\mathrm{Ca}^{2+}$ when crossbridge cycling is chemically inhibited. Putting this all together, the active damping acting between the titin

**Table 1.** The VEXAT and Hill model's elastic-tendon cat soleus MTU parameters.

The VEXAT model uses all of the Hill model's parameters which are highlighted in grey. Short forms are used to indicate: length 'len', velocity 'vel', acceleration 'acc', half 'h', activation 'act', segment 'seg', threshold 'thr', and stiffness 'stiff'. The letters 'R' or 'H' in front of a reference mean the data is from a rabbit or a human, otherwise the data is from cat soleus. References are in brackets and are coded in order of appearance as: 'H02' for *Herzog and Leonard, 2002*, 'S82' for *Sacks and Roy, 1982*, 'S96' for *Scott et al., 1996*, 'H38' for *Hill, 1938*, 'S95' for *Scott and Loeb, 1995*, 'N96' for *Netti et al., 1996*, 'R99' for *Rassier et al., 1999*, 'K94' for *Kirsch et al., 1994*, 'P05' for *Prado et al., 2005*, and 'T98' for *Trombitás et al., 1998b*. The letters following a reference indicate how the data was used to create the parameter: 'C' calculated, 'F' fit, 'E' estimated, or 'S' scaled. Most of the VEXAT model's XE and titin parameters can be used as rough parameter guesses for other muscles because we have expressed these parameters in a normalized space: the values will scale appropriately with changes to $\ell_o^M$ and $f_o^M$. Titin's force-length curves, however, should be updated if $N^{IgP}$, $N^{PEVK}$, or $N^{IgD}$ differ from the values shown below (see Appendix 2 for details). Note that the rigid-tendon cat soleus parameters differ from the table below because tendon elasticity affects the fitting of $k_o^X$, $\beta_o^X$, $\mathbf{f}^{PE}$, $\mathbf{f}^1(\tilde{\ell}^1)$, and $\mathbf{f}^2(\tilde{\ell}^2)$. Finally, the parameters related to the compressive element (F), the XE (G), and titin (H and I) can be used as initial values when simulating the MTU's other mammals. By making use of these defaults, the VEXAT model requires the same number of parameters as a Hill-type muscle model (A—E).

| Parameter | | Value | Source |
|---|---|---|---|
| | | **A. Basic parameters** | |
| Max iso force | $f_o^M$ | 21.5 N | (H02)F |
| Opt CE len | $\ell_o^M$ | 42.9 mm | (H02)F |
| Pen angle | $\alpha$ | 7.00° | (S82) |
| Act time const | $\tau_A$ | 113 ms | (H02)F |
| De-act time const | $\tau_D$ | 142 ms | (H02)F |
| | | **B. Force-velocity relation: $\mathbf{f}^V(\hat{v}^M)$** | |
| Max shortening vel | $v_{max}^M$ | $4.65\ \frac{\ell_o^M}{s}$ | (S96)F |
| $\mathbf{f}^V$ at $-\frac{1}{2}v_{max}^M$ | $\tilde{f}_1^V$ | $0.126 f_o^M$ | (S96)F |
| $\mathbf{f}^V$ at $\hat{v}^M = +0$ | $\tilde{f}_2^V$ | $1.40 f_o^M$ | (H02)F |
| $\mathbf{f}^V$ at $v_{max}^M$ | $\tilde{f}_3^V$ | $1.55 f_o^M$ | (H02)E |
| $v_{max}^M$ scaling | $s^V$ | 0.950 | (H38)F |
| | | **C. Tendon model: $\mathbf{f}^T(\tilde{\ell}^T)$, $U\hat{\mathbf{k}}^T(\tilde{\ell}^T)$** | |
| Slack len | $\ell_s^T$ | 30.5 mm | (S95)S |
| Stiffness | $\tilde{k}_o^T$ | $30.0\ \frac{f_o^M}{\ell_s^T}$ | (S95) |
| Strain at $f_o^M$ | $e_o^T$ | 0.0458 | (S95) |
| Toe force | $f_{toe}^T$ | $\frac{2}{3}f_o^M$ | (S95)E |
| Damping | $U$ | $0.057\ s$ | R(N96)F |
| | | **D. Active force-length relation: $\mathbf{f}^L(\tilde{\ell}^M)$** | |
| Opt sarcomere len | $L_o^M$ | 2.43 $\mu$m | (R99) |
| Actin len | $\tilde{L}^A$ | $0.462\ \ell_o^M$ | (R99) |

*Table 1 continued on next page*

*Table 1 continued*

| Parameter | | Value | Source |
|---|---|---|---|
| Myosin h-len | $\tilde{L}^M$ | $0.330\,\ell_o^M$ | (R99) |
| Myosin bare h-len | $\tilde{L}^B$ | $0.0175\,\ell_o^M$ | (R99) |
| Offset | $\Delta^L$ | $-\frac{2}{\tilde{k}_o^X}\ell_o^M$ | C |
| **E. Passive force-length relation: $\mathbf{f}^{PE}(\tilde{\ell}^M)$** | | | |
| Slack len | $\tilde{\ell}_s^{PE}$ | $0.872\,\ell_o^M$ | (H02)F |
| Toe len | $\tilde{\ell}_{toe}^{PE}$ | $1.39\,\ell_o^M$ | (H02)F |
| Toe force | $\tilde{f}_{toe}^{PE}$ | $1.00\,f_o^M$ | (H02)F |
| Toe stiffness | $\tilde{k}_{toe}^{PE}$ | $3.88\,\frac{f_o^M}{\ell_o^M}$ | (H02)F |
| **F. Compressive force-length relation: $\mathbf{f}^{KE}(\tilde{\ell}^M)$** | | | |
| Slack len | $\tilde{\ell}_s^{PE}$ | $\frac{1}{10}\,\ell_o^M$ | E |
| Toe len | $\tilde{\ell}_{toe}^{PE}$ | $0.00\,\ell_o^M$ | E |
| Toe force | $\tilde{f}_{toe}^{PE}$ | $1.00\,f_o^M$ | E |
| **G. XE viscoelastic model** | | | |
| Stiffness | $\tilde{k}_o^X$ | $49.1\,\frac{f_o^M}{\ell_o^M}$ | (K94)F:*Figure 12* |
| Damping | $\tilde{\beta}_o^X$ | $0.347\,\frac{f_o^M}{\ell_o^M/s}$ | (K94)F:*Figure 12* |
| Acc. time const | $\tau^S$ | 1.00e-3 s | (K94,H38)E |
| Num acc damping | D | 1.00 | (K94,H39)E |
| Low act threshold | $a_L$ | 0.0500 | (K94,H39)E |
| Len tracking gain | $G_L$ | $1000\,\frac{1}{s}$ | (K94,H39)E |
| Vel tracking gain | $G_V$ | 1000 | (K94,H39)E |
| **H. Titin & ECM Parameters** | | | |
| ECM fraction | P | 56% | R(P05) |
| PEVK attach pt | Q | 0.625 | (H02)F |
| Z-line–T12 len | $\tilde{L}^{T12}$ | $0.0412\,\ell_o^M$ | H(T98) |
| IgD rigid h-len | $\tilde{L}^{IgD}$ | $\tilde{L}^M$ | (R99) |
| No IgP domains | $N^{IgP}$ | 60.5 | H(T98)S |
| No PEVK residues | $N^{PEVK}$ | 1934.7 | H(T98)S |
| No IgD domains | $N^{IgD}$ | 19.5 | H(T98)S |
| Active damping | $\beta_A^{PEVK}$ | $71.9\,\frac{f_o^M}{\ell_o^M}$ | (H02)F |
| Passive damping | $\beta_P^{PEVK}$ | $0.1\,\frac{f_o^M}{\ell_o^M}$ | E |
| Length threshold | $\tilde{\ell}_s^M$ | $\frac{1}{2}\,\tilde{\ell}_s^{PE}$ | E |
| Act threshold | $A_o$ | 0.05 | E |

*Table 1 continued on next page*

*Table 1 continued*

| Parameter | | Value | Source |
|---|---|---|---|
| Step transition | R | 0.01 | E |
| | | | |
| I. Titin's force-length relations: $\mathbf{f}^1(\tilde{\ell}^1)$ & $\mathbf{f}^2(\tilde{\ell}^2)$ | | |
| $\mathbf{f}^1(\tilde{\ell}^1)$ slack len | $\tilde{\ell}_S^1$ | $0.0739\,\ell_o^M$ | H(T98)S, (H02)F |
| $\mathbf{f}^1(\tilde{\ell}^1)$ toe len | $\tilde{\ell}_{toe}^1$ | $0.1590\,\ell_o^M$ | H(T98)S, (H02)F |
| $\mathbf{f}^1(\tilde{\ell}^1)$ toe force | $\tilde{f}_{toe}^1$ | $(1-P)f_o^M$ | H(T98)S, (H02)F |
| $\mathbf{f}^1(\tilde{\ell}^1)$ toe stiff | $\tilde{k}_{toe}^1$ | $5.17\,\dfrac{f_o^M}{\ell_o^M}$ | H(T98)S, (H02)F |
| $\mathbf{f}^2(\tilde{\ell}^2)$ slack len | $\tilde{\ell}_S^2$ | $0.0454\,\ell_o^M$ | H(T98)S, (H02)F |
| $\mathbf{f}^2(\tilde{\ell}^2)$ toe len | $\tilde{\ell}_{toe}^2$ | $0.0977\,\ell_o^M$ | H(T98)S, (H02)F |
| $\mathbf{f}^2(\tilde{\ell}^2)$ toe force | $\tilde{f}_{toe}^2$ | $(1-P)f_o^M$ | H(T98)S, (H02)F |
| $\mathbf{f}^2(\tilde{\ell}^2)$ toe stiff | $\tilde{k}_{toe}^2$ | $8.42\,\dfrac{f_o^M}{\ell_o^M}$ | H(T98)S, (H02)F |

and actin filaments is given by the product of $u^A u^S u^L \beta_A^{PEVK}$, where $\beta_A^{PEVK}$ is the maximum damping coefficient.

With a model of the titin-actin bond derived, we can focus on how the bond location moves in response to applied forces. Since we are ignoring the mass of the titin filament, the PEVK-attachment point is balanced by the forces applied to it and the viscous forces developed between titin and actin

$$\left(u^A\,u^S\,u^L\,\beta_A^{PEVK} + \beta_P^{PEVK}\right)\tilde{v}^1 = \mathbf{f}^1(\tilde{\ell}^1) - \mathbf{f}^2(\tilde{\ell}^2) \tag{20}$$

due to the active ($u^A\,u^S\,u^L\,\beta_A^{PEVK}$) and a small amount of passive damping ($\beta_P^{PEVK}$). Since *Equation 20* is linear in $\tilde{v}^1$, we can solve directly for it

$$\tilde{v}^1 = \frac{\mathbf{f}^2(\tilde{\ell}^2) - \mathbf{f}^1(\tilde{\ell}^1)}{u^A\,u^S\,u^L\,\beta_A^{PEVK} + \beta_P^{PEVK}}. \tag{21}$$

The assumption of whether the tendon is rigid or elastic affects how the state derivative is evaluated and how expensive it is to compute. While all of the position dependent quantities can be evaluated using *Equations 6–11* and the generalized positions, evaluating the generalized velocities of a rigid-tendon and elastic-tendon model differ substantially. The CE velocity $v^M$ and pennation angular velocity $\dot{\alpha}$ of a rigid-tendon model can be evaluated directly given the path length, velocity, and the time derivatives of *Equation 6* and *Equation 8*. After $v^1$ is evaluated using *Equation 21*, the velocities of the remaining segments can be evaluated using the time derivatives of *Equations 9–11*.

Evaluating the CE rate of lengthening, $v^M$, for an elastic-tendon muscle model is more involved. As is typical of lumped parameter muscle models (*Zajac, 1989*; *Thelen, 2003*; *Millard et al., 2013*), here we assume that difference in tension, $\tilde{f}^\epsilon$, between the CE and the tendon

$$\tilde{f}^\epsilon = \tilde{f}^M \cos\alpha - f^T \approx 0 \tag{22}$$

is negligible (see Appendix 9, Note 10). During our preliminary simulations, it became clear that treating the tendon as an idealized spring degraded the ability of the model to replicate the experiment of *Kirsch et al., 1994* particularly at high frequencies. *Kirsch et al., 1994* observed a linear increase in the gain and phase profile between the output force and the input perturbation applied to the muscle. This pattern in gain and phase shift can be accurately reproduced by a spring in parallel with a damper. Due to the way that impedance combines in series (see Appendix 9, Note 11), the

models of both the CE and the tendon need to have parallel spring and damper elements so that the entire MTU, when linearized, appears to be a spring in parallel with a damping element. We model tendon force using a nonlinear spring and damper model

$$f^{\mathrm{T}} = \mathbf{f}^{\mathrm{T}}(\tilde{\ell}^{\mathrm{T}}) + U\hat{\boldsymbol{k}}^{\mathrm{T}}(\tilde{\ell}^{\mathrm{T}})\tilde{v}^{\mathrm{T}} \tag{23}$$

where the damping coefficient $U\hat{\mathbf{k}}^{\mathrm{T}}(\tilde{\ell}^{\mathrm{T}})$, is a linear scaling of the normalized tendon stiffness $\hat{\boldsymbol{k}}^{\mathrm{T}}$ by $U$, a constant scaling coefficient. We have chosen this specific damping model because it fits the data of *Netti et al., 1996* and captures the structural coupling between tendon stiffness and damping (see Appendix 2.2 and *Figure 1* for further details).

Now that all the terms in *Equation 22* have been explicitly defined, we can use *Equation 22* to solve for $v^{\mathrm{M}}$. *Equation 22* becomes linear in $v^{\mathrm{M}}$ after substituting the force models described in *Equation 23* and *Equation 15*, and the kinematic model described in *Equation 8*, *Equation 9* and *Equation 11* (along with the time derivatives of *Equations 8–11*). After some simplification we arrive at

$$\tilde{v}^{\mathrm{M}} = \frac{\left(a\mathbf{f}^{\mathrm{L}}(\tilde{\ell}^{\mathrm{S}} + \tilde{\mathrm{L}}^{\mathrm{M}})(\tilde{k}_o^{\mathrm{X}}\tilde{\ell}^{\mathrm{X}} + \tilde{\beta}_o^{\mathrm{X}}\tilde{v}^{\mathrm{S}}) + \mathbf{f}^2(\tilde{\ell}^2) + \mathbf{f}^{\mathrm{ECM}}(\tilde{\ell}^{\mathrm{ECM}})\right)\cos\alpha - \mathbf{f}^{\mathrm{KE}}(\tilde{\ell}^{\mathrm{M}}) - \mathbf{f}^{\mathrm{T}}(\tilde{\ell}^{\mathrm{T}}) - \dfrac{U\hat{\mathbf{k}}^{\mathrm{T}}(\tilde{\ell}^{\mathrm{T}})v^{\mathrm{P}}}{\ell_s^{\mathrm{T}}}}{-\dfrac{a\tilde{\beta}_0^{\mathrm{X}}\mathbf{f}^{\mathrm{L}}(\tilde{\ell}^{\mathrm{S}} + \tilde{\mathrm{L}}^{\mathrm{M}})}{2\tilde{\ell}^{\mathrm{M}}} - \dfrac{(\tilde{\beta}^{\epsilon} + \mathbf{f}^{\mathrm{ECM}}(\tilde{\ell}^{\mathrm{ECM}}))\cos\alpha}{2\tilde{\ell}^{\mathrm{M}}} - \dfrac{U\hat{\mathbf{k}}^{\mathrm{T}}(\tilde{\ell}^{\mathrm{T}})}{\ell_s^{\mathrm{T}}\cos\alpha}} \tag{24}$$

allowing us to evaluate the final state derivative in $\underline{\dot{x}}$. During simulation the denominator of $\tilde{v}^{\mathrm{M}}$ will always be finite since $\tilde{\beta}^{\epsilon} > 0$, and $\alpha < 90°$ due to the compressive element. The evaluation of $\underline{\dot{x}}$ in the VEXAT model is free of numerical singularities, giving it an advantage over a conventional Hill-type muscle models (*Millard et al., 2013*). In addition, the VEXAT's $\underline{\dot{x}}$ does not require iteration to numerically solve a root, giving it an advantage over a singularity-free formulation of the Hill model (*Millard et al., 2013*). As with previous models, initializing the model's state is not trivial and required the derivation of a model-specific method (see Appendix 3 for details).

## Biological benchmark simulations

In order to evaluate the model, we have selected three experiments that capture the responses of active muscle to small, medium, and large length changes. The small (1–3.8% $\ell_o^{\mathrm{M}}$) stochastic perturbation experiment of *Kirsch et al., 1994* demonstrates that the impedance of muscle is well described by a stiff spring in parallel with a damper, and that the spring-damper coefficients vary linearly with active force. The active impedance of muscle is such a fundamental part of motor learning that the amount of impedance, as indicated by co-contraction, is used to define how much learning has actually taken place (*Franklin et al., 2003*; *Franklin et al., 2008*): co-contraction is high during initial learning, but decreases over time as a task becomes familiar. The active lengthening experiment of *Herzog and Leonard, 2002* shows that modestly stretched (7–21% $\ell_o^{\mathrm{M}}$) biological muscle has positive stiffness even on the descending limb of the active force-length curve ($\ell_o^{\mathrm{M}} > 1$). In contrast, a conventional Hill model (*Zajac, 1989*; *Millard et al., 2013*) can have negative stiffness on the descending limb of the active-force-length curve, a property that is both mechanically unstable and unrealistic. The final active lengthening experiment of *Leonard et al., 2010* unequivocally demonstrates that the CE continues to develop active forces during extreme lengthening (329% $\ell_o^{\mathrm{M}}$) which exceeds actin-myosin overlap. Active force development beyond actin-myosin overlap is made possible by titin, and its activation-dependent interaction with actin (*Leonard et al., 2010*). The biological benchmark simulations conclude with a replication of the force-velocity experiments of *Hill, 1938* and the force-length experiments of *Gordon et al., 1966*.

The VEXAT model requires the architectural muscle parameters ($f_o^{\mathrm{M}}$, $\ell_o^{\mathrm{M}}$, $\alpha_o$, $v_{\max}^{\mathrm{M}}$, and $\ell_s^{\mathrm{T}}$) needed by a conventional Hill-type muscle model as well as additional parameters. The additional parameters are needed for these component models: the compressive element (*Equation 15* and *Equation 24*), the lumped viscoelastic XE (*Equation 1* and *Equation 2*), XE-actin dynamics (*Equation 16*), the two-segment active titin model (*Figure 3*), titin-actin dynamics (*Equation 21*), and the tendon damping model (*Equation 23*). Fortunately, there is enough experimental data in the literature that values can be found, fitted, or estimated directly for our simulations of experiments on cat soleus (*Table 1F.-I.*),

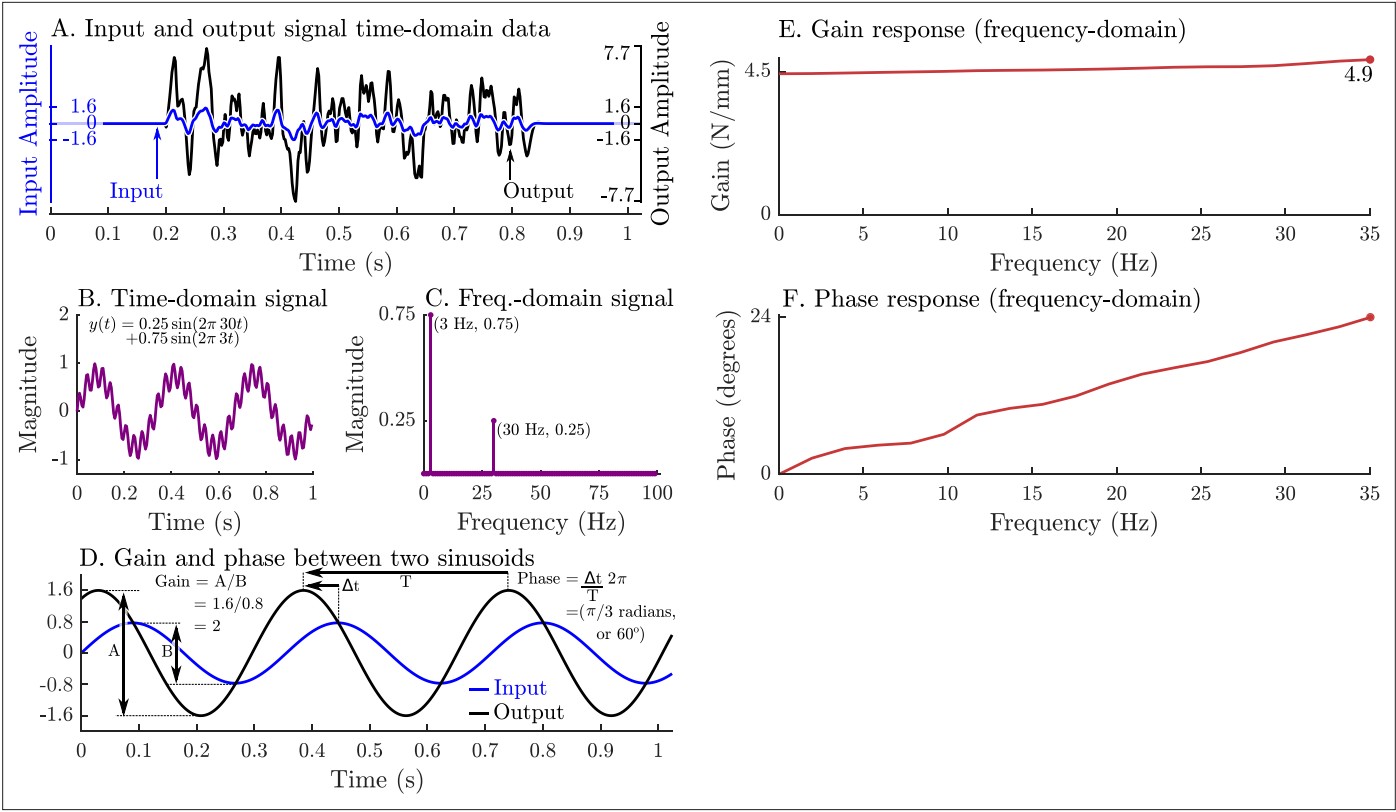

**Figure 4.** A graphical overview of a system's response in the time-domain, frequency-domain, its gain-response, and its phase-response. Evaluating a system's gain and phase response begins by applying a pseudo-random input signal to the system and measuring its output (**A**). Both the input and output signals (**A**) are transformed into the frequency domain by expressing these signals as an equivalent sum of scaled and shifted sinusoids (simple example shown in B and C). Each individual input sinusoid is compared with the output sinusoid of the same frequency to evaluate how the system scales and shifts the input to the output (**D**). This process is repeated across all sinusoid pairs to produce a function that describes how an input sinusoid is scaled (**E**) and shifted (**F**) to an output sinusoid using only the measured data (**A**).

and rabbit psoas fibrils (see Appendix 2 and Appendix 8 for rabbit psoas fibril model parameters). The parameter values we have established for the cat soleus (*Table 1F.-I.*) can serve as initial values when modeling other mammalian MTU's because these parameters have been normalized (by $f_o^M$, $\ell_o^M$, and $\ell_s^T$ where appropriate) and will scale appropriately given the architectural properties of a different MTU. By making use of these default values, the VEXAT model can be made to represent another MTU using exactly the same number of parameters as a Hill-type muscle model (*Table 1A.-E.*).

## Stochastic length perturbation experiments

In the in-situ experiment of *Kirsch et al., 1994*, the force response of a cat's soleus muscle under constant stimulation was measured as its length was changed by small amounts. *Kirsch et al., 1994* applied stochastic length perturbations (*Figure 4A*, blue line) to elicit force responses from the muscle (in this case a spring-damper *Figure 4A*, black line) across a broad range of frequencies (4–90 Hz) and across a range of small length perturbations (1–3.8% $\ell_o^M$). The resulting time-domain signals can be quite complicated (*Figure 4A*) but contain rich measurements of how muscle transforms changes in length into changes in force.

As long as muscle can be considered to be linear (a sinusoidal change in length produces a sinusoidal change in force), then system identification methods (*Oppenheim et al., 1996*; *Koopmans, 1995*) can be applied to extract a relationship between length $x(t)$ and force $y(t)$. We will give a brief overview of system identification methods here to make methods and results clearer. First, the time-domain signals ($x(t)$ and $y(t)$) are transformed into an equivalent representation in the frequency-domain ($X(s)$ and $Y(s)$) as a sum of scaled and shifted sine curves (*Figure 4B and C*) using a Fourier transform (*Oppenheim et al., 1996*). In the frequency domain, we identify an LTI system of best fit $H(s)$

that describes how muscle transforms changes in length into changes in force such that $Y(s) = H(s)X(s)$. Next, we evaluate how $H(s)$ scales the magnitude (gain) and shifts the timing (phase) of a sinusoid in $X(s)$ into a sinusoid of the same frequency in $Y(s)$ (***Figure 4D***). This process is repeated across all frequency-matched pairs of input and output sinusoids to build a function of how muscle scales (***Figure 4E***) and shifts (***Figure 4F***) input length sinusoids into output force sinusoids. The resulting transformation turns two complicated time-domain signals (***Figure 4A***) into a clear relationship in the frequency-domain that describes how muscle transforms length changes into force changes: a very slow (≈0 Hz) length change will result in an output force that is scaled by 4.5 and is in phase (***Figure 4E and F***), a 35 Hz sinusoidal length change will produce an output force that is scaled by 4.9 and leads the input signal by 24° (***Figure 4E and F***), and frequencies between 0 Hz and 35 Hz will be between these two signals in terms of scaling and phase. These patterns of gain and phase can be used to identify a network of spring-dampers that is equivalent to the underlying linear system (the system in ***Figure 4A, E and F*** is a 4.46 N/mm spring in parallel with a 0.0089 Ns/mm damper). Since experimental measurements often contain noise and small nonlinearities, the mathematical procedure used to estimate $H(s)$ and the corresponding gain and phase profiles is more elaborate than we have described (see Appendix 4 for details).

***Kirsch et al., 1994*** used system identification methods to identify LTI mechanical systems that best describes how muscle transforms input length waveforms to output force waveforms. The resulting LTI system, however, is only accurate when the relationship between input and output is approximately linear. ***Kirsch et al., 1994*** used the coherence squared, $(C_{xy})^2$, between the input and output to evaluate the degree of linearity: $Y(s)$ is a linear transformation of $X(s)$ at frequencies in which $(C_{xy})^2$ is near one, while $Y(s)$ cannot be described as a linear function of $X(s)$ at frequencies in which $(C_{xy})^2$ approaches zero. By calculating $(C_{xy})^2$ between the length perturbation and force waveforms, ***Kirsch et al., 1994*** identified the bandwidth in which the muscle's response is approximately linear. ***Kirsch et al., 1994*** set the lower frequency of this band to 4 Hz, and Figure 3 of ***Kirsch et al., 1994*** suggests that this corresponds to $(C_{xy})^2 \geq 0.67$ though the threshold for $(C_{xy})^2$ is not reported. The upper frequency of this band was set to the cutoff frequency of the low-pass filter applied to the input (15, 35, or 90 Hz). Within this bandwidth, ***Kirsch et al., 1994*** compared the response of the specimen to several candidate models and found that a parallel spring-damper fit the muscle's frequency response best. Next, ***Kirsch et al., 1994*** evaluated the stiffness and damping coefficients that best fit the muscle's frequency response. Finally, ***Kirsch et al., 1994*** evaluated how much of the muscle's time-domain response was captured by the spring-damper of best fit by evaluating the variance-accounted-for (VAF) between the two time-domain signals

$$VAF(f^{KD}, f^{EXP}) = \frac{\sigma^2(f^{EXP}) - \sigma^2(f^{KD} - f^{EXP})}{\sigma^2(f^{EXP})}. \tag{25}$$

Astonishingly, ***Kirsch et al., 1994*** found that a spring-damper of best fit has a VAF of between 78% and 99% (see Appendix 9, Note 12) when compared to the experimentally measured forces $f^{EXP}$. By repeating this experiment over a variety of stimulation levels (using both electrical stimulation and the crossed-extension reflex) ***Kirsch et al., 1994*** showed that these stiffness and damping coefficients vary linearly with the active force developed by the muscle. Furthermore, ***Kirsch et al., 1994*** repeated the experiment using perturbations that had a variety of lengths (0.4 mm, 0.8 mm, and 1.6 mm) and bandwidths (15 Hz, 35 Hz, and 90 Hz) and observed a peculiar quality of muscle: the damping coefficient of best fit increases as the bandwidth of the perturbation decreases (see Figures 3 and 10 of ***Kirsch et al., 1994*** for details). Here, we simulate the experiment of ***Kirsch et al., 1994*** to determine, first, the VAF of the VEXAT model and the Hill model in comparison to a spring-damper of best fit; second, to compare the gain and phase response of the models to biological muscle; and finally, to see if the spring-damper coefficients of best fit for both models increase with active force in a manner that is similar to the cat soleus that ***Kirsch et al., 1994*** studied.

To simulate the experiments of ***Kirsch et al., 1994*** we begin by creating the 9 stochastic perturbation waveforms used in the experiment that vary in perturbation amplitude (0.4 mm, 0.8mm, and 1.6 mm) and bandwidth (0–15 Hz, 0–35 Hz, and 0–90 Hz) (see Appendix 9, Note 13). The waveform is created using a vector that is composed of random numbers with a range of [-1,1] that begins and ends with a series of zero-valued padding points. Next, a forward pass of a second-order Butterworth filter is applied to the waveform and finally the signal is scaled to the appropriate amplitude (***Figure 5***). The

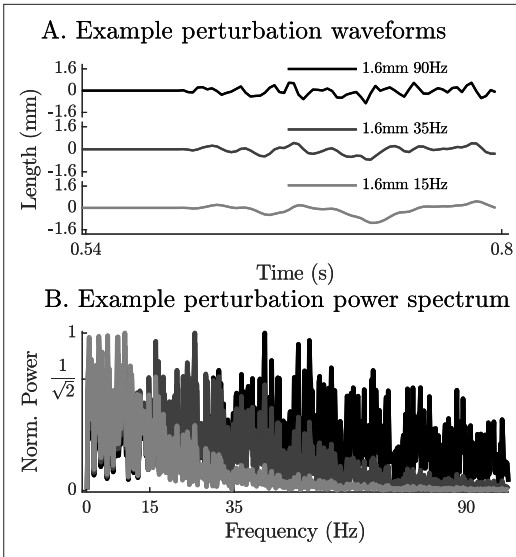

**Figure 5.** The time-domain and power-spectrum of the bandwidth-limited stochastic perturbation signals. The perturbation waveforms are constructed by generating a series of pseudo-random numbers, padding the ends with zeros, by filtering the signal using a 2nd order low-pass filter (wave forms with −3 dB cut-off frequencies of 90 Hz, 35 Hz, and 15 Hz appear in **A**) and finally by scaling the range to the desired limit (1.6 mm in **A**). Although the power spectrum of the resulting signals is highly variable, the filter ensures that the frequencies beyond the −3 dB point have less than half their original power (**B**).

muscle model is then activated until it develops a constant tension at a length of $\ell_o^M$. The musculo-tendon unit is then simulated as the length is varied using the previously constructed waveforms while activation is held constant. To see how impedance varies with active force, we repeated these simulations at ten evenly spaced tensions from 2.5N to 11.5N. Ninety simulations are required to evaluate the nine different perturbation waveforms at each of the ten tension levels. The time-domain length perturbations and force responses of the modeled muscles are used to evaluate the coherence squared of the signal, gain response, and phase responses of the models in the frequency-domain. Since the response of the models might be more nonlinear than biological muscle, we select a bandwidth that meets $(C_{xy})^2 > 0.67$ but otherwise follows the bandwidths analyzed by *Kirsch et al., 1994* (see Appendix 4 for details).

When coupled with an elastic-tendon, the 15 Hz perturbations show that neither model can match the VAF of the analysis of *Kirsch et al., 1994* (compare *Figure 6A to G*), while at 90 Hz the VEXAT model reaches a VAF of 89% (*Figure 6D*) which is within the range of 78–99% reported by *Kirsch et al., 1994*. In contrast, the Hill model's VAF at 90 Hz remains low at 58% (*Figure 6J*). While the VEXAT model has a gain profile in the frequency-domain that closer to the data of *Kirsch et al., 1994* than the Hill model (compare *Figure 6B to H and E to K*), both models have a greater phase shift than the data of *Kirsch et al., 1994* at low frequencies (compare *Figure 6C to I and F to L*). The phase response of the VEXAT model to the 90 Hz perturbation (*Figure 6F*) shows the consequences of *Equation 16*: at low frequencies the phase response of the VEXAT model is similar to that of the Hill model, while at higher frequencies the model's response becomes similar to a spring-damper. This frequency dependent response is a consequence of the first term in *Equation 16*: the value of $\tau^S$ causes the response of the model to be similar to a Hill model at lower frequencies and mimic a spring-damper at higher frequencies. Both models show the same perturbation-dependent phase-response, as the damping coefficient of best fit increases as the perturbation bandwidth decreases: compare the damping coefficient of best fit for the 15 Hz and 90 Hz profiles for the VEXAT model (listed on *Figure 6A. and D.*) and the Hill model (listed on *Figure 6G. and J.*, respectively).

The closeness of each model's response to the spring-damper of best fit changes when a rigid-tendon is used instead of an elastic-tendon. While the VEXAT model's response to the 15 Hz and 90 Hz perturbations improves slightly (compare *Figure 6A–F* to *Figure 1A–F* in Appendix 6), the response of the Hill model to the 15 Hz perturbation changes dramatically with the time-domain VAF rising from 55% to 85% (compare *Figure 6G–L* to *Figure 1G–L* in Appendix 6). Although the Hill model's VAF in response to the 15 Hz perturbation improved, the frequency response contains mixed results: the rigid-tendon Hill model's gain response is better (*Figure 1H* in Appendix 6), while the phase response is worse in comparison to the elastic-tendon Hill model. While the rigid-tendon Hill model produces a better time-domain response to the 15 Hz perturbation than the elastic-tendon Hill model, this improvement has been made with a larger phase shift between force and length than biological muscle (*Kirsch et al., 1994*).

The gain and phase profiles of both models deviate from the spring-damper of best fit due to the presence of nonlinearities, even for small perturbations. Some of the VEXAT model's nonlinearities in

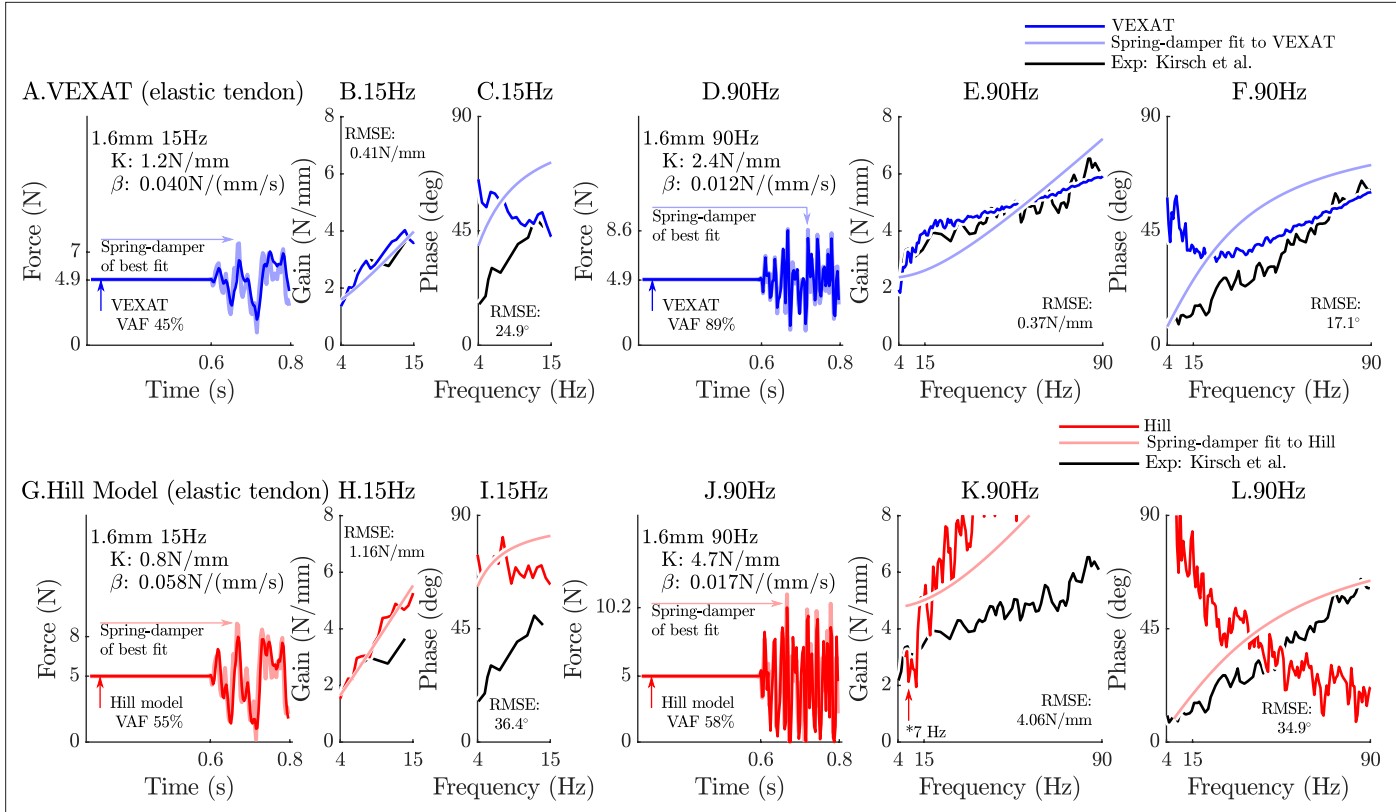

**Figure 6.** The response of the elastic-tendon models (VEXAT and Hill) to the 15Hz and 90Hz perturbations. The 15 Hz perturbations show that the VEXAT model's performance is mixed: in the time-domain (**A**.) the VAF is lower than the 78–99% analyzed by *Kirsch et al., 1994*; the gain response (**B**.) follows the profile in Figure 3 of *Kirsch et al., 1994*, while the phase response differs (**C**.). The response of the VEXAT model to the 90 Hz perturbations is much better: a VAF of 91% is reached in the time-domain (**D**.), the gain response follows the response of the cat soleus analyzed by *Kirsch et al., 1994*, while the phase-response follows biological muscle closely for frequencies higher than 30 Hz. Although the Hill's time-domain response to the 15 Hz signal has a higher VAF than the VEXAT model (**G**.), the RMSE of the Hill model's gain response (**H**.) and phase response (**I**.) shows it to be more in error than the VEXAT model. While the VEXAT model's response improved in response to the 90 Hz perturbation, the Hill model's response does not: the VAF of the time-domain response remains low (**J**.), neither the gain (**K**.) nor phase responses (**L**.) follow the data of *Kirsch et al., 1994*. Note that the Hill model's 90 Hz response was so nonlinear that the lowest frequency analyzed had to be raised from 4 Hz to 7 Hz to satisfy the criteria that $(C_{xy})^2 \geq 0.67$.

this experiment come from the tendon model (compare *Figure 6A–F* to *Figure 1A–F* in Appendix 6), since the response of the VEXAT model with a rigid-tendon stays closer to the spring-damper of best fit. The Hill model's nonlinearities originate from the underlying expressions for stiffness and damping of the Hill model, which are particularly nonlinear with a rigid-tendon model (*Figure 1G–L* in Appendix 6). The stiffness of a Hill model's CE

$$k^{\mathrm{M}} = f_o^{\mathrm{M}} \left( a \frac{\mathrm{d}\mathbf{f}^{\mathrm{L}}}{\mathrm{d}\ell^{\mathrm{M}}} \mathbf{f}^{\mathrm{V}} + \frac{\mathrm{d}\mathbf{f}^{\mathrm{PE}}}{\mathrm{d}\ell^{\mathrm{M}}} \right) \tag{26}$$

is heavily influenced by the partial derivative of $\frac{\mathrm{d}\mathbf{f}^{\mathrm{L}}}{\mathrm{d}\ell^{\mathrm{M}}}$ which has a region of negative stiffness. Although $\frac{\mathrm{d}\mathbf{f}^{\mathrm{PE}}}{\mathrm{d}\ell^{\mathrm{M}}}$ is well approximated as being linear for small length changes, $\frac{\mathrm{d}\mathbf{f}^{\mathrm{L}}}{\mathrm{d}\ell^{\mathrm{M}}}$ changes sign across $\ell_o^{\mathrm{M}}$. The damping of a Hill model's CE

$$\beta^{\mathrm{M}} = f_o^{\mathrm{M}} \left( a \mathbf{f}^{\mathrm{L}} \frac{\mathrm{d}\mathbf{f}^{\mathrm{V}}}{\mathrm{d}v^{\mathrm{M}}} \right) \tag{27}$$

also suffers from high degrees of nonlinearity for small perturbations about $v^{\mathrm{M}} = 0$ since the slope of $\frac{\mathrm{d}\mathbf{f}^{\mathrm{V}}}{\mathrm{d}v^{\mathrm{M}}}$ is positive and large when shortening, and positive and small when lengthening (*Figure 2B*). While

*Equation 26* and *Equation 27* are mathematically correct, the negative stiffness and wide ranging damping values predicted by these equations do not match experimental data (*Kirsch et al., 1994*). In contrast, the stiffness

$$\tilde{k}^{M} = \frac{a\,\tilde{k}_{o}^{X}\,\mathbf{f}^{L}(\tilde{\ell}^{S} + \tilde{L}^{M})}{2} + \frac{d\mathbf{f}^{2}(\tilde{\ell}^{2})}{d\ell^{2}}\frac{1}{2} + \frac{d\mathbf{f}^{ECM}(\tilde{\ell}^{ECM})}{d\ell^{ECM}}\frac{1}{2} \tag{28}$$

and damping

$$\tilde{\beta}^{M} = a\mathbf{f}^{L}(\tilde{\ell}^{S} + \tilde{L}^{M})\left(\tilde{\beta}_{o}^{X}\frac{d\tilde{v}^{X}}{d\tilde{v}^{M}}\right) + \tilde{\beta}^{\epsilon} \tag{29}$$

of the VEXAT's CE do not change so drastically because these terms do not depend on the slope of the force-length relation, or the force-velocity relation (see Appendix 2.5 for derivation).

By repeating the stochastic perturbation experiments across a range of isometric forces, *Kirsch et al., 1994* were able to show that the stiffness and damping of a muscle varies linearly with the active tension it develops (see Figure 12 of *Kirsch et al., 1994*). We have repeated our simulations of the experiments of *Kirsch et al., 1994* at ten nominal forces (spaced evenly between 2.5 N and 11.5 N) and compared how the VEXAT model and the Hill model's stiffness and damping coefficients (*Figure 7*) compare to Figure 12 of *Kirsch et al., 1994*. The stiffness and damping profile of the VEXAT model deviates a little from the data of *Kirsch et al., 1994* because XE's dynamics at 35 Hz are still influenced by the Hill model embedded in *Equation 16* (see Appendix 2.5). Despite this, the VEXAT model develops similar stiffness and damping profile with either a viscoelastic-tendon (*Figure 7A and B*) or a rigid-tendon (*Figure 7C and D*). In contrast, when the Hill model is coupled with an elastic-tendon both its stiffness and damping are larger than what is reported by *Kirsch et al., 1994* at the higher tensions (*Figure 7A and B*). This pattern changes when simulating a Hill model with a rigid-tendon: the model's stiffness is slightly negative (*Figure 7C*), while the model's final damping coefficient is nearly three times the value measured by *Kirsch et al., 1994* (*Figure 7D*). Though a negative stiffness may seem surprising, *Equation 26* shows a negative stiffness is possible at the nominal CE length of these simulations: just past $\ell_{o}^{M}$ the slope of the active force-length curve is negative and the slope of the passive force-length curve is negligible. The tendon model also affects the VAF of the Hill model to a large degree: the elastic-tendon Hill model has a low VAF 30–51% (*Figure 7A & B*) while the rigid-tendon Hill model has a much higher VAF of 86%. Although the VAF of the rigid-tendon Hill model is acceptable, these forces are being generated in a completely different manner than those obtained from biological muscle, as the data of *Kirsch et al., 1994* indicate (*Figure 7C and D*).

When the VAF of the VEXAT and Hill model is evaluated across a range of nominal tensions (ten values from 2.5 to 11.5 N), frequencies (15 Hz, 35 Hz, and 90 Hz), amplitudes (0.4 mm, 0.8 mm, and 1.6 mm), and tendon types (rigid and elastic) two things are clear: first, that the VEXAT model's 64–100% VAF is close to the 78–99% VAF reported by *Kirsch et al., 1994* while the Hill model's 28–95% VAF differs (*Figure 8*); and second, that there are systematic variations in VAF, stiffness, and damping across the different perturbation magnitudes and frequencies (see *Appendix 5—table 1* and *Appendix 5—table 2*). Both models produce worse VAF values when coupled with an elastic-tendon (*Figure 8A, B and C*), although the Hill model is affected most: the mean VAF of the elastic-tendon Hill model is 67% lower than the mean VAF of the rigid-tendon model for the 0.4 mm 15 Hz perturbations (*Figure 8A*). While the VEXAT model's lowest VAF occurs in response to the low frequency perturbations (*Figure 8A*) with both rigid and elastic-tendons, the Hill model's lowest VAF varies with both tendon type and frequency: the rigid-tendon Hill model has its lowest VAF in response to the 1.6 mm 90 Hz perturbations (*Figure 8C*) while the elastic-tendon Hill model has its lowest VAF in response to the 0.4 mm 15 Hz perturbations (*Figure 8A*). It is unclear if biological muscle displays systematic shifts in VAF since *Kirsch et al., 1994* did not report the VAF of each trial.

## Active lengthening on the descending limb

We now turn our attention to the active lengthening in-situ experiments of *Herzog and Leonard, 2002*. During these experiments, cat soleus muscles were actively lengthened by modest amounts (7–21% $\ell_{o}^{M}$) starting on the descending limb of the active-force-length curve ($\ell^{M}/\ell_{o}^{M} > 1$ in *Figure 2A*). This starting point was chosen specifically because the stiffness of a Hill model may actually change sign and become

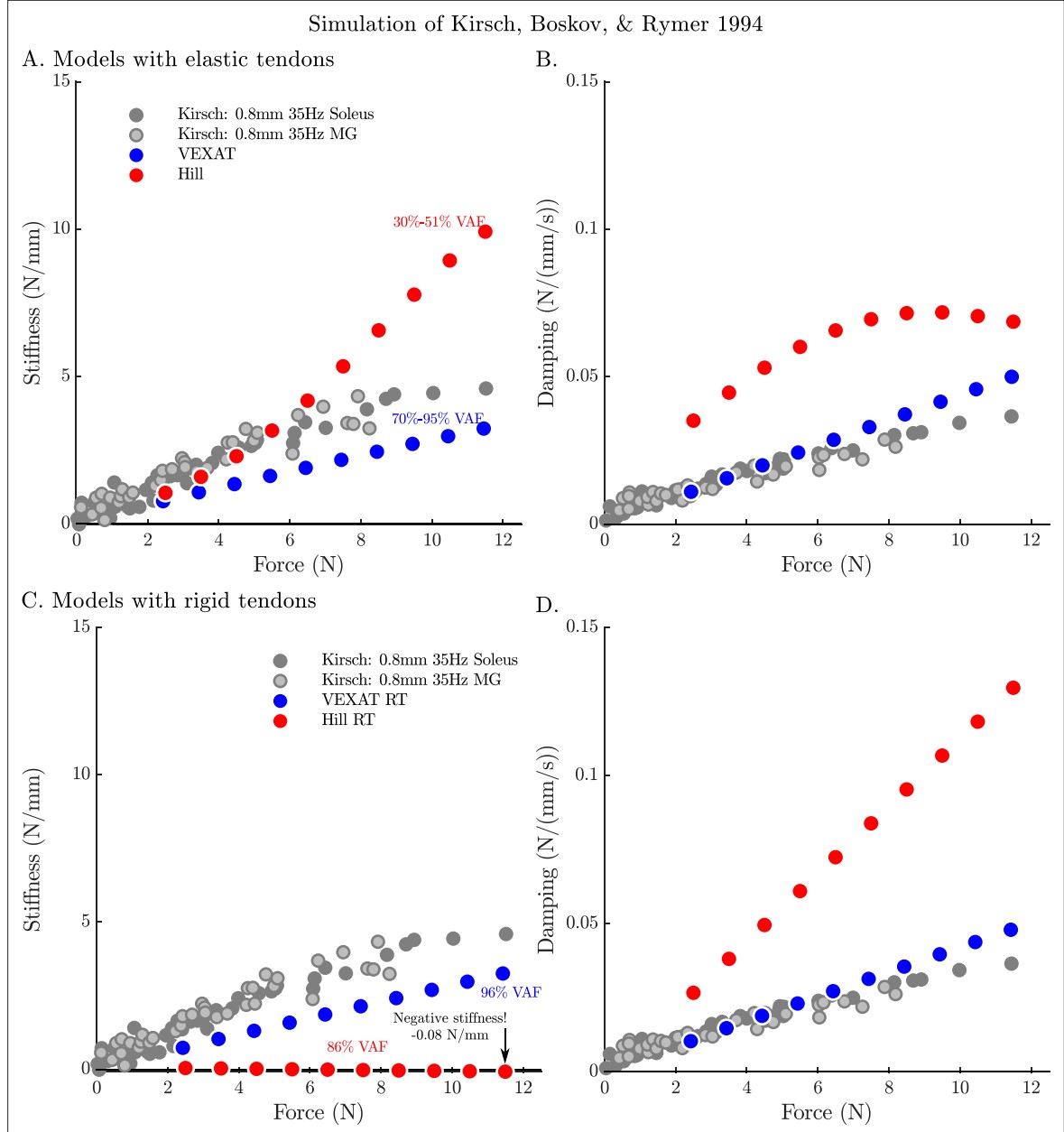

**Figure 7.** The stifffness-force and damping-force (impedance-force) relations of the models when coupled with elastic and rigid tendons. When coupled with an elastic-tendon, the stiffness (**A**) and damping (**B**) coefficients of best fit of both the VEXAT model and a Hill model increase with the tension developed by the MTU. However, both the stiffness and damping of the elastic-tendon Hill model are larger than Kirsch et al.'s coefficients (from Figure 12 of **Kirsch et al., 1994**), particularly at higher tensions. When coupled with rigid-tendon the stiffness (**C**) and damping (**D**) coefficients of the VEXAT model remain similar, as the values for $k_o^X$ and $\beta_o^X$ have been calculated to take the tendon model into account (see Appendix 2.5 for details). In contrast, the stiffness and damping coefficients of the rigid-tendon Hill model differ dramatically from the elastic-tendon Hill model: while the elastic-tendon Hill model is too stiff and damped, the rigid-tendon Hill model is not stiff enough (compare A. to C.) and far too damped (compare B. to D.). Coupling the Hill model with a rigid-tendon increases the VAF from 30–51% to 86% but this improved accuracy is made using stiffness and damping that deviates from that of biological muscle (**Kirsch et al., 1994**).

negative because of the influence of the active-force-length curve on $k^M$ as shown in **Equation 26** as $\ell^M$ extends beyond $\ell_o^M$. The experiment of **Herzog and Leonard, 2002** is important for showing that biological muscle does not exhibit negative stiffness on the descending limb of the active-force-length curve. In addition, this experiment also highlights the slow recovery of the muscle's force after stretching has ceased, and the phenomena of passive force enhancement after stimulation is removed. Here we

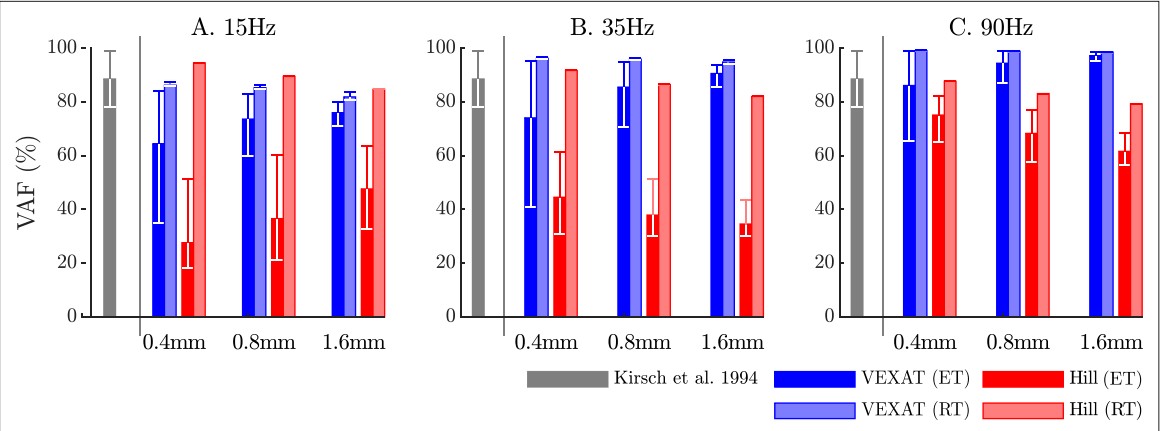

**Figure 8.** The VAF of each model varies systematically with the type of model, perturbation amplitude, frequency, and the nominal tension. *Kirsch et al., 1994* noted that the spring-damper model of best fit has a VAF of between 78–99% across all experiments. We have repeated the perturbation experiments to evaluate the VAF across a range of conditions: two different tendon models, three perturbation bandwidths (15 Hz, 35 Hz, and 90 Hz), three perturbation magnitudes (0.4 mm, 0.8 mm, and 1.6 mm), and ten nominal force values (spaced evenly between 2.5 N and 11.5 N). Each bar in the plot shows the mean VAF across all 10 nominal force values, with the whiskers extending to the minimum and maximum value that occurred in each set. The mean VAF of the VEXAT model changes by up to 36% depending on the condition, with the lowest mean VAF occurring in response to the 0.4mm 15 Hz perturbation with an elastic-tendon (**A**), and the highest mean VAF occurring in response to the 90 Hz perturbations with the rigid-tendon (**C**). In contrast, the mean VAF of the Hill model varies by up to 67% depending on the condition, with the lowest VAF occurring in the 15 Hz 0.4 mm trial with the elastic-tendon (**A**), and the highest value VAF occurring in the 15 Hz 0.4 mm trial with the rigid-tendon (**A**).

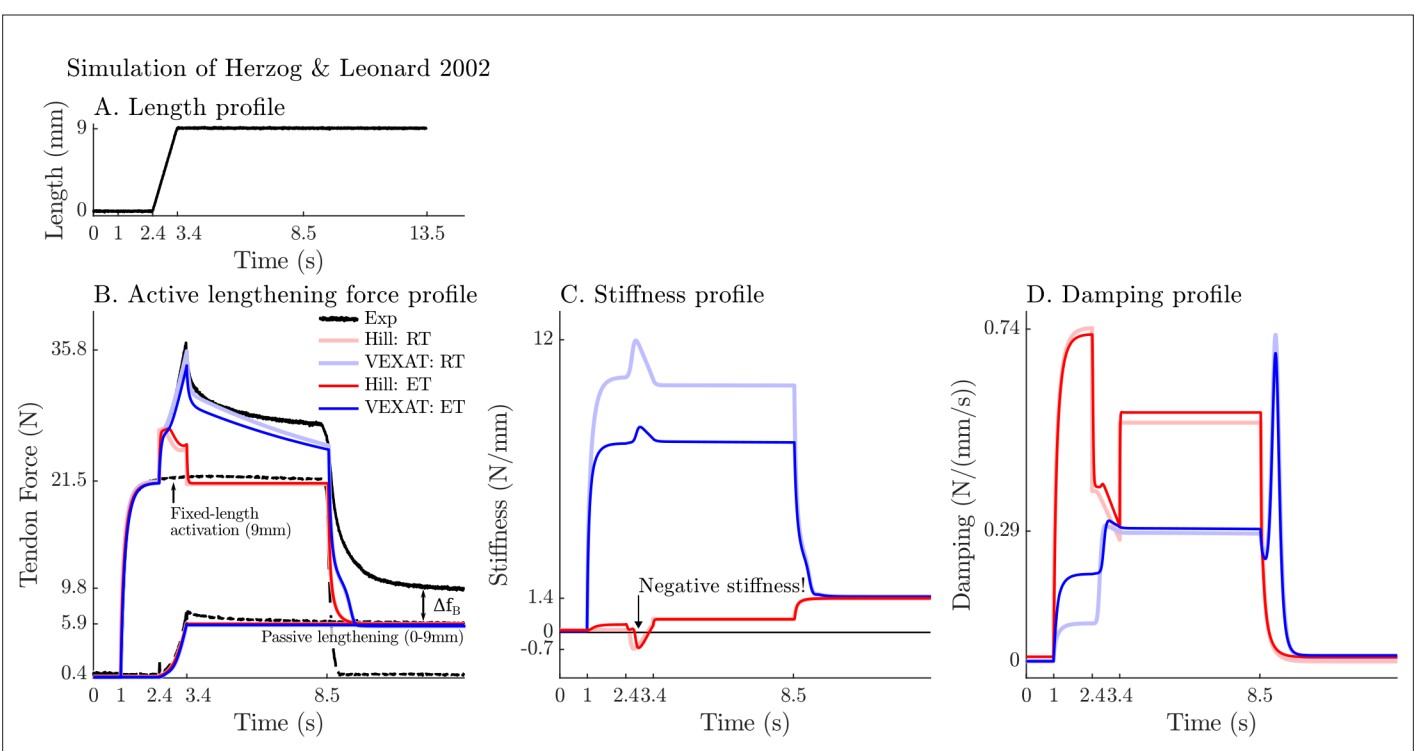

**Figure 9.** A comparison of the tension developed by the VEXAT and Hill models when actively lengthened at 9 mm/s on the descending limb of the force-length relation. *Herzog and Leonard, 2002* actively lengthened (**A**) cat soleus muscles on the descending limb of the force-length curve (where $\tilde{\ell}^M > 1$ in *Figure 2A*) and measured the force response of the MTU (**B**). After the initial transient at 2.4s the Hill model's output force drops (**B**) because of the small region of negative stiffness (**C**) created by the force-length curve. In contrast, the VEXAT model develops steadily increasing forces between 2.4 and 3.4s and has a consistent level of stiffness (**C**) and damping (**D**).

will examine the 9 mm/s ramp experiment in detail because the simulations of the 3 mm/s and 27 mm/s ramp experiments produces similar stereotypical patterns (see Appendix 7 for details).

When the active lengthening experiment of *Herzog and Leonard, 2002* is simulated (*Figure 9A*), both models produce a force transient initially (*Figure 9B*), but for different reasons. The VEXAT model's transient is created when the lumped crossbridge spring (the $\tilde{k}_o^X \tilde{\ell}^X$ term in *Equation 15*) is stretched. In contrast, the Hill model's transient is produced, not by spring forces, but by damping produced by the force-velocity curve as shown in *Equation 26*.

After the initial force transient, the response of the two models diverges (*Figure 9B*): the VEXAT model continues to develop increased tension as it is lengthened, while the Hill model's tension drops before recovering. The VEXAT model's continued increase in force is due to the titin model: when activated, a section of titin's PEVK region remains approximately fixed to the actin element (*Figure 1C*). As a result, the $\ell^2$ element (composed of part of PEVK segment and the distal Ig segment) continues to stretch and generates higher forces than it would if the muscle were being passively stretched. While both the elastic and rigid-tendon versions of the VEXAT model produce the same stereotypical ramp-lengthening response (*Figure 9B*), the rigid-tendon model develops slightly more tension because the strain of the MTU is solely borne by the CE.

In contrast, the Hill model develops less force during lengthening when it enters a small region of negative stiffness (*Figure 9B and C*) because the passive-force-length curve is too compliant to compensate for the negative slope of the active force-length curve. Similarly, the damping coefficient of the Hill model drops substantially during lengthening (*Figure 9D*). *Equation 27* and *Figure 2B* shows the reason that damping drops during lengthening: $d\mathbf{f}^V/dv^M$, the slope of the line in Fig. *Figure 2B*, is quite large when the muscle is isometric and becomes quite small as the rate of lengthening increases.

After the ramp stretch is completed (at time 3.4 s in *Figure 9B*), the tension developed by the cat soleus recovers slowly, following a profile that looks strikingly like a first-order decay. The large damping coefficient acting between the titin-actin bond slows the force recovery of the VEXAT model. We have tuned the value of $\beta_A^{\mathrm{PEVK}}$ to $71.9 f_o^M/(\ell_o^M/s)$ for the elastic-tendon model, and $77.7 f_o^M/(\ell_o^M/s)$ for the rigid-tendon model, to match the rate of force decay of the cat soleus in the data of *Herzog and Leonard, 2002*. The Hill model, in contrast, recovers to its isometric value quite rapidly. Since the Hill model's force enhancement during lengthening is a function of the rate of lengthening, when the lengthening ceases, so too does the force enhancement.

Once activation is allowed to return to zero, the data of *Herzog and Leonard, 2002* shows that the cat soleus continues to develop a tension that is $\Delta f_B$ above passive levels (*Figure 9B* for $t > 8.5s$). The force $\Delta f_B$ is known as passive force enhancement, and is suspected to be caused by titin binding to actin (*Herzog, 2019*). Since we model titin-actin forces using an activation-dependent damper, when activation goes to zero our titin model becomes unbound from actin. As such, both our model and a Hill model remain $\Delta f_B$ below the experimental data of Herzog and Leonard (*Figure 9B*) after lengthening and activation have ceased.

## Active lengthening beyond actin-myosin overlap

One of the great challenges that remains is to decompose how much tension is developed by titin (*Figure 1C*) separately from myosin (*Figure 1B*) in an active sarcomere. The active-lengthening experiment of *Leonard et al., 2010* provides some insight into this force distribution problem because they recorded active forces both within and far beyond actin-myosin overlap. The data of *Leonard et al., 2010* shows that active force continues to develop linearly during lengthening, beyond actin-myosin overlap, until mechanical failure. When activated and lengthened, the myofibrils failed at a length of $3.38\ell_o^M$ and force of $5.14 f_o^M$, on average. In contrast, during passive lengthening myofibrils failed at a much shorter length of $2.86\ell_o^M$ with a dramatically lower tension of of $1.31 f_o^M$. To show that the extraordinary forces beyond actin-myosin overlap can be ascribed to titin, *Leonard et al., 2010* repeated the experiment but deleted titin using trypsin: the titin-deleted myofibrils failed at short lengths and insignificant stresses. Using the titin model of *Equation 20* (*Figure 1A*) as an interpretive lens, the huge forces developed during active lengthening would be created when titin is bound to actin leaving the distal segment of titin to take up all of the strain. Conversely, our titin model would produce lower forces during passive lengthening because the proximal Ig, PEVK, and distal Ig regions would all be lengthening together (*Figure 3A*).

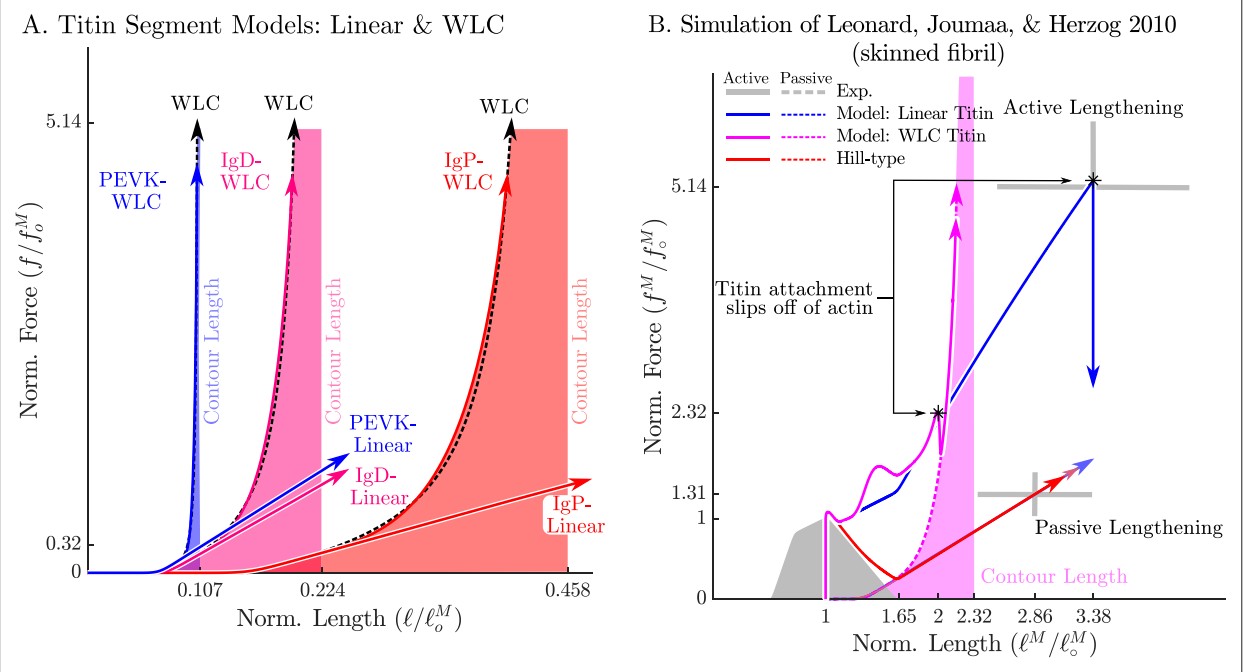

**Figure 10.** The passive force-length relations of two diffferent models of titin: a linear extrapolation and a WLC model. We consider two different versions of the force-length relation for each titin segment of the VEXAT model (**A**): a linear extrapolation, and a WLC model extrapolation. *Leonard et al., 2010* observed that active myofibrils continue to develop increasing amounts of tension beyond actin-myosin overlap (**B**, grey lines with ±1 standard deviation shown). When this experiment is replicated using the VEXAT model (**B**, blue & magenta lines) and a Hill model (**C** red lines), only the VEXAT model with the linear extrapolated titin model is able to replicate the experiment with the titin-actin bond slipping off of the actin filament at $3.38\,\ell_{\circ}^{\mathrm{M}}$.

Since the experiment of *Leonard et al., 2010* was performed on skinned rabbit myofibrils and not on whole muscle, both the VEXAT and Hill models had to be adjusted prior to simulation (see Appendix 8 for parameter values). To simulate a rabbit myofibril we created a force-length curve (*Rassier et al., 1999*) consistent with the filament lengths of rabbit skeletal muscle (*Higuchi et al., 1995*; 1.12 μm actin, 1.63 μm myosin, and 0.07 μm z-line width) and fit the force-length relations of the two titin segments to be consistent with the structure measured by *Prado et al., 2005* of rabbit psoas titin consisting of a 70–30% mix of a 3300kD and a 3400kD titin isoform (see Appendix 2.4 for fitting details and Appendix 8 for parameter values). Since this is a simulation of a fibril, we used a rigid-tendon of zero length (equivalent to ignoring the tendon), and set the pennation angle to zero.

As mentioned in the 'Model' section, because this experiment includes extreme lengths, we consider two different force-length relations for each segment of titin (*Figure 10A*): a linear extrapolation, and an extension that follows the WLC model. While both versions of the titin model are identical up to $\tilde{\ell}_{\mathrm{toe}}^{\mathrm{PE}}$, beyond $\tilde{\ell}_{\mathrm{toe}}^{\mathrm{PE}}$ the WLC model continues to develop increasingly large forces until all of the Ig domains and PEVK residues have been unfolded and the segments of titin reach a physical singularity: at this point the Ig domains and PEVK residues cannot be elongated any further without breaking molecular bonds (see Appendix 2.4 for details). Our preliminary simulations indicated that the linear titin model's titin-actin bond was not strong enough to support large tensions, and so we increased the value of $\beta_{\mathrm{A}}^{\mathrm{PEVK}}$ from 71.9 to 975 (compare *Table 1H* and *Appendix 8—table 1G*).

The Hill model was similarly modified, with the pennation angle set to zero and coupled with a rigid-tendon of zero length. Since the Hill model lacks an ECM element the passive-force-length curve was instead fitted to match the passive forces produced in the data of *Leonard et al., 2010*. No adjustments were made to the active elements of the Hill model.

When the slow active stretch (0.1 μm/sarcomere/s) of the experiment of *Leonard et al., 2010* is simulated, only the VEXAT model with the linear titin element can match the experimental data of *Leonard et al., 2010* (*Figure 10B*). The Hill model cannot produce active force for lengths greater than $1.62\ell_{\circ}^{\mathrm{M}}$ since the active force-length curve goes to zero (*Figure 2A* and *Figure 10B*) and the model lacks any element capable of producing force beyond this length. In contrast, the linear titin

model continues to develop active force until a length of 3.38 $\ell_o^M$ is reached, at which point the titin-actin bond is pulled off the end of the actin filament and the active force is reduced to its passive value.

The WLC titin model is not able to reach the extreme lengths observed by *Leonard et al., 2010*. The distal segment of the WLC titin model approaches its contour length early in the simulation and ensures that the the titin-actin bond is dragged off the end of the actin filament at 1.99 $\ell_o^M$ (*Figure 10B*). After 1.99 $\ell_o^M$ (*Figure 10B*), the tension of the WLC titin model drops to its passive value but continues to increase until the contour lengths of all of the segments of titin are reached at 2.32 $\ell_o^M$. Comparing the response of the linear model to the WLC titin model two things are clear: the linear titin model more faithfully follows the data of *Leonard et al., 2010*, but does so with titin segment lengths that exceed the maximum contour length expected for the isoform of titin in a rabbit myofibril.

This simulation has also uncovered a surprising fact: the myofibrils in the experiments of *Leonard et al., 2010* do not fail at 2.32 $\ell_o^M$, as would be expected by the WLC model of titin, but instead reach much greater lengths (*Figure 10B*). Physically, it may be possible for a rabbit myofibril to reach these lengths (without exceeding the contour lengths of the proximal Ig, PEVK, and distal Ig segments) if the bond between the distal segment of titin and myosin breaks down. This would allow the large Ig segment, that is normally bound to myosin, to uncoil and continue to develop the forces observed by *Leonard et al., 2010*. Unfortunately the mechanism which allowed the samples in Leonard et al.'s experiments to develop tension beyond titin's contour length remains unknown.

## Force-length and force-velocity

Although the active portion of the Hill model is embedded in *Equation 16*, it is not clear if the VEXAT model can still replicate force-velocity experiments of *Hill, 1938* and the force-length experiments of *Gordon et al., 1966*. Here, we simulate both of these experiments using the cat soleus model that we have used for the simulations described in the 'Model' section ('Stochastic length perturbation experiments') and compare the results to the force-length and force-velocity curves that are used in the Hill model and in *Equation 16* of the VEXAT model.

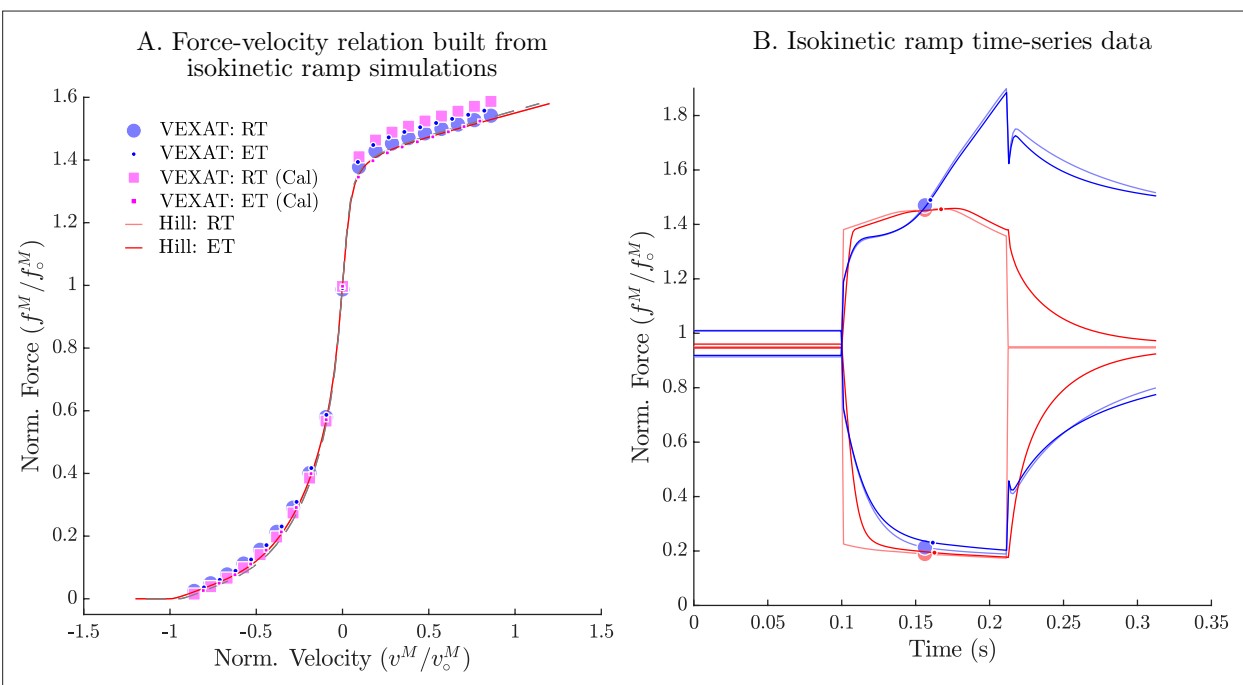

**Figure 11.** The force-velocity relation of the VEXAT and Hill models. When the experiments of *Hill, 1938* are simulated (**A**), the VEXAT model produces a force-velocity profile (blue dots) that approaches zero more rapidly during shortening than the embedded profile $\mathbf{f}^V(\cdot)$ (red lines). By scaling $v_{max}^M$ by 0.95 the VEXAT model (magenta squares) is able to closely follow the force-velocity curve of the Hill model. While the force-velocity curves between the two models are similar, the time-domain force response of the two models differs substantially (**B**). The rigid-tendon Hill model exhibits a sharp nonlinear change in force at the beginning (0.1 s) and ending (0.21 s) of the ramp stretch.

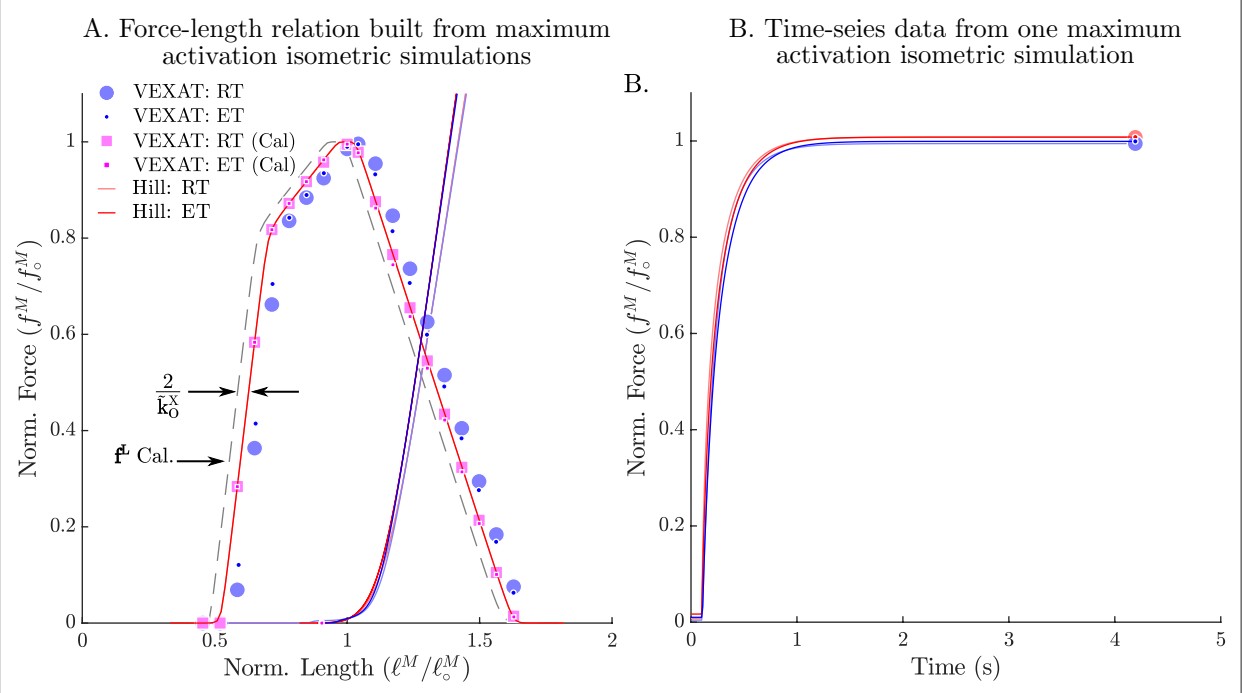

**Figure 12.** The passive and active force-length relation of the VEXAT and Hill models. When the passive and active force-length experiments of *Gordon et al., 1966* are simulated, the VEXAT model (blue dots) and the Hill model (red lines) produce slightly different force-length curves (**A**) and force responses in the time-domain (**B**). The VEXAT model produces a right shifted active force-length curve, when compared to the Hill model due to the series elasticity of the XE element. By shifting the underlying curve by $\frac{2}{k_o^X}$ to the left the VEXAT model (magenta squares) can be made to exactly match the force-length characteristic of the Hill model.

The force-velocity experiment of *Hill, 1938* is simulated by activating the model, and then by changing its length to follow a shortening ramp and a lengthening ramp. During shortening experiments, the CE shortens from $1.1\ell_o^M$ to $0.9\ell_o^M$ with the measurement of active muscle force is made at $\ell_o^M$. Lengthening experiments are similarly made by measuring muscle force mid-way through a ramp stretch that begins at $0.9\ell_o^M$ and ends at $1.1\ell_o^M$. When an elastic-tendon model is used, we carefully evaluate initial and terminal path lengths to accommodate for the stretch of the tendon so that the CE still shortens from $1.1\ell_o^M$ to $0.9\ell_o^M$ and lengthens from $0.9\ell_o^M$ to $1.1\ell_o^M$.

The VEXAT model produces forces that differ slightly from the $\mathbf{f}^V$ that is embedded in *Equation 16* while the Hill model reproduces the curve (*Figure 11*). The maximum shortening velocity of the VEXAT model is slightly weaker than the embedded curve due to the series viscoelasticity of the XE element. Although the model can be made to converge to the $\mathbf{f}^V$ curve more rapidly by decreasing $\tau^S$ this has the undesirable consequence of degrading the low-frequency response of the model when simulating the experiments of *Kirsch et al., 1994* (particularly *Figure 6C., and F.*). These small differences can be effectively removed by scaling $v_{max}^M$ by $s^V$ (*Figure 11A* has $s^V = 0.95$) to accommodate for the small decrease in force caused by the viscoelastic XE element.

The force-length experiments of *Gordon et al., 1966* were simulated by first passively lengthening the CE, and next by measuring the active force developed by the CE at a series of fixed lengths. Prior to activation, the passive CE was simulated for a brief period of time in a passive state to reduce any history effects due to the active titin element. To be consistent with the experiment of *Gordon et al., 1966*, we subtracted off the passive force from the active force before producing the active-force-length profile.

The simulation of *Gordon et al., 1966* shows that the VEXAT model (*Figure 12A*, blue dots) produces a force-length profile that is shifted to the right of the Hill model (*Figure 12A*, red line) due to the series elasticity introduced by the XE. We can solve for the size of this rightwards shift by noting that *Equation 16* will drive the $\tilde{\ell}^S$ to a length such that the isometric force developed by the XE is equal to that of the embedded Hill model

$$a\mathbf{f}^{L}(\tilde{\ell}^{S} + \tilde{L}^{M})\,\tilde{k}_{o}^{X}\tilde{\ell}^{X} = a\mathbf{f}^{L}(\tilde{\ell}^{S} + \tilde{L}^{M}) \tag{30}$$

allowing us to solve for

$$\tilde{\ell}^{X} = \frac{1}{\tilde{k}_{o}^{X}} \tag{31}$$

the isometric strain of the XE. Since there are two viscoelastic XE elements per CE, the VEXAT model has an active force-length characteristic that shifted to the right of the embedded $\mathbf{f}^{L}$ curve by a constant $\frac{2}{\tilde{k}_{o}^{X}}$. This shift, $\Delta^{L}$, can be calibrated out of the model (*Figure 12A*, magenta squares) by adjusting the $\mathbf{f}^{L}(\cdot)$ curve so that it is $\frac{2}{\tilde{k}_{o}^{X}}$ to the left of its normal position. Note that all simulations described in the previous sections made use of the VEXAT model with the calibrated force-length relation and the calibrated force-velocity relation.

## Discussion

A muscle model is defined by the experiments it can replicate and the mechanisms it embodies. We have developed the VEXAT muscle model to replicate the force response of muscle to a wide variety of perturbations (*Kirsch et al., 1994*; *Herzog and Leonard, 2002*; *Leonard et al., 2010*) while also retaining the ability to reproduce the force-velocity experiment of *Hill, 1938* and the force-length experiment of *Gordon et al., 1966*. The model we have developed uses two mechanisms to capture the force response of muscle over a large variety of time and length scales: first, a viscoelastic cross-bridge element that over brief time-scales appears as a spring-damper, and at longer time-scales mimics a Hill-model; second, a titin element that is capable of developing active force during large stretches.

The viscoelastic crossbridge and titin elements we have developed introduce a number of assumptions into the model. While there is evidence that the activation-dependent stiffness of muscle originates primarily from the stiffness of the attached crossbridges (*Veigel et al., 1998*), the origins of the activation-dependent damping observed by *Kirsch et al., 1994* have not yet been established. We assumed that, since the damping observed by *Kirsch et al., 1994* varies linearly with activation, the damping originates from the attached crossbridges. Whether this damping is intrinsic or is due to some other factor remains to be established. Next, we have also assumed that the force developed by the XE converges to a Hill model (*Millard et al., 2013*) given enough time (*Equation 16*). A recent experiment (see Figure 7 of *Tomalka et al., 2021*) suggests the force developed by the XE might decrease during lengthening rather than increasing as is typical of a Hill model (*Millard et al., 2013*). If the observations of *Tomalka et al., 2021* can be replicated, the VEXAT model will need to be adjusted so that the XE element develops less force during active lengthening while the active-titin element develops more force. Finally, we have assumed that actin-myosin sliding acceleration (due to crossbridge cycling) occurs when there is a force imbalance between the external force applied to the XE and the internal force developed by the XE as shown in *Equation 16*. This assumption is a departure from previous models: Hill-type models (*Zajac, 1989*; *Millard et al., 2013*) assume that the tension applied to the muscle instantaneously affects the actin-myosin sliding velocity; Huxley models (*Huxley, 1957*) assume that the actin-myosin sliding velocity directly affects the rate of attachment and detachment of crossbridges.

The active titin model that we have developed makes assumptions similar to *Rode et al., 2009* and *Schappacher-Tilp et al., 2015*: some parts of the PEVK segment bond to actin, and this bond cannot do any positive work on titin. The assumption that the bond between titin and actin cannot do positive work means that titin cannot be actively preloaded: it can only develop force when it is stretched. In contrast, two mechanisms have been proposed that make it possible for titin to be preloaded by crossbridge cycling: the winding filament theory of *Nishikawa et al., 2012* and the titin entanglement hypothesis of *DuVall et al., 2016*. If titin were significantly preloaded by crossbridge cycling, the titin load path would support higher forces and the myosin-actin load path would bear less force. Accordingly, the overall stiffness of the CE would be reduced, affecting our simulations of *Kirsch et al., 1994*: lower myosin-actin loads mean fewer attached crossbridges, since crossbridges are stiff in comparison to titin, the stiffness of the CE would decrease (see Appendix 1). Hopefully experimental work will clarify if titin can be actively preloaded by crossbridges in the future.

Both the viscoelastic crossbridge and active titin elements include simple myosin-actin and titin-actin bond models that improve accuracy but have limitations. First, the viscoelastic crossbridge element has been made to represent a population of crossbridges in which the contribution of any single crossbridge is negligible. Although it may be possible for the XE model to accurately simulate a maximally activated single sarcomere (which has roughly 20 attached crossbridges per half sarcomere *Huxley and Simmons, 1971*; *Howard, 1997*), the accuracy of the model will degrade as the number of attached crossbridges decreases. When only a single crossbridge remains, the XE model's output will be inaccurate because it can only generate force continuously while a real crossbridge generates force discretely each time it attaches to, and detaches from, actin. Next, we have used two equations, *Equation 16* and *Equation 21*, that assume myosin-actin and titin-actin interactions are temperature-invariant and scale linearly with size ($\ell_o^M$ and $f_o^M$). In contrast, myosin-actin interactions and some titin-actin interactions are temperature-sensitive (*Minajeva et al., 2002*; *Roots et al., 2012*) and may not scale linearly with size. In the 'Active lengthening beyond actin-myosin overlap' section we had to adjust the active titin damping parameter, $\beta_A^{PEVK}$, to simulate the myofibril experiments of *Leonard et al., 2010*, perhaps because the assumptions of temperature-invariance and size-linearity were not met: the initial value for $\beta_A^{PEVK}$ came from fitting to in-situ experimental data (*Herzog and Leonard, 2002*) from whole muscle that was warmer ($35 - 36.5°C$ vs $20 - 21°C$) and larger ($\ell_o^M$ of 42.9mm vs. $10 - 15\mu m$) than the myofibrils (*Leonard et al., 2010*). While the cat soleus XE and titin model parameters (*Table 1G, H1*) can be used as rough default values, these parameters should be refit to accurately simulate muscle that differs in scale or temperature from cat soleus. Finally, the VEXAT model in its current form ignores phenomena related to submaximal contractions: the shift in the peak of the force-length relation (*Stephenson and Wendt, 1984*), and the scaling of the maximum shortening velocity (*Chow and Darling, 1999*). We hope to include these phenomena in a later version of the VEXAT model to more accurately simulate submaximal contractions.

The model we have proposed can replicate phenomena that occur at a wide variety of time and length scales: the experiments of *Kirsch et al., 1994* which occur over small time and length scales; and the active lengthening experiments of *Herzog and Leonard, 2002* and *Leonard et al., 2010* which occur over physiological and supra-physiological length scales. In contrast, we have shown in the 'Biological benchmark simulations' section that a Hill-type model compares poorly to biological muscle when the same set of experiments are simulated. We expect that a Huxley model (*Huxley, 1957*) is also likely to have difficulty reproducing the experiment of *Kirsch et al., 1994* because the model lacks an active damping element. Since titin was discovered (*Maruyama, 1976*) long after Huxley's model was proposed (*Huxley, 1957*), a Huxley model will be unable to replicate any experiment that is strongly influenced by titin such as the experiment of *Leonard et al., 2010*.

Although there have been several more recent muscle model formulations proposed, none have the properties to simultaneously reproduce the experiments of *Kirsch et al., 1994*, *Herzog and Leonard, 2002*, *Leonard et al., 2010*, *Hill, 1938*, and *Gordon et al., 1966*. Linearized impedance models (*Hogan, 1985*; *Mussa-Ivaldi et al., 1985*) can reproduce the experiments of *Kirsch et al., 1994*, but these models lack the nonlinear components needed to reproduce the force-length experiments of *Gordon et al., 1966*, and the force-velocity experiments of *Hill, 1938*. The models of *Forcinito et al., 1998*, and *Tahir et al., 2018* have a structure that places a contractile element in series with an elastic-tendon. While this is a commonly used structure, at high frequencies the lack of damping in the tendon will drive the phase shift between length and force to approach zero. The measurements and model of *Kirsch et al., 1994*, in contrast, indicate that the phase shift between length and force approaches ninety degrees with increasing frequencies. Although the Hill-type models of *Haeufle et al., 2014* and *Günther et al., 2018* have viscoelastic tendons, these models have no representation of the viscoelasticity of the CE's attached crossbridges. Similar to the Hill-type muscle model evaluated in this work (*Millard et al., 2013*), it is likely that models of *Haeufle et al., 2014* and *Günther et al., 2018* will not be able to match the frequency response of biological muscle. While the model of *Tamura and Saito, 2002* is one of the few models that can develop force-enhancement and force-depression (*Tamura et al., 2005*), it is unlikely that this model will be able to reproduce the frequency response of biological muscle because it uses spring-damping elements in series: *Kirsch et al., 1994* showed that the frequency response of spring-damping elements in series poorly fits the response of biological muscle. The models of *De Groote et al., 2017* and *De Groote et al., 2018* introduced a short-range-stiffness element in parallel to a Hill model to capture the stiffness of biological muscle.

While the formulations presented in *De Groote et al., 2017* and *De Groote et al., 2018* improves upon a Hill model it is unlikely to reproduce the experiment of *Kirsch et al., 1994* because we have shown in the 'Active lengthening beyond actin-myosin overlap' section that a Hill model has a frequency response that differs from biological muscle. The muscle model of *Rode et al., 2009* uses a Hill model for the CE and so we expect that this model will have the same difficulties reproducing the experiment of *Kirsch et al., 1994*. The model of *Schappacher-Tilp et al., 2015* extends a Huxley model (*Huxley, 1957*) by adding a detailed titin element. Similar to a Huxley model, the model of *Schappacher-Tilp et al., 2015* will likely have difficulty reproducing the experiment of *Kirsch et al., 1994* because it is missing an active damping element.

While developing this model, we have come across open questions that we hope can be addressed in the future. How does muscle stiffness and damping change across the force-length curve? Does stiffness and damping change with velocity? What are the physical origins of the active damping observed by *Kirsch et al., 1994*? What are the conditions that affect passive-force enhancement, and its release? In addition to pursuing these questions, we hope that other researchers continue to contribute experiments that are amenable to simulation, and to develop musculotendon models that overcome the limitations of our model. To help others build upon our work, we have made the source code of the model and all simulations presented in this paper available online (see the elife 2023 branch of https://github.com/mjhmilla/Millard2021ImpedanceMuscle; copy archived at *Millard, 2024*).

## Acknowledgements

Financial support is gratefully acknowledged from the Deutsche Forschungsgemeinschaft (DFG, German Research Foundation) under Germany's Excellence Strategy (EXC 2075 – 390740016) through the Stuttgart Center for Simulation Science (SimTech), from DFG grant no. MI 2109/1–1, the Lighthouse Initiative Geriatronics by StMWi Bayern (Project X, grant no. 5140951), and the Natural Sciences and Engineering Research Council of Canada (RGPIN-2020–03920).

## Additional information

### Funding

| Funder | Grant reference number | Author |
| --- | --- | --- |
| Deutsche Forschungsgemeinschaft | EXC 2075 - 390740016 | Matthew Millard |
| Deutsche Forschungsgemeinschaft | MI 2109/1-1 | Matthew Millard |
| Bayerische Staatsministerium für Wirtschaft, Landesentwicklung und Energie | Project X, grant no. 5140951 | David W Franklin |
| Natural Sciences and Engineering Research Council of Canada | RGPIN-2020-03920 | Walter Herzog |

The funders had no role in study design, data collection and interpretation, or the decision to submit the work for publication.

### Author contributions

Matthew Millard, Conceptualization, Resources, Data curation, Software, Formal analysis, Funding acquisition, Validation, Investigation, Visualization, Methodology, Writing – original draft, Project administration, Writing – review and editing; David W Franklin, Supervision, Writing – review and editing, DF exchanged emails and conversations with MM over many years that were key to the development of the viscoelastic crossbridge model; Walter Herzog, Supervision, Writing – review and editing, WH exchanged hundreds of emails with MM discussing ideas and providing experimental literature references that were key to the development of the active titin model

## Author ORCIDs
Matthew Millard ⓘ https://orcid.org/0000-0001-7627-564X
David W Franklin ⓘ https://orcid.org/0000-0001-9530-0820
Walter Herzog ⓘ https://orcid.org/0000-0002-5341-0033

Reviewer #1 (Public review): https://doi.org/10.7554/eLife.88344.4.sa1
Reviewer #2 (Public review): https://doi.org/10.7554/eLife.88344.4.sa2
Author response https://doi.org/10.7554/eLife.88344.4.sa3

## Additional files

### Supplementary files
• MDAR checklist

### Data availability
All of the simulation results in this study can be generated using the code available from the elife2023 branch of https://github.com/mjhmilla/Millard2023VexatMuscle (copy archived at *Millard, 2024*). All of the code is publicly available either under the APACHE-2 or MIT licenses as indicated in the file header and also by the licensing auditing tool https://api.reuse.software/. The code repository also includes a selection of manually digitized data sets from past papers in the 'experiment' folder, as well as raw experimental data in the 'HerzogLeonard2002' folder that WH has made publicly available.

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

## Appendix 1

### The stiffness of the actin-myosin and titin load paths

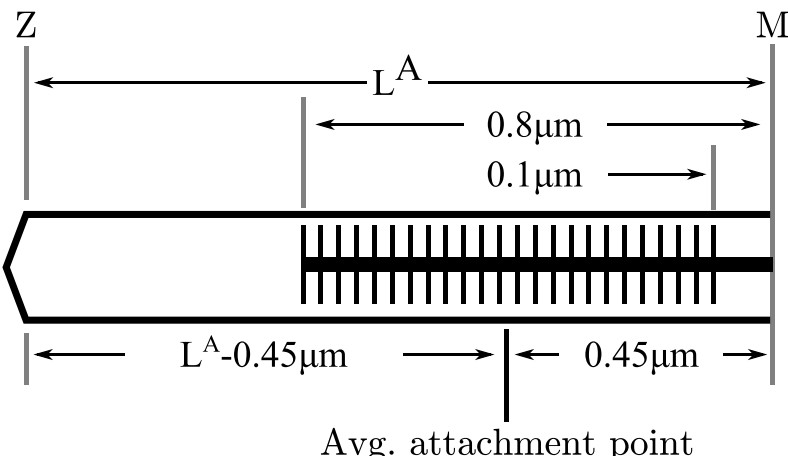

**Appendix 1—figure 1.** The average point of attachment between myosin and actin in a half-sarcomere at the optimal length. To evaluate the stiffness of the actin-myosin load path, we first determine the average point of attachment. Since the actin filament length varies across species we label it $L^A$. Across rabbits, cats and human skeletal muscle myosin geometry is consistent *Rassier et al., 1999*: a half-myosin is 0.8 μm in length with a 0.1 μm bare patch in the middle. Thus at full overlap the average point of attachment is 0.45 μm from the M-line, or $L^A - 0.45$ μm from the Z-line at $\ell_o^M$. The lumped stiffness of the actin-myosin load path of a half-sarcomere is the stiffness of three springs in series: a spring representing a $L^A - 0.45$ μm length of actin, a spring representing the all attached crossbridges, and a spring representing a 0.45 μm section of myosin.

A single half-myosin can connect to the surrounding six actin filaments through 97.9 crossbridges. A 0.800 μm half-myosin has a pair crossbridges over 0.700 μm of its length every 14.3nm which amounts to 97.9 per half-myosin (*Huxley, 1969*). Although 97.9 crossbridges does not make physical sense, here we will evaluate the stiffness of the CE assuming that fractional crossbridges exist and that attached crossbridges can be perfectly distributed among the six available actin filaments: the alternative calculation is more complicated and produces stiffness values that differ only in the 3rd significant digit. Assuming a duty cycle of 20% (*Howard, 1997*) (values between 5–90% have been reported by *Finer et al., 1994*), at full actin-myosin overlap there will be 19.6 crossbridges attached to the six surrounding actin filaments. Assuming that these 19.6 crossbridges are evenly distributed between the six actin filaments, each single actin will be attached to 3.26 attached crossbridges.

At full overlap, the Z-line is 1 actin filament length $L^A$ (1.12 μm in rabbits according to *Higuchi et al., 1995*) from the M-line. The average point of crossbridge attachment is in the middle of the half-myosin at a distance of 0.45 μm from the M-line (0.1 μm is bare and 0.35 μm is half of the remaining length), which is $L^A - 0.45$ μm from the Z-line (*Figure 1*, *Appendix 1—figure 1*). A single actin filament has a stiffness of 48 − 68pN/nm (*Higuchi et al., 1995*) while a single crossbridge has a stiffness of 0.69 ± 0.47pN/nm (*Veigel et al., 1998*). Since stiffness scales inversely with length, actin's stiffness between the Z-line and the average attachment point is 81.8 − 121pN/nm. Finally, the stiffness of each actin filament and its 3.26 attached crossbridges is 0.712 − 3.67pN/nm and all six together have a stiffness of 4.27 − 22.0pN/nm.

Myosin has a similar stiffness as a single actin filament (*Tajima et al., 1994*), with the section between the average attachment point and the M-line having a stiffness of 76.9 − 113pN/nm. The final active stiffness of half-sarcomere is 4.05 − 18.4pN/nm which comes from comes from the series connection of the group of six actin filaments, with 19.6 crossbridges, and finally the single myosin filament. When this procedure is repeated assuming that only a single cross-bridge is attached the stiffness drops to 0.22 − 1.15pN/nm, which is slightly less than the stiffness of a single crossbridge (see main_ActinMyosinAndTitinStiffness.m in the elife2023 branch of accompanying code repository for details).

The force-length profile of a single rabbit titin has been measured by *Kellermayer et al., 1997* using laser tweezers to apply cyclical stretches. By digitizing *Figure 4B* (blue line) of *Kellermayer*

*et al., 1997* we arrive at a stiffness for titin of 0.0058 − 0.0288pN/nm at 2µm (for a total sarcomere length of 4µm or $1.62\ell_o^M$), and 0.0505 − 0.0928pN/nm at 4µm (8µm or $3.25\ell_o^M$). Since there are 6 titin filaments acting in parallel for each half-sarcomere, we end up with the total stiffness for titin ranging between 0.0348 − 0.173pN/nm at 2µm and 0.303 − 0.557pN/nm at 4µm. When activated, the stiffness of our rabbit psoas linear-titin model (described in the 'Active lengthening beyond actin-myosin overlap' section and fitted in Appendix 2.4, and with the parameters shown in Appendix 8) doubles, which would increase titin's stiffness to 0.0696 − 0.346pN/nm at 2µm and 0.606 − 1.11pN/nm at 4µm.

Comparing the actin-myosin and titin stiffness ranges (*Figure 2*, *Appendix 1—figure 2*) makes it clear that the stiffness of actin-myosin with 1 attached crossbridge (AM:Low in *Figure 2*, *Appendix 1—figure 2*) is comparable to the highest stiffness values we have estimated for titin (AT:High in *Figure 2*, *Appendix 1—figure 2*). When all 20% of the available crossbridges are attached (AM:High in *Figure 2*, *Appendix 1—figure 2*), the average stiffness of the actin-myosin load path is roughly one order of magnitude stiffer than the highest stiffness values of titin (AT:High in *Figure 2*, *Appendix 1—figure 2*), and two to three orders of magnitude higher than the lowest stiffness titin load path (PT:Low in *Figure 2*). Similarly, the maximum XE stiffness and titin stiffness in this work are separated by roughly an order of magnitude: the cat soleus model has a XE stiffness of $49.1f_o^M/\ell_o^M$ and maximum active titin stiffness of $8.42f_o^M/\ell_o^M$ (*Table 1*); while the rabbit psoas fibril model has a XE stiffness of $49.1f_o^M/\ell_o^M$ and maximum active titin stiffness of $5.25f_o^M/\ell_o^M$ (Appendix 8).

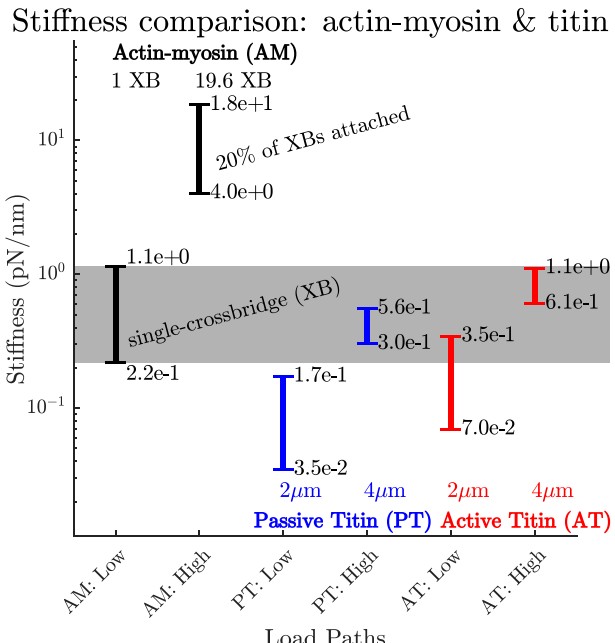

**Appendix 1—figure 2.** A comparison of how actin-myosin and titin's stiffness varies. The stiffness of a rabbit's actin-myosin load path with a single attached crossbridge (1 XB) exceeds the stiffness of its titin filament at lengths of 2µm ($1.62\ell_o^M$) (compare AM:Low to PT:Low and PT:High). Only when titin is stretched to 4µm ($3.25\ell_o^M$) does its stiffness (PT:High and AT:High) become comparable to the actin-myosin with a single attached crossbridge (AM:Low). At higher activations and modest lengths, the stiffness of the actin-myosin load path (AM: High) exceeds the stiffness of titin (PT: Low and AT:Low) by between two and three orders of magnitude. At higher activations and longer lengths, the stiffness of the actin-myosin load path (AM: High) exceeds the stiffness of titin by roughly an order of magnitude (PT:High and AT:High).

## Appendix 2

### 2.1. Model fitting

Many of the experiments simulated in this work (**Kirsch et al., 1994**; **Herzog and Leonard, 2002**) have been performed using cat soleus muscle. While we have been able to extract some architectural parameters directly from the experiments we simulate ($f_o^M$ and $\ell_o^M$ from **Herzog and Leonard, 2002**) from **Herzog and Leonard, 2002**), we have had to rely on the literature mentioned in **Table 1** for the remaining parameters. The remaining properties of the model can be solved by first building a viscoelastic damping model of the tendon; next, by solving for the intrinsic stiffness and damping properties of the CE; and finally, by fitting the passive curves ($f^1(\tilde{\ell}^1)$ and $\mathbf{f}^2(\tilde{\ell}^2)$) to simultaneously fit the passive force-length curve recorded by **Herzog and Leonard, 2002**, using a mixture of tension from titin and the ECM that is consistent with the data of **Prado et al., 2005**, all while maintaining the geometric relationship between $\mathbf{f}^{IgP}$ and $\mathbf{f}^{PEVK}$ as measured by **Trombitás et al., 1998b**.

### 2.2. Fitting the tendon's stiffness and damping

Similar to previous work (**Millard et al., 2013**), we model the force-length relation of the tendon using a quintic Bézier spline (**Appendix 2—figure 1A**) that begins at $(\tilde{\ell}^T, \tilde{f}^T) = (1.0, 0)$ (where $\tilde{\ell}^T$ is tendon length normalized by $\ell_s^T$, and $\tilde{f}^T$ is tension normalized by $f_o^M$), ends at $(\tilde{\ell}^T, \tilde{f}^T) = (1.0 + e_{toe}^T, f_{toe}^T)$ with a normalized stiffness of $\tilde{k}^T$, and uses the constants $f_{toe}^T = 2/3$ and $\tilde{k}^T = 1.375/e_o^T$ (given $30.0 f_o^M/\ell_o^M$ from **Scott and Loeb, 1995**, $e_o^T$ is thus 4.58%). Using the experimental data of **Netti et al., 1996**, we have also constructed a curve to evaluate the damping coefficient of the tendon. The normalized tendon stiffness (termed storage modulus by **Netti et al., 1996**) and normalized tendon damping (termed loss modulus by **Netti et al., 1996**) both have a similar shape as the tendon is stretched from slack to $e_o^T$ (**Appendix 2—figure 1B and C**). The similarity in shape is likely not a coincidence.

The nonlinear force-length characteristics (**Appendix 2—figure 1A**) of tendon originate from its microscopic structure. Tendon is composed of many fiber bundles with differing resting lengths (**Netti et al., 1996**). Initially, the tendon's fiber bundles begin crimped, but gradually stretch as the tendon lengthens, until finally all fiber bundles are stretched and the tendon achieves its maximum stiffness (**Appendix 2—figure 1B**) and damping (**Appendix 2—figure 1C**; **Netti et al., 1996**). Accordingly, in **Equation 23**, we have described the normalized damping of the tendon as being equal to the normalized stiffness of the tendon scaled by a constant $U$. To estimate $U$, we have used the measurements of **Netti et al., 1996**; **Figure 1B and C** and have solved a least-squares problem

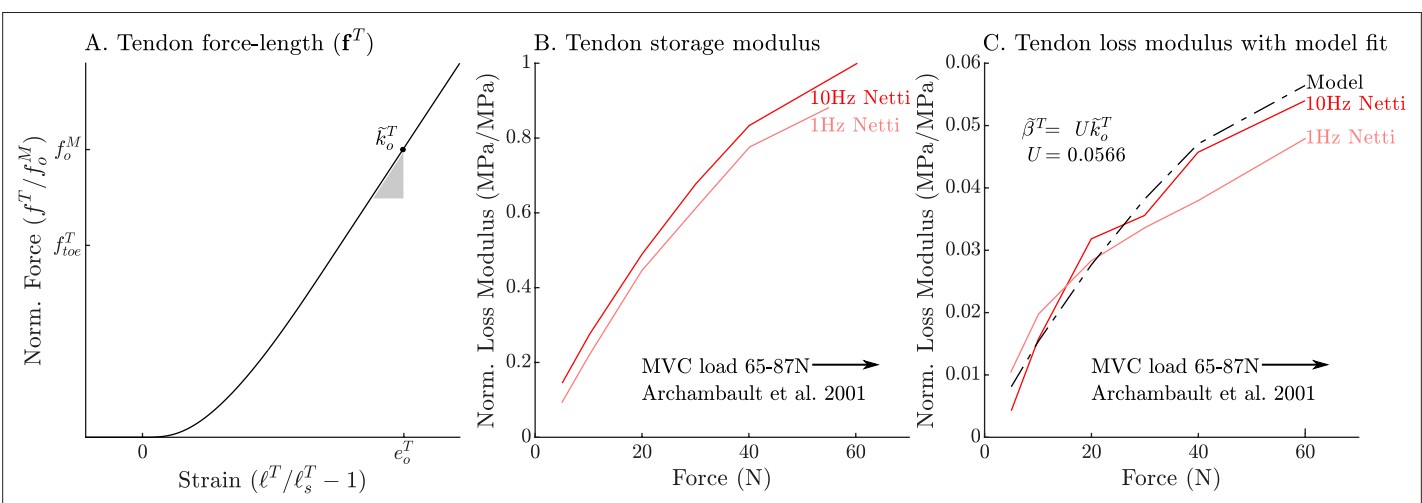

**Appendix 2—figure 1.** The models used to define the force, storage modulus, and loss modulus of tendon. The normalized tendon force-length curve (**A**) has been been fit to match the cat soleus tendon stiffness measurements of **Scott and Loeb, 1995**. The data of **Netti et al., 1996** allow us to develop a model of tendon damping as a linear function of tendon stiffness. By normalizing the measurements of **Netti et al., 1996** by the maximum storage modulus we obtain curves that are equivalent to the normalized stiffness (**B**) and damping (**C**) of an Achilles tendon from a rabbit. Both normalized tendon stiffness and damping follow similar curves, but at different scales, allowing us to model tendon damping as a linear function of tendon stiffness (**C**).

$$\min \sum_{i}^{n} \left( \left( U\hat{k}_i^{\mathrm{T}} \right) - \hat{\beta}_i^{\mathrm{T}} \right)^2 \tag{32}$$

to arrive at a value of $U = 0.057$. The resulting damping model (*Figure 1C*) fits the measurements of *Netti et al., 1996* closely.

## 2.3. Fitting the CE's Impedance

We can now calculate the normalized impedance of the XE using the viscoelastic-tendon model we have constructed and the measurements of *Kirsch et al., 1994* of the impedance of the entire MTU. Since an MTU is structured with a CE in series with a tendon, the compliance of the MTU is given by

$$\frac{1}{k^{\mathrm{M}}} = \frac{1}{k_{\mathrm{AT}}^{\mathrm{M}}} + \frac{1}{k^{\mathrm{T}}} \tag{33}$$

where $k_{\mathrm{AT}}^{\mathrm{M}}$ is the stiffness of the CE in the direction of the tendon. We can calculate $k^{\mathrm{M}}$ directly by fitting a line to the stiffness vs tension plot that appears in Figure 12 of *Kirsch et al., 1994* (0.8 mm, 0–35 Hz perturbation) and resulting in $k^{\mathrm{M}} = 2.47$ N/mm at a nominal force of 5 N. Here we use a nominal tension of 5 N so that we can later compare our model to the 5 N frequency response reported in Figure 3 of *Kirsch et al., 1994*. Since *Kirsch et al., 1994* did not report the architectural properties of the experimental specimens, we assume that the architectural properties of the cat used in Kirsch et al.'s experiments are similar to the properties listed in *Table 1*. We evaluate the stiffness of the tendon model by stretching it until it develops the nominal tension of Kirsch et al.'s *Figure 3* data (5 N), and then compute its derivative which amounts to $k^{\mathrm{T}} = 16.9$ N/mm. Finally, using *Equation 33*, we can solve for a value of $k_{\mathrm{AT}}^{\mathrm{M}} = 2.90$ N/mm. Since the inverse of damping adds for damping elements in series

$$\frac{1}{\beta^{\mathrm{M}}} = \frac{1}{\beta_{\mathrm{AT}}^{\mathrm{M}}} + \frac{1}{\beta^{\mathrm{T}}} \tag{34}$$

we can use a similar procedure to evaluate $\beta_{\mathrm{AT}}^{\mathrm{M}}$, the damping of the CE along the tendon. The value of $\beta^{\mathrm{M}}$ that best fits the damping vs. tension plot that appears in Figure 12 of *Kirsch et al., 1994* at a nominal tension of 5 N is 0.0198 Ns/mm. The tendon damping model we have just constructed develops 0.697 Ns/mm at a nominal load of 5 N. Using *Equation 34*, we arrive at $\beta_{\mathrm{AT}}^{\mathrm{M}} = 0.020$ Ns/mm. Due to the pennation model, the stiffness and damping values of the CE differ from those along the tendon.

The stiffness of the CE along the tendon is

$$k_{\mathrm{AT}}^{\mathrm{M}} = \left( \frac{\partial f_{\mathrm{AT}}^{\mathrm{M}}}{\partial \ell^{\mathrm{M}}} \right) \left( \frac{\partial \ell^{\mathrm{M}}}{\partial \ell_{\mathrm{AT}}^{\mathrm{M}}} \right) \tag{35}$$

which can be expanded to

$$k_{\mathrm{AT}}^{\mathrm{M}} = \left( \left( \frac{\partial f^{\mathrm{M}}}{\partial \ell^{\mathrm{M}}} \right) \cos \alpha - f^{\mathrm{M}} \sin \alpha \left( \frac{\partial \alpha}{\partial \ell^{\mathrm{M}}} \right) \right) \frac{\partial \ell^{\mathrm{M}}}{\partial \ell_{\mathrm{AT}}^{\mathrm{M}}}. \tag{36}$$

Since we are using a constant thickness pennation model

$$\alpha = \arcsin \left( \frac{\mathrm{H}}{\ell^{\mathrm{M}}} \right) \tag{37}$$

and thus

$$\frac{\partial \alpha}{\partial \ell^{\mathrm{M}}} = \frac{1}{\sqrt{1 - \left( \mathrm{H}/\ell^{\mathrm{M}} \right)^2}} \left( \frac{-\mathrm{H}}{\left( \ell^{\mathrm{M}} \right)^2} \right) \tag{38}$$

which simplifies to

$$\frac{\partial \alpha}{\partial \ell^{\mathrm{M}}} = \frac{-H}{\left(\ell^{\mathrm{M}}\right)^2 \cos \alpha}. \tag{39}$$

Similarly, the constant thickness pennation model means that

$$\ell_{\mathrm{AT}}^{\mathrm{M}} = \ell^{\mathrm{M}} \cos \alpha \tag{40}$$

which leads to

$$\frac{\partial \ell^{\mathrm{M}}}{\partial \ell_{\mathrm{AT}}^{\mathrm{M}}} = \frac{1}{\cos \alpha} \tag{41}$$

Recognizing that

$$k^{\mathrm{M}} = \left( \frac{\partial f^{\mathrm{M}}}{\partial \ell^{\mathrm{M}}} \right) \tag{42}$$

we can solve for $k^{\mathrm{M}}$ in terms of $k_{\mathrm{AT}}^{\mathrm{M}}$ by solving **Equation 36** for $k^{\mathrm{M}}$ and substituting the values of **Equation 39**, and **Equation 41**. In this case, the values of $k^{\mathrm{M}}$ (4.37 N/mm) and $k_{\mathrm{AT}}^{\mathrm{M}}$ (4.37 N/mm) are the same to three significant figures.

We can use a similar process to transform $\beta_{\mathrm{AT}}^{\mathrm{M}}$ into $\beta^{\mathrm{M}}$ using the pennation model by noting that

$$\beta_{\mathrm{AT}}^{\mathrm{M}} = \left( \frac{\partial f_{\mathrm{AT}}^{\mathrm{M}}}{\partial v^{\mathrm{M}}} \right) \left( \frac{\partial v^{\mathrm{M}}}{\partial v_{\mathrm{AT}}^{\mathrm{M}}} \right) \tag{43}$$

which expands to a much smaller expression

$$\beta_{\mathrm{AT}}^{\mathrm{M}} = \left( \left( \frac{\partial f^{\mathrm{M}}}{\partial v^{\mathrm{M}}} \right) \cos \alpha \right) \left( \frac{\partial v^{\mathrm{M}}}{\partial v_{\mathrm{AT}}^{\mathrm{M}}} \right). \tag{44}$$

than **Equation 36** since $\alpha$ does not depend on $v^{\mathrm{M}}$, and thus $\partial \alpha / \partial v^{\mathrm{M}} = 0$. By taking a time derivative of **Equation 40**, we arrive at

$$v_{\mathrm{AT}}^{\mathrm{M}} = v^{\mathrm{M}} \cos \alpha - \ell^{\mathrm{M}} \sin \alpha \left( \frac{\partial \alpha}{\partial \ell^{\mathrm{M}}} \right) v^{\mathrm{M}} \tag{45}$$

which allows us to solve for

$$\frac{\partial v^{\mathrm{M}}}{\partial v_{\mathrm{AT}}^{\mathrm{M}}} = \frac{1}{\cos \alpha - \ell^{\mathrm{M}} \sin \alpha \left( \partial \alpha / \partial \ell^{\mathrm{M}} \right)} \tag{46}$$

By recognizing that

$$\beta^{\mathrm{M}} = \frac{\partial f^{\mathrm{M}}}{\partial v^{\mathrm{M}}} \tag{47}$$

and using **Equation 44** and **Equation 46** we can evaluate $\beta^{\mathrm{M}}$ in terms of $\beta_{\mathrm{AT}}^{\mathrm{M}}$. Similar to $k^{\mathrm{M}}$, the value of $\beta^{\mathrm{M}}$ (0.020 Ns/mm) is close to $\beta_{\mathrm{AT}}^{\mathrm{M}}$ (0.020 Ns/mm). When this same procedure is applied to the stiffness and damping coefficients extracted from the gain and phase profiles from Figure 3 of **Kirsch et al., 1994**, the values of $k^{\mathrm{M}}$ and $\beta^{\mathrm{M}}$ differ (4.37 N/mm and 0.0090 Ns/mm) from the results produced using the data of **Figure 12** (2.90 N/mm and 0.020 Ns/mm). Likely these differences arise because we have been forced to use a fixed maximum isometric force for all specimens when, in reality, this property varies substantially. We now turn our attention to fitting the titin and ECM elements, since we cannot determine how much of $k^{\mathrm{M}}$ and $\beta^{\mathrm{M}}$ are due to the XE until the titin and ECM elements have been fitted.

## 2.4. Fitting the force-length curves of titin's segments

The nonlinear force-length curves used to describe titin ($\mathbf{f}^1(\tilde{\ell}^1)$ and $\mathbf{f}^2(\tilde{\ell}^2)$ in series), and the ECM ($\mathbf{f}^{\mathrm{ECM}}(\tilde{\ell}^{\mathrm{ECM}})$) must satisfy three conditions: the total force-length curve produced by the sum of the

ECM and titin must match the observed passive-force-length relation (*Herzog and Leonard, 2002*); the proportion of titin's contribution relative to the ECM must be within measured bounds (*Prado et al., 2005*); and finally the stiffness of the $\mathbf{f}^2(\tilde{\ell}^2)$ must be a linear scaling of $\mathbf{f}^1(\tilde{\ell}^1)$ to match the observations of *Trombitás et al., 1998b*.

First, we fit the passive force-length curve to the data of *Herzog and Leonard, 2002* to serve as a reference. The curve $\mathbf{f}^{PE}$ begins at the normalized length and force coordinates of $(1 + e_0^{PE}, 0)$ with a slope of 0, ends at $(1 + e_1^{PE}, 1.0)$ with a slope of $k_1^{PE} = 2/(e_1^{PE} - e_0^{PE})$, and is linearly extrapolated outside of this region. We solve for the $e_0^{PE}$ and $e_1^{PE}$ such that

$$\min \sum_i^n \left( \mathbf{f}^{PE}\left( \ell_i^{PE}/\ell_o^M \right) - f_i^{PE}/f_o^M \right)^2 \tag{48}$$

the squared differences between $\mathbf{f}^{PE}$ and the passive force-length data of *Herzog and Leonard, 2002* (*Figure 2A* shows both the data and the fitted $\mathbf{f}^{PE}$ curve) are minimized. While $\mathbf{f}^{PE}$ is not used directly in the model, it serves as a useful reference for constructing the ECM and titin force-length curves. We assume that the ECM force-length curve is a linear scaling of $\mathbf{f}^{PE}$

$$\mathbf{f}^{ECM}(\tilde{\ell}^{ECM}) = P\mathbf{f}^{PE}(\tilde{\ell}^M). \tag{49}$$

where P is a constant. In this work, we set P to 56% which is the average ECM contribution that *Prado et al., 2005* measured across 5 different rabbit skeletal muscles (see Appendix 9, Note 14). The remaining fraction, $1 - P$, of the force-length curve comes from titin.

In mammalian skeletal muscle, titin has three elastic segments (*Prado et al., 2005*) connected in series: the proximal Ig segment, the PEVK segment, and the distal Ig segment that is between the PEVK region and the myosin filament (*Figure 1A*). *Trombitás et al., 1998b* labelled the PEVK region of titin with antibodies allowing them to measure the distance between the Z-line and the proximal Ig/PEVK boundary ($^Z\ell^{IgP/PEVK}$), and the distance between the Z-line and the PEVK/distal Ig boundary ($^Z\ell^{PEVK/IgD}$), while the passive sarcomere was stretched from 2.35–4.46 μm. By fitting functions to the data of *Trombitás et al., 1998b*, we can predict the length of any of titin's segments under the following assumptions: the T12 segment is rigid (*Figure 1A*), the distal Ig segment that overlaps with myosin is rigid (*Figure 1A*), and that during passive stretching the tension throughout the titin filament is uniform. Since the sarcomeres in the experiments of *Trombitás et al., 1998b* were passively stretched it is reasonable to assume that tension throughout the free part of the titin filament is uniform because the bond between titin and actin depends on calcium (*Dutta et al., 2018*; *Kellermayer and Granzier, 1996*) and crossbridge attachment (*Leonard et al., 2010*).

We begin by digitizing the data of Figure 5 of *Trombitás et al., 1998b* and using the least-squares method to fit lines to $^Z\ell^{IgP/PEVK}$ and $^Z\ell^{PEVK/IgD}$ (where the superscripts mean $^{from}\ell^{to}$ and so $^Z\ell^{IgP/PEVK}$ is the distance from the Z-line to the border of the IgP/PEVK segments). From these lines of best fit we can evaluate the normalized length of the proximal Ig segment

$$\tilde{\ell}^{IgP} = \left( ^Z\ell^{IgP/PEVK} - L^{T12} \right) / \ell_o^M, \tag{50}$$

the normalized length of the PEVK segment

$$\tilde{\ell}^{PEVK} = \left( ^Z\ell^{PEVK/IgD} - ^Z\ell^{IgP/PEVK} \right) / \ell_o^M, \tag{51}$$

and the normalized length of the distal Ig segment

$$\tilde{\ell}^{IgD} = \left( \frac{1}{2}\ell^M - ^Z\ell^{PEVK/IgD} \right) / \ell_o^M \tag{52}$$

as a function of sarcomere length. Next, we extract the coefficients for linear functions that evaluate the lengths of

$$\tilde{\ell}^{IgP}(\tilde{\ell}^M) = A^{IgP}\tilde{\ell}^M + b^{IgP}, \tag{53}$$

$$\tilde{\ell}^{PEVK}(\tilde{\ell}^M) = A^{PEVK}\tilde{\ell}^M + b^{PEVK}, \tag{54}$$

and

$$\tilde{\ell}^{\mathrm{IgD}}(\tilde{\ell}^{\mathrm{M}}) = \mathrm{A}^{\mathrm{IgD}}\tilde{\ell}^{\mathrm{M}} + \mathrm{b}^{\mathrm{IgD}} \tag{55}$$

given $\tilde{\ell}^{\mathrm{M}}$. The coefficients that best fit the data from *Trombitás et al., 1998b* appear in *Appendix 2—table 1*.

These functions can be scaled to fit a titin filament of a differing geometry. Many of the experiments simulated in this work used cat soleus. Although the lengths of titin filament segments in cat soleus have not been measured, we assume that it is a scaled version of a human soleus titin filament (68 proximal Ig domains, 2174 PEVK residues, and 22 distal Ig domains *Trombitás et al., 1998b*) since both muscles contain predominately slow-twitch fibers: slow twitch fibers tend to express longer, more compliant titin filaments (*Prado et al., 2005*). Since the optimal sarcomere length in cat skeletal muscle is shorter than in human skeletal muscle (2.43 µm vs. 2.73 µm from *Rassier et al., 1999*) the coefficients for *Equations 53–55* differ slightly (see the feline soleus column in *Appendix 2—table 1*). In addition, by assuming that the titin filament of cat skeletal muscle is a scaled version of the titin filament found in human skeletal muscle, we have implicitly assumed that the cat's skeletal muscle titin filament has 60.5 proximal Ig domains, 1934.7 PEVK residues, and 19.6 distal Ig domains. Although a fraction of a domain does not make physical sense, we have not rounded to the nearest domain and residue to preserve the sarcomere length-based scaling.

In contrast, the rabbit psoas fibril used in the simulation of *Leonard et al., 2010* has a known titin geometry (50 proximal Ig domains, 800 PEVK residues, and 22 distal Ig domains as measured by *Prado et al., 2005*) which differs substantially from the isoform of titin expressed in the human soleus. To create the rabbit psoas titin length functions $\tilde{\ell}_{\mathrm{R}}^{\mathrm{IgP}}(\tilde{\ell}^{\mathrm{M}})$, $\tilde{\ell}_{\mathrm{R}}^{\mathrm{PEVK}}(\tilde{\ell}^{\mathrm{M}})$, and $\tilde{\ell}_{\mathrm{R}}^{\mathrm{IgD}}(\tilde{\ell}^{\mathrm{M}})$, we begin by scaling the human soleus PEVK length function $\tilde{\ell}_{\mathrm{H}}^{\mathrm{PEVK}}(\tilde{\ell}^{\mathrm{M}})$ by the relative proportion of PEVK residues of 800/2174. The length of the two Ig segments

$$\tilde{\ell}_{\mathrm{R}}^{\mathrm{Ig}}(\tilde{\ell}^{\mathrm{M}}) = \frac{1}{2}\tilde{\ell}^{\mathrm{M}} - \tilde{\mathrm{L}}^{\mathrm{T12}} - \tilde{\mathrm{L}}^{\mathrm{IgD}} - \left(800/2174\right)\tilde{\ell}_{\mathrm{H}}^{\mathrm{PEVK}}(\tilde{\ell}^{\mathrm{M}}) \tag{56}$$

is what remains from the half-sarcomere once the rigid lengths of titin (0.100 µm for $\mathrm{L}^{\mathrm{T12}}$ and 0.8150 µm for $\mathrm{L}^{\mathrm{IgD}}$ *Higuchi et al., 1995*) and the PEVK segment length have been subtracted away. The function that describes $\tilde{\ell}_{\mathrm{R}}^{\mathrm{IgP}}(\tilde{\ell}^{\mathrm{M}})$ and $\tilde{\ell}_{\mathrm{R}}^{\mathrm{IgD}}(\tilde{\ell}^{\mathrm{M}})$ can then be formed by scaling $\tilde{\ell}_{\mathrm{R}}^{\mathrm{Ig}}(\tilde{\ell}^{\mathrm{M}})$ by the proportion of Ig domains in each segment

$$\tilde{\ell}_{\mathrm{R}}^{\mathrm{IgP}}(\tilde{\ell}^{\mathrm{M}}) = \left(\frac{50}{50+22}\right)\tilde{\ell}_{\mathrm{R}}^{\mathrm{Ig}}(\tilde{\ell}^{\mathrm{M}}), \text{and} \tag{57}$$

$$\tilde{\ell}_{\mathrm{R}}^{\mathrm{IgD}}(\tilde{\ell}^{\mathrm{M}}) = \left(\frac{22}{50+22}\right)\tilde{\ell}_{\mathrm{R}}^{\mathrm{Ig}}(\tilde{\ell}^{\mathrm{M}}) \tag{58}$$

which produce the coefficients that appear in the rabbit psoas column in *Appendix 2—table 1*. While we have applied this approach to extend the results of *Trombitás et al., 1998b* to a rabbit psoas, in principle this approach can be applied to any isoform of titin provided that its geometry is known, and the Ig domains and PEVK residues in the target titin behave similarly to those in human soleus titin.

**Appendix 2—table 1.** The relations that define how titin's segments lengthen as a sarcomere is passively stretched.

The coefficients of the normalized lengths of $\tilde{\ell}^{\mathrm{IgP}}(\tilde{\ell}^{\mathrm{M}})$, $\tilde{\ell}^{\mathrm{PEVK}}(\tilde{\ell}^{\mathrm{M}})$, $\tilde{\ell}^{\mathrm{IgD}}(\tilde{\ell}^{\mathrm{M}})$ and from *Equation 53-Equation 55* under passive lengthening. These coefficients (human soleus column) have been extracted from data of Figure 5 of *Trombitás et al., 1998b* using a least-squares fit. Since Figure 5 of *Trombitás et al., 1998b* plots the change in segment length of a single titin filament against the change in length of the entire sarcomere, the resulting slopes are in length normalized units. The slopes sum to 0.5, by construction, to reflect the fact that these three segments of titin stretch at half the rate of the entire sarcomere (assuming symmetry). The cat soleus titin segment coefficients have been formed using a simple scaling of the human soleus titin segment coefficients, and so, are similar. Rabbit psoas titin geometry (*Prado et al., 2005*) differs dramatically from human soleus titin (*Trombitás et al., 1998b*) and produce a correspondingly large difference in the coefficients that

describe the length of the segments of rabbit psoas titin.

| Coefficient | Human | Feline | Rabbit |
|---|---|---|---|
| | Soleus *Trombitás et al., 1998a* | Soleus | Psoas |
| $A^{IgP}$ | 0.177 | 0.177 | 0.262 |
| $b^{IgP}$ | –0.101 | –0.113 | –0.189 |
| $A^{PEVK}$ | 0.266 | 0.266 | 0.122 |
| $b^{PEVK}$ | –0.197 | –0.221 | –0.100 |
| $A^{IgD}$ | 0.057 | 0.057 | 0.115 |
| $b^{IgD}$ | –0.033 | –0.033 | –0.083 |

The only detail that remains is to establish the shape of the IgP, PEVK, and IgD force-length curves. Studies of individual titin filaments, and of its segments, make it clear that titin is mechanically complex. For example, the tandem Ig segments (the IgD and IgP segments) are composed of many folded domains (titin from human soleus has two Ig segments that together have nearly 100 domains *Trombitás et al., 1998b*). Each domain appears to be a simple nonlinear spring until it unfolds and elongates by nearly 25 nm in the process (*DuVall et al., 2013*). Unfolding events appear to happen individually during lengthening experiments (*DuVall et al., 2013*), with each unfolding event occurring at a slightly greater tension than the last, giving an Ig segment a force-length curve that is saw-toothed. Although detailed models of titin exist that can simulate the folding and unfolding of individual Ig domains, this level of detail comes at a cost of a state for each Ig domain which can add up to nearly a hundred extra states (*Schappacher-Tilp et al., 2015*) in total.

Active and passive lengthening experiments at the sarcomere-level hide the complexity that is apparent when studying individual titin filaments. The experiments of *Leonard et al., 2010* show that the sarcomeres in a filament (from rabbit psoas) mechanically fail when stretched passively to an average length of $2.86\ell_o^M$, but can reach $3.38\ell_o^M$ when actively lengthened. *Leonard et al., 2010* showed that titin was the filament bearing these large forces since the sarcomeres were incapable of developing active or passive tension when the experiment was repeated after the titin filaments were chemically cut. It is worth noting that the forces measured by *Leonard et al., 2010* contain none of the complex saw-tooth pattern indicative of unfolding events even though 72 of these events would occur as each proximal and distal Ig domain fully unfolded and reached its maximal length (see Appendix 9, Note 15). Although we cannot be sure how many unfolding events occurred during the experiments of *Leonard et al., 2010* due to sarcomere non-homogeneity (*Johnston et al., 2016*), it is likely that many Ig unfolding events occurred since the average sarcomere length at failure $3.38\ell_o^M$ was longer than the maximum length of $2.4$–$2.7$ $\ell_o^M$ that would be predicted from the geometry of rabbit psoas titin (see Appendix 9, Note 16).

Since we are interested in a computationally efficient model that is accurate at the whole muscle level, we model titin as a multi-segmented nonlinear spring but omit the states needed to simulate the folding and unfolding of Ig domains. Simulations of active lengthening using our titin model will exhibit the enhanced force development that appears in experiments (*Herzog and Leonard, 2002*; *Leonard et al., 2010*), but will lack the nonlinear saw-tooth force-length profile that is measured when individual titin filaments are lengthened (*DuVall et al., 2013*). To have the broadest possible application, we will fit titin's force-length curves to provide reasonable results for both moderate (*Herzog and Leonard, 2002*) and large active stretches (*Leonard et al., 2010*). Depending on the application, it may be more appropriate to use a stiffer force-length curve for the Ig segment if normalized sarcomere lengths stays within $1.5$ $\ell_o^M$ and no unfolding events occur as was done by *Trombitás et al., 1998b*.

To ensure that the serially connected force-length curves of $\mathbf{f}^1(\tilde{\ell}^1)$ and $\mathbf{f}^2(\tilde{\ell}^2)$ closely reproduce $(1 - P)\mathbf{f}^{PE}(\tilde{\ell}^M)$, we are going to use affine transformations of $\mathbf{f}^{PE}$ to describe $\mathbf{f}^1(\tilde{\ell}^1)$ and $\mathbf{f}^2(\tilde{\ell}^2)$. The total stiffness of the half-sarcomere titin model is given by

$$\tilde{k}^{\mathrm{Ti}} = 2\left(1 - \mathrm{P}\right)\frac{\partial \mathbf{f}^{\mathrm{PE}}}{\partial \tilde{\ell}^{\mathrm{M}}} \tag{59}$$

which is formed by the series connection of $\mathbf{f}^1(\tilde{\ell}^1)$ and $\mathbf{f}^2(\tilde{\ell}^2)$

$$\frac{1}{\tilde{k}^{\mathrm{Ti}}} = \frac{1}{\tilde{k}^1} + \frac{1}{\tilde{k}^2}. \tag{60}$$

Since each of titin's segments is exposed to the same tension in the experiment of *Trombitás et al., 1998b*, the slopes of the lines that *Equations 53–55* describe are directly proportional to the relative compliance (inverse of stiffness) between of each of titin's segments. Using this fact, we can define the normalized stretch rates of the proximal titin segment

$$\mathrm{C}^1 = \mathrm{A}^{\mathrm{IgP}} + \mathrm{A}^{\mathrm{PEVK}}\mathrm{Q} = \frac{\Delta\ell^{\mathrm{PEVK}}\mathrm{Q} + \Delta\ell^{\mathrm{IgP}}}{\Delta\ell^{\mathrm{M}}} \tag{61}$$

and the distal titin segment

$$\mathrm{C}^2 = \mathrm{A}^{\mathrm{PEVK}}\left(1 - \mathrm{Q}\right) + \mathrm{A}^{\mathrm{IgD}} = \frac{\Delta\ell^{\mathrm{PEVK}}\left(1 - \mathrm{Q}\right) + \Delta\ell^{\mathrm{IgD}}}{\Delta\ell^{\mathrm{M}}} \tag{62}$$

which are proportional to the compliance of two titin segments in our model. When both the $\mathbf{f}^1(\tilde{\ell}^1)$ and $\mathbf{f}^2(\tilde{\ell}^2)$ curves are beyond the toe region the stiffness is a constant and is given by

$$\tilde{k}^1_{\mathrm{toe}} = \frac{\Delta\tilde{f}}{\Delta\tilde{\ell}^1} \tag{63}$$

and

$$\tilde{k}^2_{\mathrm{toe}} = \frac{\Delta\tilde{f}}{\Delta\tilde{\ell}^2}. \tag{64}$$

Dividing *Equation 63* by *Equation 64* eliminates the unknown $\Delta f$ and results in an expression that relates the ratio of the terminal linear stiffness of $\mathbf{f}^1(\tilde{\ell}^1)$ and $\mathbf{f}^2(\tilde{\ell}^2)$

$$\frac{\tilde{k}^1_{\mathrm{toe}}}{\tilde{k}^2_{\mathrm{toe}}} = \frac{\Delta\tilde{\ell}^2}{\Delta\tilde{\ell}^1} = \frac{\mathrm{C}^2}{\mathrm{C}^1} \tag{65}$$

to the relative changes in *Equation 61* and *Equation 62*. Substituting *Equation 65*, and *Equation 60* into *Equation 59* yields the expression

$$\frac{1}{\tilde{k}^2_{\mathrm{toe}}\left(\mathrm{C}^2/\mathrm{C}^1\right)} + \frac{1}{\tilde{k}^2_{\mathrm{toe}}} = \frac{1}{\tilde{k}^{\mathrm{Ti}*}} \tag{66}$$

which can be simplified to

$$\tilde{k}^2_{\mathrm{toe}} = \left(\frac{\mathrm{C}^1 + \mathrm{C}^2}{\mathrm{C}^2}\right)\tilde{k}^{\mathrm{Ti}*} \tag{67}$$

and this expression can be evaluated using the terminal stiffness of titin $\tilde{k}^{\mathrm{Ti}*}$ and the coefficients listed in *Appendix 2—table 1*. Substituting *Equation 67* into *Equation 65* yields

$$\tilde{k}^1_{\mathrm{toe}} = \left(\frac{\mathrm{C}^1 + \mathrm{C}^2}{\mathrm{C}^1}\right)\tilde{k}^{\mathrm{Ti}*}. \tag{68}$$

The curves $\mathbf{f}^1(\tilde{\ell}^1)$ and $\mathbf{f}^2(\tilde{\ell}^2)$ can now be formed by scaling and shifting the total force-length curve of titin $(1 - \mathrm{P})\mathbf{f}^{\mathrm{PE}}$. By construction, titin's force-length curve develops a tension of $(1 - \mathrm{P})$, and has reached its terminal stiffness, when the CE reaches a length $\tilde{\ell}^{\mathrm{M}*}$ such that $\mathbf{f}^{\mathrm{PE}}(\tilde{\ell}^{\mathrm{M}*}) = 1$. Using *Equation 53-55* and the appropriate coefficients in *Appendix 2—table 1*, we can evaluate the normalized length developed by the $\ell^1$ segment

$$\tilde{\ell}_{\text{toe}}^1 = \tilde{\ell}^{\text{IgP}}(\tilde{\ell}^{\text{M*}}) + Q\tilde{\ell}^{\text{PEVK}}(\tilde{\ell}^{\text{M*}}) \tag{69}$$

and $\ell^2$ segment

$$\tilde{\ell}_{\text{toe}}^2 = \left(1 - Q\right)\tilde{\ell}^{\text{PEVK}}(\tilde{\ell}^{\text{M*}}) + \tilde{\ell}^{\text{IgD}}(\tilde{\ell}^{\text{M*}}) \tag{70}$$

at a CE length of $\tilde{\ell}^{\text{M*}}$. The $\mathbf{f}^1(\tilde{\ell}^1)$ curve is formed by shifting and scaling the $(1 - P)\mathbf{f}^{\text{PE}}$ curve so that it develops a normalized tension of $(1 - P)$ and a stiffness of $\tilde{k}_{\text{toe}}^1$ at a length of $\tilde{\ell}_{\text{toe}}^1$. Similarly, the $\mathbf{f}^2(\tilde{\ell}^2)$ curve is made by shifting and scaling the $(1 - P)\mathbf{f}^{\text{PE}}$ curve to develop a normalized tension of $(1 - P)$ and a stiffness of $\tilde{k}_{\text{toe}}^2$ at a length of $\tilde{\ell}_{\text{toe}}^2$.

By construction, the spring network formed by the $\mathbf{f}^{\text{ECM}}(\tilde{\ell}^{\text{ECM}})$, $\mathbf{f}^1(\tilde{\ell}^1)$, and $\mathbf{f}^2(\tilde{\ell}^2)$ curves follows the fitted $\mathbf{f}^{\text{PE}}$ curve (*Figure 3A*) such that the ECM curve makes up 56% of the contribution. When the CE is active and $\tilde{\ell}^1$ is effectively fixed in place, the distal segment of titin contributes higher forces since $\tilde{\ell}^2$ undergoes higher strains (*Figure 3A*). Finally, when the experiment of *Trombitás et al., 1998b* are simulated the movements of the IgP/PEVK and PEVK/IgD boundaries in the titin model closely follow the data (*Figure 3C*).

The process we have used to fit the ECM and titin's segments makes use of data within modest normalized CE lengths (2.35–4.46 µm, or 0.86–1.63 $\ell_o^{\text{M}}$ *Trombitás et al., 1998b*). Scenarios in which the CE reaches extremely long lengths, such as during injury or during the experiment of *Leonard et al., 2010*, require fitting titin's force-length curve beyond the typical ranges observed in-vivo. The WLC model has been used successfully to model the force-length relation of individual titin segments (*Trombitás et al., 1998b*) at extreme lengths. In this work, we consider two different extensions to $\mathbf{f}^1(\tilde{\ell}^1)$ and $\mathbf{f}^2(\tilde{\ell}^2)$: a linear extrapolation, and the WLC model. Since the fitted $\mathbf{f}^{\text{PE}}$ curve is linearly extrapolated, so too are the $\mathbf{f}^{\text{ECM}}(\tilde{\ell}^{\text{ECM}})$, $\mathbf{f}^1(\tilde{\ell}^1)$, and $\mathbf{f}^2(\tilde{\ell}^2)$ curves by default. Applying the WLC to our titin curves requires a bit more effort.

We have modified the WLC to include a slack length $\tilde{\text{L}}_{\text{S}}^{\text{W}}$ (the superscript W means WLC) so that the WLC model can made to be continuous with $\mathbf{f}^1(\tilde{\ell}^1)$ and $\mathbf{f}^2(\tilde{\ell}^2)$. The normalized force developed by our WLC model is given by

$$\mathbf{f}^{\text{W}} = \begin{cases} B\left(\tilde{\ell}^{\text{W}} + \dfrac{1}{4(1 - \tilde{\ell}^{\text{W}})^2} - \dfrac{1}{4}\right) & \tilde{\ell}^{\text{W}} > 0 \\ 0 & \text{otherwise} \end{cases} \tag{71}$$

where B is a scaling factor and the normalized segment length $\tilde{\ell}^{\text{W}}$ is defined as

$$\tilde{\ell}^{\text{W}} = \frac{\ell^{\text{W}} - \tilde{\text{L}}_{\text{S}}^{\text{W}}}{\text{L}_{\text{C}}^{\text{W}} - \tilde{\text{L}}_{\text{S}}^{\text{W}}} \tag{72}$$

where $\tilde{\text{L}}_{\text{S}}^{\text{W}}$ is the slack length, and $\text{L}_{\text{C}}^{\text{W}}$ is the contour length of the segment. To extend the $\mathbf{f}^1(\tilde{\ell}^1)$ curve to follow the WLC model, we first note the normalized contour length of the $\ell^1$ segment

$$\tilde{\text{L}}_{\text{C}}^{1W} = \frac{\text{N}^{\text{IgP}}25\text{nm} + Q\text{N}^{\text{PEVK}}0.38\text{nm}}{\ell_o^{\text{M}}} \tag{73}$$

by counting the number of proximal Ig domains ($\text{N}^{\text{IgP}}$), the number of PEVK residues ($Q\text{N}^{\text{PEVK}}$) associated with $\ell^1$ and by scaling each by the maximum contour length of each Ig domain (25 nm *DuVall et al., 2013*), and each PEVK residue (between 0.32 *Trombitás et al., 1998b* and 0.38 nm *Cantor and Schimmel, 1980* see pg. 254). This contour length defines the maximum length of the segment, when all of the Ig domains and PEVK residues have been unfolded. Similarly, the contour length of $\tilde{\text{L}}_{\text{C}}^{2W}$ is given by

$$\tilde{\text{L}}_{\text{C}}^{2W} = \frac{\text{N}^{\text{IgD}}25\text{nm} + \left(1 - Q\right)\text{N}^{\text{PEVK}}0.38\text{nm}}{\ell_o^{\text{M}}}. \tag{74}$$

Next, we define the slack length by linearly extrapolating backwards from the final fitted force $(1 - P)$

$$\tilde{L}_{\rm S}^{1W} = \frac{(1-{\rm P})}{\tilde{k}_{\rm toe}^1}, \tag{75}$$

and similarly

$$\tilde{L}_{\rm S}^{2W} = \frac{(1-{\rm P})}{\tilde{k}_{\rm toe}^2}. \tag{76}$$

We can now solve for B in **Equation 71** so that $\mathbf{f}^1(\tilde{\ell}^1)$ and $\mathbf{f}^2(\tilde{\ell}^2)$ are continuous with each respective WLC extrapolation. However, we do not use the WLC model directly because it contains a singularity which is problematic during numerical simulation. Instead, we add an additional Bézier segment to fit the WLC extension that spans between forces of $(1-{\rm P})$ and twice the normalized failure force $(2 \times 5.14 f_{\rm o}^{\rm M})$ noted by **Leonard et al., 2010**. To fit the shape of the final Bézier segment, we adjust the locations of the internal control points to minimize the squared differences between the modified WLC model and the final Bézier curve (**Figure 10A**). The final result is a set of curves ($\mathbf{f}^1(\tilde{\ell}^1), \mathbf{f}^2(\tilde{\ell}^2)$, and $\mathbf{f}^{\rm ECM}(\tilde{\ell}^{\rm ECM})$) which, between forces 0 and $(1-{\rm P})$, will reproduce $\mathbf{f}^{\rm PE}$, the measurements of **Trombitás et al., 1998b**, and do so with titin-ECM balance similar to the measurements of **Prado et al., 2005**. For forces beyond $(1-{\rm P})$, the curve will follow the segment-specific WLC model up to twice the expected failure tension noted by **Leonard et al., 2010**.

## 2.5. Fitting the XE's impedance

With the passive curves established, we can return to the problem of identifying the normalized maximum stiffness $\tilde{k}_{\rm o}^{\rm X}$ and damping $\tilde{\beta}_{\rm o}^{\rm X}$ of the lumped XE element. Just prior to discussing titin, we had evaluated the impedance of the cat soleus CE from Figure 12 of **Kirsch et al., 1994** to be $k^{\rm M} = 2.90$ N/mm and $\beta^{\rm M} = 0.020$ Ns/mm at a nominal active tension of 5 N. The normalized stiffness $k^{\rm M}$ can be found by taking the partial derivative of **Equation 15** with respect to $\tilde{\ell}^{\rm M}$

$$k^{\rm M} = a \frac{\partial \mathbf{f}^{\rm L}(\tilde{\ell}^{\rm S} + \tilde{\rm L}^{\rm M})}{\partial \tilde{\ell}^{\rm M}} (\tilde{k}_{\rm o}^{\rm X} \tilde{\ell}^{\rm X} + \tilde{\beta}_{\rm o}^{\rm X} \tilde{v}^{\rm X}) + a\mathbf{f}^{\rm L}(\tilde{\ell}^{\rm S} + \tilde{\rm L}^{\rm M}) \left( \tilde{k}_{\rm o}^{\rm X} \frac{\partial \tilde{\ell}^{\rm X}}{\partial \tilde{\ell}^{\rm M}} \right) + \frac{\partial \mathbf{f}^2(\tilde{\ell}^2)}{\partial \tilde{\ell}^{\rm M}} + \frac{\partial \mathbf{f}^{\rm ECM}(\tilde{\ell}^{\rm ECM})}{\partial \tilde{\ell}^{\rm M}}. \tag{77}$$

By noting that all of our chosen state variables in **Equation 13** are independent and by making use of the kinematic relationships in **Equation 9** and **Equation 10** we can reduce **Equation 77** to

$$k^{\rm M} = \frac{a \tilde{k}_{\rm o}^{\rm X} \mathbf{f}^{\rm L}(\tilde{\ell}^{\rm S} + \tilde{\rm L}^{\rm M})}{2} + \frac{\mathrm{d}\mathbf{f}^2(\tilde{\ell}^2)}{\mathrm{d}\ell^2} \frac{1}{2} + \frac{\mathrm{d}\mathbf{f}^{\rm ECM}(\tilde{\ell}^{\rm ECM})}{\mathrm{d}\ell^{\rm ECM}} \frac{1}{2} \tag{78}$$

and solve for $\tilde{k}_{\rm o}^{\rm X}$

$$\tilde{k}_{\rm o}^{\rm X} = \frac{2}{a\mathbf{f}^{\rm L}(\tilde{\ell}^{\rm S} + \tilde{\rm L}^{\rm M})} \left( k^{\rm M} - \frac{\mathrm{d}\mathbf{f}^2(\tilde{\ell}^2)}{\mathrm{d}\ell^2} \frac{1}{2} - \frac{\mathrm{d}\mathbf{f}^{\rm ECM}(\tilde{\ell}^{\rm ECM})}{\mathrm{d}\ell^{\rm ECM}} \frac{1}{2} \right) \tag{79}$$

When using to the data from Figure 12 in **Kirsch et al., 1994**, we end up with $\tilde{k}_{\rm o}^{\rm X} = 49.1 f_{\rm o}^{\rm M}/\ell_{\rm o}^{\rm M}$ for the elastic-tendon model, and $\tilde{k}_{\rm o}^{\rm X} = 41.8 f_{\rm o}^{\rm M}/\ell_{\rm o}^{\rm M}$ for the rigid-tendon model. When this procedure is repeated for Figure 3 of **Kirsch et al., 1994** (from a different specimen), we are left with $\tilde{k}_{\rm o}^{\rm X} = 74.5 f_{\rm o}^{\rm M}/\ell_{\rm o}^{\rm M}$ for the elastic-tendon model and $\tilde{k}_{\rm o}^{\rm X} = 59.1 f_{\rm o}^{\rm M}/\ell_{\rm o}^{\rm M}$ for the rigid-tendon model. The value for $\tilde{k}_{\rm o}^{\rm X}$ is much larger than $k^{\rm M}$ because the $a$ needed to generate 5 N is only 0.231. Similarly, we can form the expression for the normalized damping of the CE by taking the partial derivative of **Equation 15** with respect to $\tilde{v}^{\rm M}$

$$\tilde{\beta}^{\rm M} = a\mathbf{f}^{\rm L}(\tilde{\ell}^{\rm S} + \tilde{\rm L}^{\rm M}) \left( \tilde{\beta}_{\rm o}^{\rm X} \frac{\mathrm{d}\tilde{v}^{\rm X}}{\mathrm{d}\tilde{v}^{\rm M}} \right) + \tilde{\beta}^{\epsilon}. \tag{80}$$

As with $k^{\rm M}$, the expression for $\tilde{\beta}^{\rm M}$ can be reduced to

$$\tilde{\beta}_{\rm o}^{\rm X} = \frac{2}{a\mathbf{f}^{\rm L}(\tilde{\ell}^{\rm S} + \tilde{\rm L}^{\rm M})} \left( \tilde{\beta}^{\rm M} - \tilde{\beta}^{\epsilon} \right) \tag{81}$$

which evaluates to $\tilde{\beta}_o^X = 0.347 f_o^M/(\ell_o^M/s)$ for both the elastic and rigid-tendon models using the data from Figure 12 of **Kirsch et al., 1994**. The damping coefficients of the elastic and rigid-tendon models is similar because the damping coefficient of the musculotendon is dominated by the damping coefficient of CE. When the data from Figure 3 of **Kirsch et al., 1994** is used, the damping coefficients of the elastic and rigid-tendon models are $\tilde{\beta}_o^X = 0.155 f_o^M/(\ell_o^M/s)$ and $\tilde{\beta}_o^X = 0.153 f_o^M/(\ell_o^M/s)$ respectively.

The dimensionless parameters $\tilde{k}_o^X$ and $\tilde{\beta}_o^X$ can be used to approximate the properties of other MTUs given $f_o^M$ and $\ell_o^M$. The stiffness and damping of the lumped crossbridge element will scale linearly with $f_o^M$ and inversely with $\ell_o^M$ provided the impedance properties of individual crossbridges, and the maximum number of crossbridges attached per sarcomere, is similar between a feline's skeletal muscle sarcomeres and those of the target MTU. This approximation is rough, however, since the values for $\tilde{k}_o^X$ and $\tilde{\beta}_o^X$ (**Appendix 2—table 2**) have a relative error of 41% and 76% when fit to Figure 3 and Figure 12 from **Kirsch et al., 1994**. In addition, when simulated, the stiffness and damping of the LTI system of best fit may differ from $\tilde{k}_o^X$ and $\tilde{\beta}_o^X$ at low frequencies because the movement of the attachment point has been ignored in **Equation 79** and **Equation 81**. This approximation explains why the VEXAT's stiffness profile (**Figure 7A, C**) is below the data of **Kirsch et al., 1994**, despite having used this data to fit the $k^M$ and $\tilde{\beta}^M$ terms in **Equation 79** and **Equation 81**. The accuracy of this approximation, however, improves at higher frequencies (**Figure 6E and F**) because the attachment point's movements become increasingly limited due to the time constant $k^M$ in **Equation 16**. Unfortunately, this is a trade-off due to the formulation of **Equation 16**: the VEXAT model can fit the data of **Kirsch et al., 1994** at low frequencies, or high frequencies, but not both simultaneously.

**Appendix 2—table 2.** Normalized titin and crossbridge parameters fit to data from the literature.

| Symbol | Value | Unit | Source |
|---|---|---|---|
| $\tilde{k}^{Ti}$ | 3.88 | $f_o^M/\ell_o^M$ | *Herzog and Leonard, 2002* |
| | | | *Prado et al., 2005* |
| $\tilde{k}^1$ | 5.17 | $f_o^M/\ell_o^M$ | *Trombitás et al., 1998b* |
| $\tilde{k}^2$ | 8.42 | $f_o^M/\ell_o^M$ | *Trombitás et al., 1998b* |
| $\tilde{k}_o^X$ | 74.5 | $f_o^M/(\ell_o^M)$ | *Kirsch et al., 1994* (**Figure 3**) |
| $\tilde{\beta}_o^X$ | 0.155 | $f_o^M/(\ell_o^M/s)$ | *Kirsch et al., 1994* (**Figure 3**) |
| $\tilde{k}_o^X$ | 49.1 | $f_o^M/(\ell_o^M)$ | *Kirsch et al., 1994* (**Figure 12**) |
| $\tilde{\beta}_o^X$ | 0.347 | $f_o^M/(\ell_o^M/s)$ | *Kirsch et al., 1994* (**Figure 12**) |

# Appendix 3

## Model initialization

Solving for an initial state is challenging since we are given $a$, $\ell^P$, and $v^P$ and must solve for $v^S$, $\ell^S$, and $\ell^1$ for a rigid-tendon model, and additionally $\ell^M$ if an elastic-tendon model is used. The type of solution that we look for is one that produces no force or state transients soon after a simulation begins in which activation and path velocity is well approximated as constant. Our preliminary simulations found that satisfactory solutions were found by iterating over both $\tilde{\ell}^M$ and $\tilde{v}^M$ using a nested bisection search that looks for values which approximately satisfies *Equation 22*, result in small values for $\dot{\tilde{v}}^S$ from *Equation 16*, and begins with balanced forces between the two segment titin model in *Equation 20*.

In the outer loop, we iterate over values of $\tilde{\ell}^M$. Given $a$, $\ell^P$, $v^P$, and a candidate value of $\tilde{\ell}^M$, we can immediately solve for $\alpha$ and $\ell^T$ using the pennation model. We can numerically solve for the value of another state, $\ell^1$, using the kinematic relationship between $\ell^M$ and $\ell^1$ and by assuming that the two titin segments are in a force equilibrium

$$\mathbf{f}^1(\tilde{\ell}^1) - \mathbf{f}^2(\tilde{\ell}^2) = 0. \tag{82}$$

In the inner loop, we iterate over values of $\tilde{v}^M$ between $0$ and $v^P \cos \alpha$ (we ignore solutions in which the sign of $v^M$ and $v^T$ differ) to find the value of $\tilde{v}^M$ that best satisfies *Equation 22*. Prior to evaluating *Equation 22*, we need to set both $\tilde{v}^X$ and $\tilde{\ell}^X$. Here we choose a value for $\tilde{v}^X$ that will ensure that the XE is not producing transient forces

$$\tilde{v}^X = 0 \tag{83}$$

and we use fixed-point iteration to solve for $\tilde{\ell}^X$ such that *Equation 16* evaluates to zero. Now the value of $\dot{\tilde{v}}^S$ can be directly evaluated using the candidate value of $\tilde{v}^M$, the first derivative of *Equation 9*, and the fact that we have set $\tilde{v}^X$ to zero. Finally, the error of this specific combination of $\tilde{\ell}^M$ and $\tilde{v}^M$ is evaluated using *Equation 22*, where the best solution leads to the lowest absolute value for of $\tilde{f}^\epsilon$ in *Equation 22*. If a rigid-tendon model is being initialized the procedure is simpler because the inner loop iterating over $\tilde{v}^M$ is unnecessary: given that $v^P$ and $\tilde{v}^X$ are zero, the velocities $\tilde{v}^M$ and $\tilde{v}^S$ can be directly solved using the first derivative of *Equation 9*. While in principle any root solving method can be used to solve this problem, we have chosen to use the bisection method to avoid local minima.

## Appendix 4

### Evaluating a muscle model's frequency response

To analyze the the frequency response of a muscle to length perturbation we begin by evaluating the length change

$$x(t) = \ell^{\mathrm{MT}} - \hat{\ell}^{\mathrm{MT}} \tag{84}$$

and force change

$$y(t) = f^{\mathrm{T}} - \hat{f}^{\mathrm{T}} \tag{85}$$

with respect to the nominal length ($\hat{\ell}^{\mathrm{MT}}$) and nominal force ($\hat{f}^{\mathrm{T}}$). If we approximate the muscle's response as a linear time invariant transformation $h(t)$ we can express

$$y(t) = h(t) * x(t) \tag{86}$$

where $*$ is the convolution operator. Each of these signals can be transformed into the frequency-domain (**Oppenheim et al., 1996**) by taking the Fourier transform $\mathcal{F}(\cdot)$ of the time-domain signal, which produces a complex (with real and imaginary parts) signal. Since convolution in the time-domain corresponds to multiplication in the frequency-domain, we have

$$Y(s) = H(s)X(s). \tag{87}$$

In **Equation 87**, we are interested in solving for $H(s)$. While it might be tempting to evaluate $H(s)$ as

$$H(s) = \frac{Y(s)}{X(s)} \tag{88}$$

the result will poorly estimate $H(s)$ because $Y(s)$ is only approximated by $H(s)X(s)$: $Y(s)$ may contain nonlinearities, non-stationary signals, and noise that cannot be described by $H(s)X(s)$.

Using cross-spectral densities, **Koopmans, 1995** (pg. 140) derived the estimator

$$H(s) = \frac{G_{yx}}{G_{xx}} \tag{89}$$

that minimizes the squared errors between $Y(s)$ and its linear approximation of $H(s)X(s)$. The cross-spectral density $G_{xy}$ between $x(t)$ and $y(t)$ is given by

$$G_{xy} = \mathcal{F}\left(x(t) \star y(t)\right) \tag{90}$$

the Fourier transform of the cross-correlation ($\star$) between $x(t)$ and $y(t)$. When the order of $x(t)$ and $y(t)$ are reversed in **Equation 90** the result is $G_{yx}$, while $G_{xx}$ and $G_{yy}$ are produced by taking the Fourier transform of $x(t) \star x(t)$ and $y(t) \star y(t)$, respectively.

Although the estimator of **Koopmans, 1995** is a great improvement over **Equation 88**, the accuracy of the estimate can be further improved using Welch's method (**Welch, 1967**). Welch's method (**Welch, 1967**) breaks up the time domain signal into K segments, transforms each segment into the frequency domain, and returns the average across all segments. Using Welch's method (**Welch, 1967**) with K segments allows us to evaluate

$$H(s) = \frac{G_{yx}^{\mathrm{K}}}{G_{xx}^{\mathrm{K}}} \tag{91}$$

which has a lower frequency resolution than **Equation 89**, but an improved accuracy in $H(s)$. Now we can evaluate the gain of $H(s)$ as

$$\left|H(s)\right| = \sqrt{\mathbb{R}(H(s))^2 + \mathbb{I}(H(s))^2} \tag{92}$$

while the phase of $H(s)$ is given by

$$\phi = \arctan\left(\frac{\mathbb{I}\left(H(s)\right)}{\mathbb{R}\left(H(s)\right)}\right) \tag{93}$$

where $\mathbb{R}(H(s))$ and $\mathbb{I}(H(s))$ are the real and imaginary parts of $H(s)$, respectively.

The transfer function estimated in *Equation 91* is meaningful only when $y(t)$ can be approximated as a linear time-invariant function of $x(t)$. By evaluating the coherence (*Koopmans, 1995*) (pg. 137) between $x(t)$ and $y(t)$

$$C_{xy}(s) = \frac{\left|G_{xy}(s)\right|}{\sqrt{G_{yy}(s)G_{xx}(s)}} \tag{94}$$

we can determine the strength of the linear association between $X(s)$ and $Y(s)$ at each frequency. When $C_{xy}$ is close to 1 it means that $Y(s)$ is well approximated by $H(s)X(s)$. As $C_{xy}$ approaches 0, it means that the approximation of $Y(s)$ by $H(s)X(s)$ becomes poor.

*Kirsch et al., 1994* analyzed a bandwidth that spanned from 4 Hz up to the cutoff frequency of the low-pass filter applied to the input signal $x(t)$ (15 Hz, 35 Hz, and 90 Hz). Unfortunately, we cannot use this bandwidth directly when analyzing model output because we have no guarantee that the simulated output is sufficiently linear in this range. Instead, to strike a balance between accuracy and consistency with *Kirsch et al., 1994*, we analyze the bandwidth that is common to the defined range of *Kirsch et al., 1994* and has the minimum acceptable $(C_{xy})^2$ of 0.67 that is pictured in *Kirsch et al., 1994*.

# Appendix 5

## Simulation summary data of *Kirsch et al., 1994*

**Appendix 5—table 1.** Mean normalized stiffness coefficients (A.), mean normalized damping coeffcients (B.), VAF (C.), and the bandwidth (D.) of linearity (coherence squared >0.67) for models with elastic-tendons.

Here, the proposed model has been fitted to Figure 12 of *Kirsch et al., 1994*, while the experimental data from *Kirsch et al., 1994* comes from *Figures 9 and 10*. Experimental data from Figure 12 of *Kirsch et al., 1994* has not been included in this table because it would only contribute 1 entry and would overwrite values from *Figures 9 and 10*. The impedance experiments at each combination of perturbation amplitude and frequency have been evaluated at 10 different nominal forces linearly spaced between 2.5 N and 11.5 N. The results presented in the table are the mean values of these ten simulations. The VAF is evaluated between the model and the spring-damper of best fit, rather than to the response of biological muscle (which was not published by *Kirsch et al., 1994*). Finally, model values for the VAF (C.) and the bandwidth of linearity (D.) that are worse than those published by *Kirsch et al., 1994* appear in bold font.

| | Kirsch et al. | | | Model | | | Hill | | |
|---|---|---|---|---|---|---|---|---|---|
| A. Norm. Stiffness $\left(\frac{K}{F}\right)$ | 15 Hz | 35 Hz | 90 Hz | 15 Hz | 35 Hz | 90 Hz | 15 Hz | 35 Hz | 90 Hz |
| 0.4 mm | 0.56 | 0.85 | 0.87 | 0.33 | 0.32 | 0.28 | 0.45 | 1.02 | 1.68 |
| 0.8 mm | 0.46 | | | 0.30 | 0.30 | 0.24 | 0.30 | 0.66 | 1.37 |
| 1.6 mm | 0.28 | 0.38 | 0.50 | 0.22 | 0.23 | 0.19 | 0.18 | 0.36 | 0.96 |
| | | | | | | | | | |
| B. Norm. damping $\left(\frac{\beta}{F}\right)$ | 15 Hz | 35 Hz | 90 Hz | 15 Hz | 35 Hz | 90 Hz | 15 Hz | 35 Hz | 90 Hz |
| 0.4 mm | 0.0118 | 0.0049 | 0.0038 | 0.0059 | 0.0044 | 0.0039 | 0.0196 | 0.0105 | 0.0029 |
| 0.8 mm | 0.0118 | 0.0049 | 0.0038 | 0.0060 | 0.0044 | 0.0039 | 0.0157 | 0.0098 | 0.0033 |
| 1.6 mm | 0.0118 | 0.0049 | 0.0038 | 0.0062 | 0.0045 | 0.0039 | 0.0112 | 0.0079 | 0.0029 |
| | | | | | | | | | |
| VAF (%) | 15 Hz | 35 Hz | 90 Hz | 15 Hz | 35 Hz | 90 Hz | 15 Hz | 35 Hz | 90 Hz |
| 0.4 mm | | | | **64** | **74** | 86 | **28** | **44** | 75 |
| 0.8 mm | | 78–99% | | **74** | 85 | 94 | **37** | **38** | 68 |
| 1.6 mm | | | | 76 | 91 | 97 | **48** | **35** | 62 |
| | | | | | | | | | |
| Bandwidth (Hz) s.t. Coherence²>0.67 | 15 Hz | 35 Hz | 90 Hz | 15 Hz | 35 Hz | 90 Hz | 15 Hz | 35 Hz | 90 Hz |
| 0.4 mm | 4–15 | 4–35 | 4–90 | 4–15 | 4–35 | 4–90 | 4–15 | 4–35 | 4–90 |
| 0.8 mm | 4–15 | 4–35 | 4–90 | 4–15 | 4–35 | 4–90 | 4–15 | 4–35 | **7**–90 |
| 1.6 mm | 4–15 | 4–35 | 4–90 | 4–15 | 4–35 | 4–90 | 4–15 | 4–35 | **7**–90 |

**Appendix 5—table 2.** Mean normalized stiffness coefficients (A.), mean normalized damping coeffcients (B.), VAF (C.), and the bandwidth (D.) of linearity (coherence squared >0.67) for models with rigid tendons.

All additional details are identical to those of *Appendix 5—table 1* except the tendon of the model is rigid.

| | Kirsch et al. | | | Model | | | Hill | | |
|---|---|---|---|---|---|---|---|---|---|
| A. Norm. Stiffness $\left(\frac{K}{F}\right)$ | 15 Hz | 35 Hz | 90 Hz | 15 Hz | 35 Hz | 90 Hz | 15 Hz | 35 Hz | 90 Hz |
| 0.4 mm | 0.56 | 0.85 | 0.87 | 0.32 | 0.31 | 0.28 | 0.05 | 0.01 | 0.01 |
| 0.8 mm | 0.46 | | | 0.30 | 0.29 | 0.25 | 0.05 | 0.00 | 0.01 |
| 1.6 mm | 0.28 | 0.38 | 0.50 | 0.23 | 0.23 | 0.20 | 0.04 | 0.00 | 0.02 |
| B. Norm. damping $\left(\frac{\beta}{F}\right)$ | 15 Hz | 35 Hz | 90 Hz | 15 Hz | 35 Hz | 90 Hz | 15 Hz | 35 Hz | 90 Hz |
| 0.4 mm | 0.0118 | 0.0049 | 0.0038 | 0.0054 | 0.0042 | 0.0038 | 0.0217 | 0.0172 | 0.0125 |
| 0.8 mm | 0.0118 | 0.0049 | 0.0038 | 0.0055 | 0.0042 | 0.0038 | 0.0148 | 0.0111 | 0.0078 |
| 1.6 mm | 0.0118 | 0.0049 | 0.0038 | 0.0057 | 0.0043 | 0.0039 | 0.0094 | 0.0068 | 0.0046 |
| VAF (%) | 15 Hz | 35 Hz | 90 Hz | 15 Hz | 35 Hz | 90 Hz | 15 Hz | 35 Hz | 90 Hz |
| 0.4 mm | | | | 86 | 96 | 99 | 95 | 92 | 88 |
| 0.8 mm | | 78–99% | | 85 | 96 | 99 | 89 | 86 | 83 |
| 1.6 mm | | | | 82 | 95 | 99 | 85 | 82 | 79 |
| Bandwidth (Hz) s.t. Coherence²>0.67 | 15 Hz | 35 Hz | 90 Hz | 15 Hz | 35 Hz | 90 Hz | 15 Hz | 35 Hz | 90 Hz |
| 0.4 mm | 4–15 | 4–35 | 4–90 | 4–15 | 4–35 | 4–90 | 4–15 | 4–35 | **13**–90 |
| 0.8 mm | 4–15 | 4–35 | 4–90 | 4–15 | 4–35 | 4–90 | 4–15 | 4–35 | **13**–90 |
| 1.6 mm | 4–15 | 4–35 | 4–90 | 4–15 | 4–35 | 4–90 | 4–15 | 4–35 | **13**–90 |

# Appendix 6

## Supplementary plots: Gain and phase response rigid-tendon muscle models

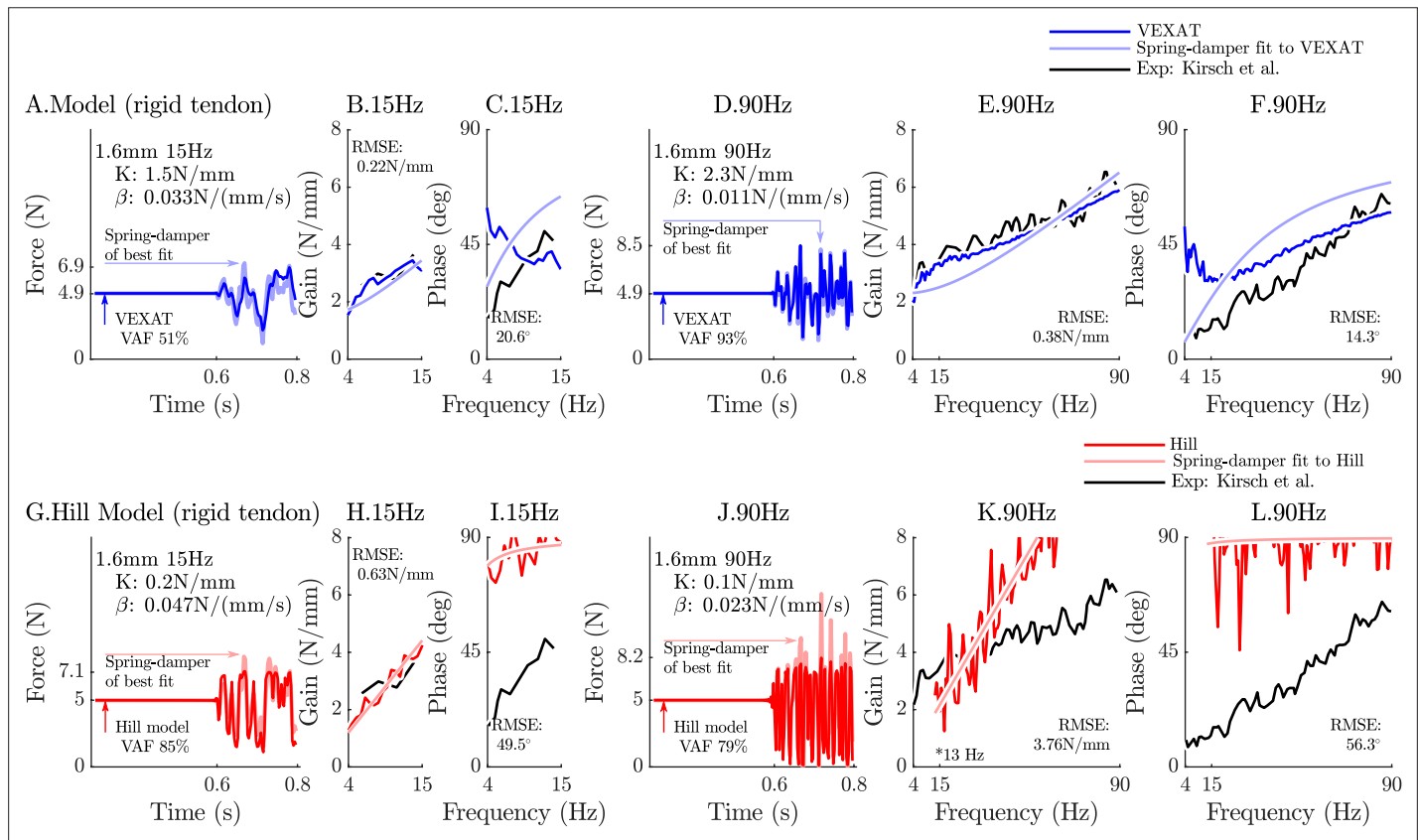

**Appendix 6—figure 1.** The response of the rigid-tendon models (VEXAT and Hill) to the 15Hz and 90Hz perturbations. When coupled with a rigid-tendon, the VEXAT model's VAF (**A**), gain response (**B**), and phase response (**C**) more closely follows the data of Figure 3 from *Kirsch et al., 1994* than when an elastic-tendon is used. This improvement in accuracy is also observed at the 90 Hz perturbation (**D, E, and F**), though the phase response of the model departs from the data of *Kirsch et al., 1994* for frequencies lower than 30 Hz. Parts of the Hill model's response to the 15 Hz perturbation are better with a rigid-tendon, with a higher VAF (**G**), a lower RMSE gain-response (**H**). but have a poor phase-response (**I**). In response to the higher frequency perturbations, the Hill model's response is poor with an elastic (see *Figure 6*) or rigid-tendon. The VAF in response to the 90 Hz perturbation remains low (**J**), and neither the gain (**K**) nor the phase response of the Hill model (**L**) follow the data of *Kirsch et al., 1994*. The rigid-tendon Hill model's nonlinearity was so strong that the lowest frequency analyzed had to be raised from 4 Hz to 21 Hz to meet the criteria that $(C_{xy})^2 \geq 0.67$.

# Appendix 7

## Supplementary plots: active lengthening on the descending limb

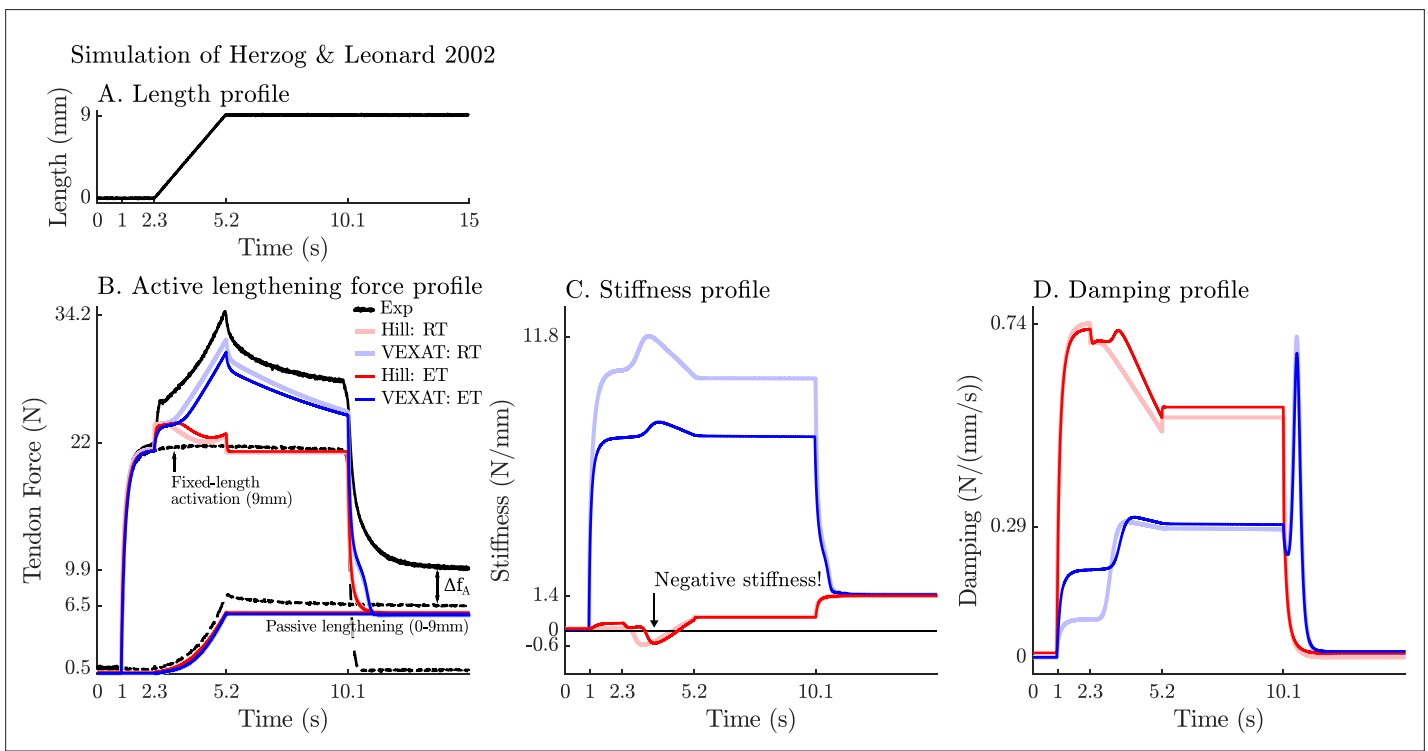

**Appendix 7—figure 1.** The response of the VEXAT and Hill models to active lengthening on the descending limb at 3 mm/s. Simulation results of the 3 mm/s (**A**) active lengthening experiment of *Herzog and Leonard, 2002* (**B**). As with the 9 mm/s trial, the Hill model's force response drops during the ramp due to a small region of negative stiffness introduced by the descending limb of the force-length curve (**C**), and a reduction in damping (**D**) due to the flattening of the force-velocity curve. Note: neither model was fitted to this trial.

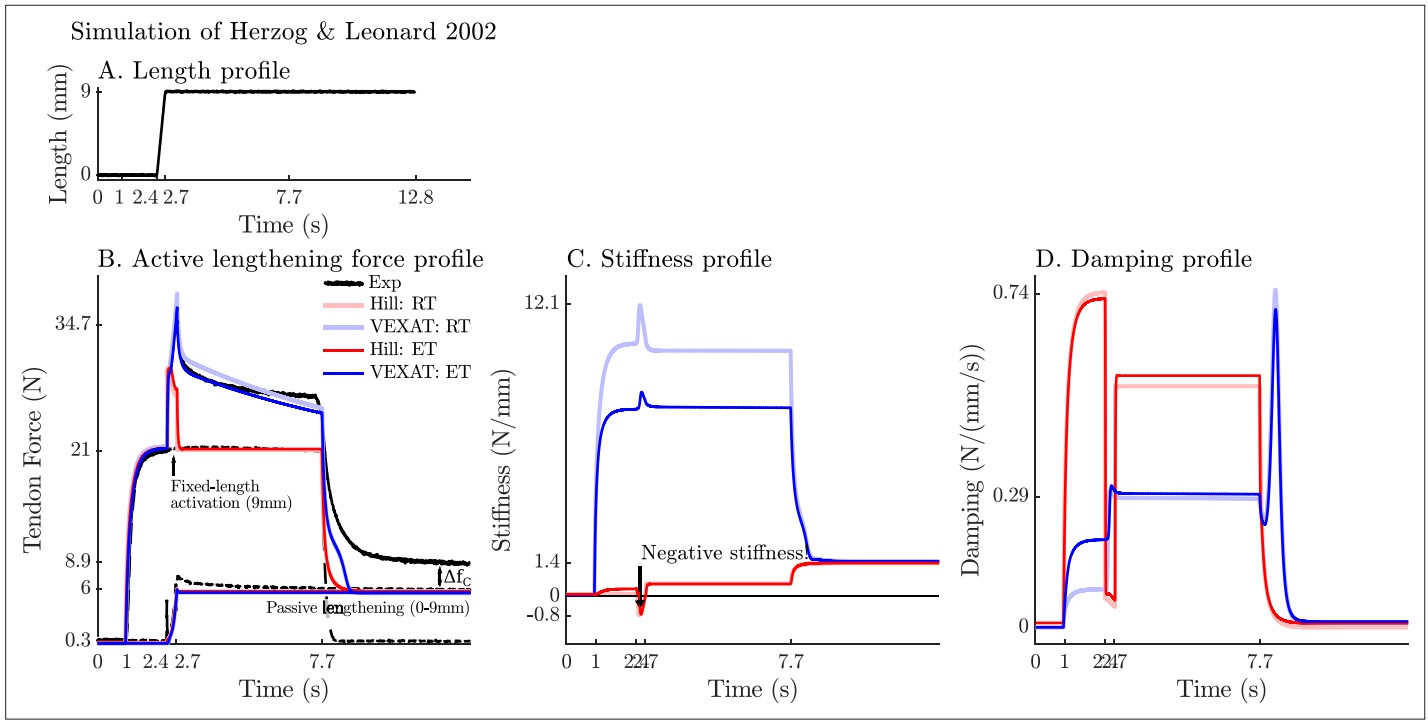

**Appendix 7—figure 2.** The response of the VEXAT and Hill models to active lengthening on the descending limb at 27 mm/s. Simulation results of the 27 mm/s (**A**) active lengthening experiment of *Herzog and Leonard, 2002* (**B**). As with the prior simulations the Hill model exhibits a small region of negative stiffness introduced by the descending limb of the force-length curve (**C**) and a drop in damping (**D**). Note: neither model was fitted to this trial.

# Appendix 8

## Rabbit psoas fibril model parameters

**Appendix 8—table 1.** The VEXAT and Hill model's fitted rabbit psoas fibril MTU parameters. As in *Table 1*, parameters shared by the VEXAT and Hill model are highlighted in grey. Short forms are used to save space: length 'len', velocity 'vel', acceleration 'acc', half 'h', activation 'act', segment 'seg', threshold 'thr', and stiffness 'stiff'. The letter preceding a reference indicates the experimental animal:'C' for cat, 'H' for human, while nothing at all is rabbit skeletal muscle. References are in brackets and are coded in order of appearance as: 'M13' for *Millard et al., 2013*, 'H95' for *Higuchi et al., 1995*, 'L10' for *Leonard et al., 2010*, 'T98' for *Trombitás et al., 1998b*, and 'P05' for *Prado et al., 2005*. Letters following a reference indicate how the data was used to evaluate the parameter: 'A' for arbitrary for simulating *Leonard et al., 2010*, 'n/a' for a parameter that is not applicable to a fibril model, '—' value taken from the cat soleus MTU, 'C' calculated, 'F' fit, 'E' estimated, 'S' scaled, and 'D' for default if a default value from another model was used. Clearly the parameters that appear in this Table do not represent a generic rabbit psoas fibril model, but instead a rabbit psoas fibril model that is sufficient to simulate the experiment of *Leonard et al., 2010*. Finally, values for $N^{IgP}$, $N^{PEVK}$, and $N^{IgD}$ were obtained by taking a 70% and 30% average of the values for 3300 kD and 3400 kD titin to match the composition of rabbit psoas titin as closely as possible.

| Parameter | | Value | Source |
|---|---|---|---|
| | **A. Basic parameters** | | |
| Max iso force | $f_o^M$ | 1 N | A |
| Opt CE len | $\ell_o^M$ | 1 mm | A |
| Pen angle | $\alpha$ | 0° | A |
| Act time const | $\tau_A$ | 10 ms | A,H(M13)D |
| De-act time const | $\tau_D$ | 40 ms | A,H(M13)D |
| | **B. Force-velocity relation: $\mathbf{f}^V(\tilde{v}^M)$** | | |
| Max shortening vel | $v_{max}^M$ | $4.5\,\frac{\ell_o^M}{s}$ | H(M13)D |
| $\mathbf{f}^V$ at $-\frac{1}{2}v_{max}^M$ | $\tilde{f}_1^V$ | $0.1 f_o^M$ | H(M13)D |
| $\mathbf{f}^V$ at $\hat{v}^M = +0$ | $\tilde{f}_2^V$ | $1.3 f_o^M$ | H(M13)D |
| $\mathbf{f}^V$ at $v_{max}^M$ | $\tilde{f}_3^V$ | $1.45 f_o^M$ | H(M13)D |
| $v_{max}^M$ scaling | $s^V$ | 0.950 | — |
| | **C. Active force-length relation: $\mathbf{f}^L(\tilde{\ell}^M)$** | | |
| Opt sarcomere len | $L_o^M$ | 2.46 $\mu$m | (H95) |
| Actin len | $\tilde{L}^A$ | $0.455\,\ell_o^M$ | (H95) |
| Myosin h-len | $\tilde{L}^M$ | $0.331\,\ell_o^M$ | (H95) |
| Myosin bare h-len | $\tilde{L}^B$ | $0.0163\,\ell_o^M$ | (H95) |
| Offset | $\Delta^L$ | $-\frac{2}{\tilde{k}_o^X}\ell_o^M$ | C |
| | **D. Passive force-length relation: $\mathbf{f}^{PE}(\tilde{\ell}^M)$** | | |

*Appendix 8—table 1 Continued on next page*

*Appendix 8—table 1 Continued*

| Parameter | | Value | Source |
|---|---|---|---|
| Slack len | $\tilde{\ell}_s^{PE}$ | $1\,\ell_o^M$ | (L10)F |
| Toe len | $\tilde{\ell}_{toe}^{PE}$ | $1.71\,\ell_o^M$ | (L10)F |
| Toe force | $\tilde{f}_{toe}^{PE}$ | $0.31 f_o^M$ | (L10)F |
| Toe stiffness | $\tilde{k}_{toe}^{PE}$ | $0.870\,\frac{f_o^M}{\ell_o^M}$ | (L10)F |
| E. Compressive force-length relation: $\mathbf{f}^{KE}(\tilde{\ell}^M)$ | | | |
| Slack len | $\tilde{\ell}_s^{PE}$ | $\frac{1}{10}\ell_o^M$ | E |
| Toe len | $\tilde{\ell}_{toe}^{PE}$ | $0.00\,\ell_o^M$ | E |
| Toe force | $\tilde{f}_{toe}^{PE}$ | $1.00 f_o^M$ | E |
| F. XE viscoelastic model | | | |
| Stiffness | $\tilde{k}_o^X$ | $49.1\,\frac{f_o^M}{\ell_o^M}$ | — |
| Damping | $\tilde{\beta}_o^X$ | $0.347\,\frac{f_o^M}{\ell_o^M/s}$ | — |
| Acc. time const | $\tau^S$ | 1.00e-3 s | — |
| Num acc damping | D | 1.00 | — |
| Low act threshold | $a_L$ | 0.0500 | — |
| Len tracking gain | $G_L$ | $1000\,\frac{1}{s}$ | — |
| Vel tracking gain | $G_V$ | 1000 | — |
| G. Titin & ECM Parameters | | | |
| ECM fraction | P | 0 | E |
| PEVK attach pt | Q | 0.675 | (L10)F |
| Z-line–T12 len | $\tilde{L}^{T12}$ | $0.0407\,\ell_o^M$ | H(T98) |
| IgD rigid h-len | $\tilde{L}^{IgD}$ | $\tilde{L}^M$ | (H95) |
| No IgP domains | $N^{IgP}$ | 45.1 | (P05)C |
| No PEVK residues | $N^{PEVK}$ | 695 | (P05)C |
| No IgD domains | $N^{IgD}$ | 22 | (P04)C |
| Active damping | $\beta_A^{PEVK}$ | $975\,\frac{f_o^M}{\ell_o^M}$ | (L10)F |
| Passive damping | $\beta_P^{PEVK}$ | $0.1\,\frac{f_o^M}{\ell_o^M}$ | — |
| Length threshold | $\tilde{\ell}_s^M$ | $\frac{1}{2}\tilde{\ell}_s^{PE}$ | — |
| Act threshold | $A_o$ | 0.05 | — |
| Step transition | R | 0.01 | — |
| H. Titin's force-length relations: $\mathbf{f}^1(\tilde{\ell}^1)$ & $\mathbf{f}^2(\tilde{\ell}^2)$ | | | |

*Appendix 8—table 1 Continued on next page*

*Appendix 8—table 1 Continued*

| Parameter | | Value | Source |
|---|---|---|---|
| $\mathbf{f}^1(\tilde{\ell}^1)$ slack len | $\tilde{\ell}_S^1$ | $0.137\,\ell_o^M$ | H(T98)S,(L10)F |
| $\mathbf{f}^1(\tilde{\ell}^1)$ toe len | $\tilde{\ell}_{toe}^1$ | $0.264\,\ell_o^M$ | H(T98)S,(L10)F |
| $\mathbf{f}^1(\tilde{\ell}^1)$ toe force | $\tilde{f}_{toe}^1$ | $0.163\,f_o^M$ | H(T98)S,(L10)F |
| $\mathbf{f}^1(\tilde{\ell}^1)$ toe stiff | $\tilde{k}_{toe}^1$ | $2.55\,\frac{f_o^M}{\ell_o^M}$ | H(T98)S,(L10)F |
| $\mathbf{f}^2(\tilde{\ell}^2)$ slack len | $\tilde{\ell}_S^2$ | $0.067\,\ell_o^M$ | H(T98)S,(L10)F |
| $\mathbf{f}^2(\tilde{\ell}^2)$ toe len | $\tilde{\ell}_{toe}^2$ | $0.129\,\ell_o^M$ | H(T98)S,(L10)F |
| $\mathbf{f}^2(\tilde{\ell}^2)$ toe force | $\tilde{f}_{toe}^2$ | $0.163\,f_o^M$ | H(T98)S,(L10)F |
| $\mathbf{f}^2(\tilde{\ell}^2)$ toe stiff | $\tilde{k}_{toe}^2$ | $5.25\,\frac{f_o^M}{\ell_o^M}$ | H(T98)S,(L10)F |

## Appendix 9

### Article footnotes

### Note 1
Small in the context of an LTI system is larger than the short-range as defined by *Rack and Westbury, 1974*: the response of an LTI system can include both length and velocity dependence, while the short-range of *Rack and Westbury, 1974* ends where velocity dependence begins.

### Note 2
A Matlab implementation of the model and all simulated experiments are available from https://github.com/mjhmilla/Millard2023VexatMuscle under the branch elife2023. All of the code is available under the APACHE-2 and MIT software licenses as indicated in the repository.

### Note 3
The term rheological is used because the model includes a component that deforms with plastic flow in response to an applied force.

### Note 4
A change of ±4mm to a typical cat soleus with an $\ell_\text{o}^\text{M}$ = 41.7 ± 1.3mm (*Sacks and Roy, 1982*).

### Note 5
8 – 20 mm⁄s ($v_\text{max}^\text{M}$) for a muscle with a maximum shortening velocity of 180 mm⁄s (*Forcinito et al., 1998*).

### Note 6
Although activation normally refers to the presence of $Ca^{2+}$ ions in the sarcomere, $Ca^{2+}$ ions alone are insufficient to cause titin to develop enhanced lengthening forces. In addition, crossbridge attachment appears to be necessary: when crossbridge attachment is inhibited titin is not able to develop enhanced forces in the presence of $Ca^{2+}$ during lengthening (*Leonard et al., 2010*).

### Note 7
Which means that the second derivative of the curve is continuous.

### Note 8
For readers who require an activation model with continuity to the second-derivative, the model of *De Groote et al., 2016* is recommended.

### Note 9
Note that we have used the symbols D, and not β, because the D terms damp the acceleration of actin-myosin movement and as such cannot be interpreted as a viscous damping term. In contrast, viscous damping terms are indicated using the β symbol.

### Note 10
Physically this assumption is equivalent to treating the CE and the tendon as massless. In general, this assumption is quite reasonable since a cubic centimeter of muscle has a mass of roughly 1.0 g but can generate tensions of between 35-137 N (*Buchanan, 1995*). With such a low mass and a high maximum isometric force, the cubic centimeter of muscle would have to be accelerated at an incredible 3500-13,700 m/s² before the inertial forces would be within 10% of the maximum isometric tension. Since everyday movements require comparatively tiny accelerations, ignoring inertial forces of muscle results in relatively small errors.

### Note 11
The impedance ($z$) of two serially connected components ($z_1$ and $z_2$) is given by $1/z = 1/z_1 + 1/z_2$, or $z = (z_1 z_2)(z_1 + z_2)$.

## Note 12
*Kirsch et al., 1994* note on page 765 a VAF of 88-99% for the medial gastrocnemius, and 8-10% lower for the soleus.

## Note 13
For brevity we will refer to the -3 dB frequency of the perturbation waveform rather than the entire bandwidth.

## Note 14
Figure 8 of *Prado et al., 2005* shows titin's contribution ranging from values ranging from (24%-57%) which means that the ECM's contribution ranges from (43%-76%).

## Note 15
Referred to as contour lengths in a worm-like chain model (*Trombitás et al., 1998b*).

## Note 16
Rabbit psoas titin (*Prado et al., 2005*) attaches at the Z-line with a 100nm rigid segment that spans to T12 epitope, is followed by 50 Ig domains, 800 PEVK residues, and another 22 Ig domains until it attaches to the 800 nm half-myosin filament which can also be considered rigid. If the Ig domains were all unfolded (adding around 25 nm; *DuVall et al., 2013*) and each PEVK residue could reach a maximum length of between 0.32nm (see Figure 5 of *Trombitás et al., 1998a* where the PEVK segment reaches a length of 700nm which is a strain of 0.32 nm for each of the 2174 residues) to 0.38 nm (*Cantor and Schimmel, 1980*, pg. 254), two titins in series would reach a length of 2(100nm + 72(25nm) + 800(0.32nm-0.38nm) + 800 nm) = 5192-6008nm. Since rabbit sarcomeres have an $\ell_o^M$ of 2.2 μm a sarcomere could be stretched to a length between 5192-6008nm, or 2.4-2.7 $\ell_o^M$, before the contour lengths of the tandem Ig and PEVK segments is reached.

