## [Editor Report · eLife assessment]

This is a **valuable** study that develops a new model of the way muscle responds to perturbations, synthesizing models of how it responds to small and large perturbations, both of which are used to predict how muscles function for stability but also how they can be injured, and which tend to be predicted poorly by classic Hill-type models. The evidence presented to support the model is **solid**, since it outperforms Hill-type models in a variety of conditions. Although the combination of phenomenological and mechanistic aspects of the model may sometimes make it challenging to interpret the output, the work will be of interest to those developing realistic models of the stability and control of movement in humans or other animals.

---

## [Referee Report · Reviewer #1 (Public review)]

Muscle models are important tools in the fields of biomechanics and physiology. Muscle models serve a wide variety of functions, including validating existing theories, testing new hypotheses, and predicting forces produced by humans and animals in health and disease. This paper attempts to provide an alternative to Hill-type muscle models that includes contributions of titin to force enhancement over multiple time scales. Due to the significant limitations of Hill-type models, alternative models are needed and therefore the work is important and timely.

The effort to include a role for titin in muscle models is a major strength of the methods and results. The results clearly demonstrate the weaknesses of Hill models and the advantages of incorporating titin into theoretical treatments of muscle mechanics. Another strength is to address muscle mechanics over a large range of time scales.

The authors succeed in demonstrating the need to incorporate titin in muscle models, and further show that the model accurately predicts in situ force of cat soleus (Kirsch et al. 1994; Herzog & Leonard, 2002) and rabbit posts myofibrils (Leonard et al. 2010). However, it remains unclear whether the model will be practical for use with data from different muscles or preparations. Several ad hoc modifications were described in the paper, and the degree to which the model requires parameter optimization for different muscles, preparations and experiment types remains unclear.

---

## [Referee Report · Reviewer #2 (Public review)]

This model of skeletal muscle includes springs and dampers which aim to capture the effect of crossbridge and titin stiffness during the stretch of active muscle. While both crossbridge and titin stiffness have previously been incorporated, in some form, into models, this model is the first to simultaneously include both. The authors suggest that this will allow for the prediction of muscle force in response to short-, mid- and long-range stretches. All these types of stretch are likely to be experienced by muscle during in vivo perturbations, and are known to elicit different muscle responses. Hence, it is valuable to have a single model which can predict muscle force under all these physiologically relevant conditions. In addition, this model dramatically simplifies sarcomere structure to enable this muscle model to be used in multi-muscle simulations of whole-body movement.

In order to test this model, its force predictions are compared to 3 sets of experimental data which focus on short-, mid- and long-range perturbations, and to the predictions of a Hill-type muscle model. The choice of data sets is excellent and provide a robust test of the model's ability to predict forces over a range of length perturbations. However, I find the comparison to a Hill-type muscle model to be somewhat limiting. It is well established that Hill-type models do not have any mechanism by which they can predict the effect of active muscle stretch. Hence, that the model proposed here represents an improvement over such a model is not a surprise. Many other models, some of which are also simple enough to be incorporated into whole-body simulations, have incorporated mechanistic elements which allow for the prediction of force responses to muscle stretch. And it is not clear from the results presented here that this model would outperform such models.

The paper begins by outlining the phenomenological vs mechanistic approaches taken to muscle modelling, historically. It appears, although is not directly specified, that this model combines these approaches. A somewhat mechanistic model of the response of the crossbridges and titin to active stretch is combined with a phenomenological implementation of force-length and force-velocity relationships. This combination of approaches may be useful improving the accuracy of predictions of muscle models and whole-body simulations, which is certainly a worthy goal. However, it also may limit the insight that can be gained. For example, it does not seem that this model could reflect any effect of active titin properties on muscle shortening. In addition, it is not clear to me, either physiologically or in the model, what drives the shift from the high stiffness in short-range perturbations to the somewhat lower stiffness in mid-range perturbations.

---

## [Author Response]

The following is the authors’ response to the previous reviews.

**eLife assessment**
This is a valuable study that develops a new model of the way muscle responds to perturbations, synthesizing models of how it responds to small and large perturbations, both of which are used to predict how muscles function for stability but also how they can be injured, and which tend to be predicted poorly by classic Hill-type models. The evidence presented to support the model is solid, since it outperforms Hill-type models in a variety of conditions. Although the combination of phenomenological and mechanistic aspects of the model may sometimes make it challenging to interpret the output, the work will be of interest to those developing realistic models of the stability and control of movement in humans or other animals.
**Reviewer #1 (Public Review):**
Muscle models are important tools in the fields of biomechanics and physiology. Muscle models serve a wide variety of functions, including validating existing theories, testing new hypotheses, and predicting forces produced by humans and animals in health and disease. This paper attempts to provide an alternative to Hill-type muscle models that includes contributions of titin to force enhancement over multiple time scales. Due to the significant limitations of Hill-type models, alternative models are needed and therefore the work is important and timely.The effort to include a role for titin in muscle models is a major strength of the methods and results. The results clearly demonstrate the weaknesses of Hill models and the advantages of incorporating titin into theoretical treatments of muscle mechanics. Another strength is to address muscle mechanics over a large range of time scales.The authors succeed in demonstrating the need to incorporate titin in muscle models, and further show that the model accurately predicts in situ force of cat soleus (Kirsch et al. 1994; Herzog & Leonard, 2002) and rabbit posts myofibrils (Leonard et al. 2010). However, it remains unclear whether the model will be practical for use with data from different muscles or preparations. Several ad hoc modifications were described in the paper, and the degree to which the model requires parameter optimization for different muscles, preparations and experiment types remains unclear.I think the authors should state how many parameters require fitting to the data vs the total number of model parameters. It would also be interesting for the authors to discuss challenges associated with modeling ex vivo and in vivo data sets, due to differences in means of stimulation vs. model inputs.(1) I think the authors should state how many parameters require fitting to the data vs the total number of model parameters.

The total number of model parameters are listed in Table 1. Each parameter has, in addition, references listed for the source of data (if one exists) along with how the data were used (’C’ calculate, ’F’ fit, ’E’ estimated, or ’S’ for scaled) for the specific simulations that appear in this paper. While this is a daunting number of parameters, only a few of these parameters must be updated when modeling a new musculotendon.

Similar to a Hill-type muscle model, at least 5 parameters are needed to fit the VEXAT model to a specific musculotendon: maximum isometric force (fiso), optimal contractile element (CE) length, pennation angle, maximum shortening velocity, and tendon slack length. However, similar to a Hill model, it is only possible to use this minimal set of parameters by making use of default values for the remaining set of parameters. The defaults we have used have been extracted from mammalian muscle (see Table 1) and may not be appropriate for modeling muscle tissue that differs widely in terms of the ratio of fast/slow twitch fibers, titin isoform, temperature, and scale.

Even when these defaults are appropriate, variation is the rule for biological data rather than the exception. It will always be the case that the best fit can only be obtained by fitting more of the model’s parameters to additional data. Standard measurements of the active force-length relation, passive forcelength relation, and force-velocity relations are quite helpful to improve the accuracy of the model to a specific muscle. It is challenging to improve the fit of the model’s cross-bridge (XE) and titin models because the data required are so rare. The experiments of Kirsch et al., Prado et al, and Trombitas et´ al. are unique to our knowledge. However, if more data become available, it is relatively straight forward to update the model’s parameters using the methods described in Appendix B or the code that appears online (https://github.com/mjhmilla/Millard2023VexatMuscle).

We have modified the manuscript to make it clear that, in some circumstances, the burden of parameter identification for the VEXAT model can be as low as a Hill model:

- Section 3: last two sentences of the 2nd paragraph, found at: Page 10, column 2, lines 1-12 of MillardFranklinHerzog v3.pdf and 05 MillardFranklinHerzog v2 v3 diff.pdf

- Table 1: last two sentences of the caption, found at: Page 11 of MillardFranklinHerzog v3.pdf and 05 MillardFranklinHerzog v2 v3 diff.pdf

(2) It would also be interesting for the authors to discuss challenges associated with modeling ex vivo and in vivo data sets, due to differences in means of stimulation vs. model inputs.

All of the experiments simulated in this work are in-situ or ex-vivo. So far the main challenges of simulating any experiment have been quite consistent across both in-situ and ex-vivo datasets: there are insufficient data to fit most model parameters to a specific specimen and, instead, defaults from the literature must be used. In an ideal case, a specimen would have roughly ten extra trials collected so that the maximum isometric force, optimal fiber length, active force-length relation, passive force-length relation (upto ≈ 0_._6_f_oM), and the force-velocity relations could be identified from measurements rather than relying on literature values. Since most lab specimens are viable for a small number of trials (with the exception of cat soleus), we don’t expect this situation to change in future.

However, if data are available the fitting process is pretty straight forward for either in-situ or ex-vivo data: use a standard numerical method (for example non-linear least squares, or the bisection method) to adjust the model parameters to reduce the errors between simulation and experiment. The main difficulty, as described in the previous paragraph, is the availability of data to fit as many parameters as possible for a specific specimen. As such, the fitting process really varies from experiment to experiment and depends mainly on the richness of measurements taken from a specific specimen, and from the literature in general.

Working from in-vivo data presents an entirely different set of challenges. When working with human data, for example, it’s just not possible to directly measure muscle force with tendon buckles, and so it is never completely clear how force is distributed across the many muscles that typically actuate a joint. Further, there is also uncertainty in the boundary condition of the muscle because optical motion capture markers will move with respect to the skeleton. Video fluoroscopy offers a method of improving the accuracy of measured boundary conditions, though only for a few labs due to its great expense. A final boundary condition remains impossible to measure in any case: the geometry and forces that act at the boundaries as muscle wraps over other muscles and bones. Fitting to in-vivo data are very difficult.

While this is an interesting topic, it is tangent to our already lengthy manuscript. Since these reviews are public, we’ll leave it to the motivated reader to find this text here.

**Reviewer #2 (Public Review):**
This model of skeletal muscle includes springs and dampers which aim to capture the effect of crossbridge and titin stiffness during the stretch of active muscle. While both crossbridge and titin stiffness have previously been incorporated, in some form, into models, this model is the first to simultaneously include both. The authors suggest that this will allow for the prediction of muscle force in response to short-, mid- and long-range stretches. All these types of stretch are likely to be experienced by muscle during in vivo perturbations, and are known to elicit different muscle responses. Hence, it is valuable to have a single model which can predict muscle force under all these physiologically relevant conditions. In addition, this model dramatically simplifies sarcomere structure to enable this muscle model to be used in multi-muscle simulations of whole-body movement.In order to test this model, its force predictions are compared to 3 sets of experimental data which focus on short-, mid- and long-range perturbations, and to the predictions of a Hill-type muscle model. The choice of data sets is excellent and provide a robust test of the model’s ability to predict forces over a range of length perturbations. However, I find the comparison to a Hill-type muscle model to be somewhat limiting. It is well established that Hill-type models do not have any mechanism by which they can predict the effect of active muscle stretch. Hence, that the model proposed here represents an improvement over such a model is not a surprise. Many other models, some of which are also simple enough to be incorporated into whole-body simulations, have incorporated mechanistic elements which allow for the prediction of force responses to muscle stretch. And it is not clear from the results presented here that this model would outperform such models.The paper begins by outlining the phenomenological vs mechanistic approaches taken to muscle modelling, historically. It appears, although is not directly specified, that this model combines these approaches. A somewhat mechanistic model of the response of the crossbridges and titin to active stretch is combined with a phenomenological implementation of force-length and force-velocity relationships. This combination of approaches may be useful improving the accuracy of predictions of muscle models and whole-body simulations, which is certainly a worthy goal. However, it also may limit the insight that can be gained. For example, it does not seem that this model could reflect any effect of active titin properties on muscle shortening. In addition, it is not clear to me, either physiologically or in the model, what drives the shift from the high stiffness in short-range perturbations to the somewhat lower stiffness in mid-range perturbations.(1) It is well established that Hill-type models do not have any mechanism by which they can predict the effect of active muscle stretch.

While many muscle physiologists are aware of the limitations of the Hill model, these limitations are not so well known among computational biomechanists. There are at least two reasons for this gap: there are few comprehensive evaluations of Hill models against several experiments, and some of the differences are quite nuanced. For example, active lengthening experiments can be replicated reasonably well using a Hill model if the lengthening is done on the ascending limb of the force length curve. Clearly the story is quite different on the descending limb as shown in Figure 9. Similarly, as Figure 8 shows, by choosing the right combination of tendon model and perturbation bandwidth it is possible to get reasonably accurate responses from the Hill model to stochastic length changes. Yet when a wide variety of perturbation bandwidths, magnitudes, and tendon models are tested it is clear that the Hill model cannot, in general, replicate the response of muscle to stochastic perturbations. For these reasons we think many of the Hill model’s drawbacks have not been clearly understood by computational biomechanists for many years now.

(2) Many other models, some of which are also simple enough to be incorporated into whole-body simulations, have incorporated mechanistic elements which allow for the prediction of force responses to muscle stretch. And it is not clear from the results presented here that this model would outperform such models.

We agree that it will be valuable to benchmark other models in the literature using the same set of experiments. Hopefully we, or perhaps others, will have the good fortune to secure research funding to continue this benchmarking work. This will, however, be quite challenging: few muscle models are accompanied by a professional-quality open-source implementation. Without such an implementation it is often impossible to reproduce published results let alone provide a fair and objective evaluation of a model.

(3) For example, it does not seem that this model could reflect any effect of active titin properties on muscle shortening.

The titin model described in the paper will provide an enhancement of force during a stretch-shortening cycle. This certainly would be an interesting next experiment to simulate in a future paper.

(4) In addition, it is not clear to me, either physiologically or in the model, what drives the shift from the high stiffness in short-range perturbations to the somewhat lower stiffness in mid-range perturbations.

We can only respond to what drives the frequency dependent stiffness in the model, though we’re quite interested in what happens physiologically. Hopefully that there are some new experiments done to examine this phenomena in the future. In the case of the model, the reasons are pretty straight forward: the formulation of Eqn. 16 is responsible for this shift.

Equation 16 has been formulated so that the acceleration of the attachment point of the XE is driven by the force difference between the XE and a reference Hill model (numerator of the first term in Eqn. 16) which is then low pass filtered (denominator of the first term in Eqn. 16). Due to this formulation the attachment point moves less when the numerator is small, or when the differences in the numerator change rapidly and effectively become filtered out. When the attachment point moves less, more of the CE’s force output is determined by variations in the length of the XE and its stiffness.

On the other hand, the attachment point will move when the numerator of the first term in Eqn. 16 is large, or when those differences are not short lived. When the attachment point moves to reduce the strain in the XE, the force produced by the XE’s spring-damper is reduced. As a result, the CE’s force output is less influenced by variations of the length of the XE and its stiffness.

**Reviewer #2 (Recommendations for the Authors):**
I find the clarity of the manuscript to be much improved following revision. While I still find the combination of phenomenological and mechanistic approaches to be a little limiting with regards to our understanding of muscle contraction, the revised description of small length changes makes the interpretation much less confusing.Similarly, while I agree that Hill-type models are widely used their limitations have been addressed extensively and are very well established. Hence, moving forward I think it would be much more valuable to start to compare these newer models to one another rather than just showing an improvement over a Hill model under (very biologically important) conditions which that model has no capacity to predict forces.(1) While I still find the combination of phenomenological and mechanistic approaches to be a little limiting with regards to our understanding of muscle contraction *...*

We have had to abstract some of the details of reality to have a model that can be used to simulate hundreds of muscles. In contrast, FiberSim produced by Kenneth Campbell’s group uses much less abstraction and might be of greater interest to you. FiberSim’s models include individual cross-bridges, titin molecules, and an explicit representation of the spatial geometry of a sarcomere. While this model is a great tool for testing muscle physiology questions through simulation, it is computationally expensive to use this model to simulate hundreds of muscles simultaneously.

Kosta S, Colli D, Ye Q, Campbell KS. FiberSim: A flexible open-source model of myofilament-level contraction. Biophysical journal. 2022 Jan 18;121(2):175-82. https://campbell-muscle-lab.github.io/FiberSim/

(2) Similarly, while I agree that Hill-type models are widely used their limitations have been addressed extensively and are very well established.

Please see our response 1 to Reviewer # 1.

(3) Hence, moving forward I think it would be much more valuable to start to compare these newer models to one another rather than just showing an improvement over a Hill model under (very biologically important) conditions which that model has no capacity to predict forces.

Please see our response to 2 to Reviewer #1.